# Developmental signals control chromosome segregation fidelity during pluripotency and neurogenesis by modulating replicative stress

Anchel de Jaime-Soguero [1,15], Janina Hattemer[1,15], Anja Bufe [1], Alexander Haas[2], Jeroen van den Berg[3,4,5,6], Vincent van Batenburg[3,4,5,6], Biswajit Das[7], Barbara di Marco [8], Stefania Androulaki[1], Nicolas Böhly [2], Jonathan J. M. Landry[9], Brigitte Schoell[10], Viviane S. Rosa[11], Laura Villacorta[9], Yagmur Baskan[1], Marleen Trapp[12], Vladimir Benes [9], Andrei Chabes [7], Marta Shahbazi [11], Anna Jauch[10], Ulrike Engel [13], Annarita Patrizi [12], Rocio Sotillo [14], Alexander van Oudenaarden [3,4,5,6], Josephine Bageritz [1], Julieta Alfonso[8], Holger Bastians [2,16] & Sergio P. Acebrón [1,16] ✉

Human development relies on the correct replication, maintenance and segregation of our genetic blueprints. How these processes are monitored across embryonic lineages, and why genomic mosaicism varies during development remain unknown. Using pluripotent stem cells, we identify that several patterning signals—including WNT, BMP, and FGF—converge into the modulation of DNA replication stress and damage during S-phase, which in turn controls chromosome segregation fidelity in mitosis. We show that the WNT and BMP signals protect from excessive origin firing, DNA damage and chromosome missegregation derived from stalled forks in pluripotency. Cell signalling control of chromosome segregation declines during lineage specification into the three germ layers, but re-emerges in neural progenitors. In particular, we find that the neurogenic factor FGF2 induces DNA replication stress-mediated chromosome missegregation during the onset of neurogenesis, which could provide a rationale for the elevated chromosomal mosaicism of the developing brain. Our results highlight roles for morphogens and cellular identity in genome maintenance that contribute to somatic mosaicism during mammalian development.

Organismal viability requires faithful duplication and transmission of the genetic material across different cellular lineages. As such, intrinsic cellular mechanisms tightly monitor DNA replication and repair during S-phase, as well as chromosome segregation during mitosis[1–4]. Despite these conserved cellular checkpoints, the rate and distribution of de novo genomic variations are neither low nor homogenous across all developmental lineages and cell types[5–8]. In particular, the majority of human preimplantation embryos, and up to 30% of neural progenitors during mammalian brain development, present numerical chromosomal aberrations[5,9–12], while other embryonic lineages display low levels of mosaicism. The existence of these developmental bottlenecks suggests that cell fate-dependent

---

mechanisms might differentially affect chromosome segregation fidelity during human development.

Whole-chromosome missegregation can be caused by mitotic defects in the spindle, centrosomes or chromatid cohesion, as well as increased microtubule plus end assembly rates leading to erroneous merotelic microtubule–kinetochore attachments[1,13]. Structural and partial chromosome errors can often be tracked to DNA replication stress arising from slowed or stalled replication forks during S-phase, as well as to impaired DNA repair, which can lead to ultra-fine bridges and acentric chromosomes during anaphase[14–16]. Intriguingly, DNA replication stress can link both structural and numerical chromosomal instability (CIN), including through the elevation of microtubule polymerisation dynamics in mitosis[14,17].

Cell lineage specification is controlled by morphogens, patterning signals and growth factors that form distinct gradients to shape embryos, direct tissue patterning, and regulate cell fate[18–22]. Critically, CIN risks weakening the robustness of these developmental and cellular signalling programmes[23–26]. Here, we investigate whether morphogens, patterning signals and growth factors can in turn regulate chromosomal stability during human lineage specification (Fig. 1a).

To gain access and map signalling decisions across bifurcating fate choices from pluripotency to human lineage specification, we turn into primed pluripotent stem cells, which represent a bona fide model of the epiblast before gastrulation. We show that developmental signals control chromosome segregation fidelity during pluripotency and neurogenesis, but not in other investigated embryonic lineages, by modulating the cellular response to replicative stress. We identify that WNT/GSK3 signalling sits at the helm of this regulatory cascade by protecting cells from different sources of DNA replication stress,

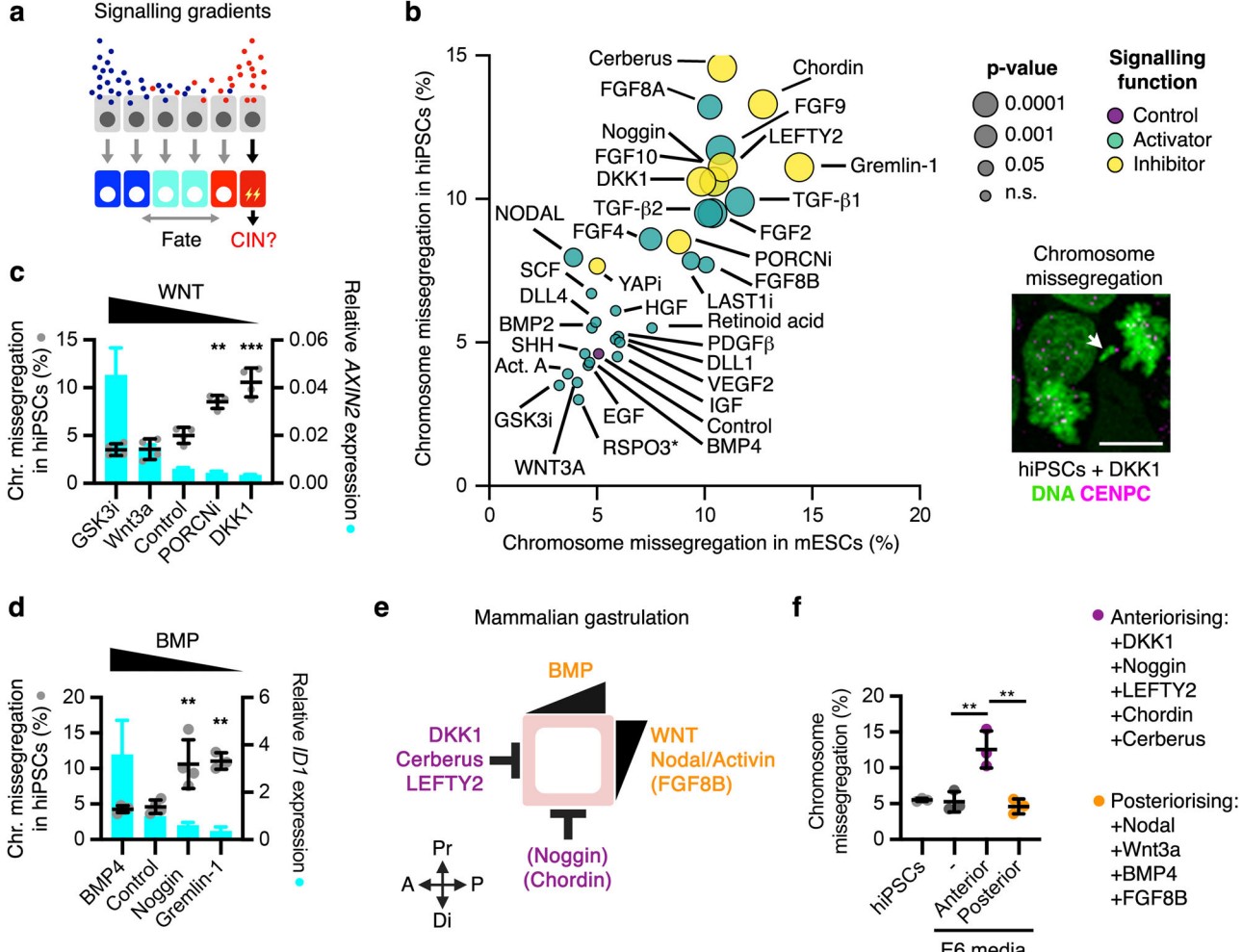

**Fig. 1 | Embryo patterning signals regulate chromosome segregation fidelity in PSCs. a** Schematic of signalling-driven patterning and the hypothesised role in CIN. **b** Chromosome segregation analyses in mESCs and hiPSCs upon treatment for 16 h with different pathway-specific activating signals (agonists), inhibiting signals (antagonists), or small molecule compounds. Data are mean and *P*-value from one-way ANOVA analyses with multiple comparisons with Dunnet corrections of *n* = 3–6 biological experiments with >100 anaphases per condition in each experiment. An example of a DKK1-treated hiPSC in anaphase is shown. DAPI stains the DNA and CENPC marks kinetochores. Scale bar = 10 μm. **c, d** Chromosome missegregation (*n* = 3-4 biological replicates) from (**b**) and representative qRT-PCR analyses of the WNT target gene *AXIN2* (**c**) or the BMP target gene *ID1* (**d**) in hiPSCs upon treatment for 16 h with the indicated compounds and proteins (the experiment was repeated three times). Data are plotted as mean ± s.d. of *n* = 3 technical replicates. *P*-values

from one-way ANOVA analyses with multiple comparisons with Dunnet corrections from the indicated groups, **P* < 0.01, ***P* < 0.001. **e** Schematic of the signalling axes driving gastrulation in mammalian embryos. Note that Chordin and Noggin do not establish the anterior visceral endoderm in mammals, but are required for subsequent anterior patterning[35] (see also Supplementary Data 1). **f** Chromosome segregation analyses in hiPSCs upon co-treatment with the indicated signals promoting anteriorisation or posteriorisation during mammalian gastrulation. Data are mean ± s.d. of *n* = 3 biological replicates, with >100 anaphases analysed in each condition per replicate). Source data for all experiments are provided as a Source data file. *P*-values from one-way ANOVA analyses with multiple comparisons and Dunnet corrections from the indicated groups. From left to right: **P* = 0.0025, **P* = 0.0011. **a, e** Created with BioRender.com released under a Creative Commons Attribution-NonCommercial-NoDerivs 4.0 International license[139].

including other signalling cascades and the perturbation of core replication components. Given the spatio-temporal patterns of activation and co-occurrence of the examined signalling cascades during mammalian development, our findings could provide a rationale for understanding why chromosomal mosaicism is prevalent in early embryogenesis and the developing brain.

## Results

### Signalling control of chromosome segregation in pluripotency

We have previously identified that WNT ligands promote chromosome segregation fidelity in somatic and cancer cells[27–29], although the underlying mechanisms and physiological relevance remained uncharacterised. We hypothesised that WNT and possibly other developmental signalling pathways could exert control of chromosomal stability in targeted embryonic lineages (Fig. 1a), thereby resulting in spatiotemporal distribution of chromosomal mosaicism during development. To examine the roles of cell signalling on chromosome segregation fidelity during pluripotency, we treated primed human induced pluripotent stem cells (hiPSCs) with morphogens, growth factors, and chemical inhibitors targeting the main developmental signalling pathways that regulate stem cell maintenance, lineage specification and embryonic patterning (Supplementary Data 1). These factors and compounds were used at physiologically relevant conditions, i.e. by inhibiting endogenous signals or activating pathways at commonly used concentrations during cell lineage specification studies (Supplementary Data 1). Beyond hiPSCs, we also analysed mouse embryonic stem cells (mESCs) to determine the conservation of the effects between mammals and across naive and primed pluripotency. We treated hiPSCs and mESCs with signalling modulators for 16 h, and analysed chromosome missegregation rates by immunofluorescence in three independent experiments for a total of ~19,000 anaphases (Fig. 1b). Inhibition of autocrine WNT signalling by its physiological antagonist DKK1, inhibition of endogenous bone morphogenetic protein (BMP) signalling using Noggin, Chordin or Gremlin-1, or inhibition of WNT, BMP and NODAL by Cerberus, increased >2-fold the chromosome missegregation rate in both mESCs and hiPSCs (Fig. 1b). Furthermore, activation of transforming growth factor (TGF) by TGF-β1/2, inhibition of Nodal signalling by LEFTY2, as well as fibroblast growth factor (FGF) signalling activation by various ligands (FGF2/4/8A/8B/9), also promoted chromosome segregation errors in mESCs and hiPSCs (Fig. 1b). For factors and compounds regulating the same pathway, signalling activity levels correlated with their effects on chromosome segregation fidelity (Fig. 1c, d), suggesting that signalling gradients could differentially impact chromosomal stability. Accordingly, co-treatment with WNT3A and DKK1 resulted in a concentration-dependent response (Supplementary Fig. 1a). Human PSCs represent a great model to study gastrulation in vitro[30]. Intriguingly, we observed that signals promoting anteriorisation (DKK1, Noggin, LEFTY2, Cerberus, and Chordin) during mammalian gastrulation were strongly associated with higher chromosome missegregation rate in pluripotent stem cells, compared with signals promoting posteriorisation (WNT3A, BMP4, Activin A/Nodal, and FGF8B) (Fig. 1e, Supplementary Fig. 1b, and Supplementary Data 1). Furthermore, co-treatment for 16 h with the anteriorising signals DKK1, Noggin, LEFTY2, Cerberus, and Chordin in E6 media, but not with a cocktail of posteriorizing signals, resulted in high levels of chromosome missegregation in hiPSCs (Fig. 1F and Supplementary Fig. 1c, d)). In summary, these analyses suggest that 5 (WNT, BMP, FGF, Nodal, and TGFβ) out of the 15 signalling pathways analysed impact chromosome segregation fidelity in pluripotent stem cells.

We decided to focus our studies on WNT, BMP, and FGF signalling (Fig. 2a), which are critical drivers of lineage specification and embryonic patterning[22,30–38]. In addition to mESCs and hiPSCs (Fig. 1b), we found that inhibition of endogenous WNT and BMP signalling using DKK1 and Noggin, respectively, as well as FGF signalling activation by

FGF2, also induced chromosome missegregation in primed human embryonic stem cells (hESCs) (Supplementary Fig. 2a, b).

To confirm the specificity of the investigated signalling cascades, we performed rescue experiments by targeting their downstream effectors. Inhibition of GSK3 using CHIR99021 (GSK3 inhibitor, GSK3i), which activates the canonical WNT pathway downstream of the receptor complex[39] (Fig. 2a), rescued DKK1-induced chromosome missegregation in hiPSCs (Fig. 2b). Furthermore, transient inhibition of the FGF receptor with a specific small compound (CAS 192705-79-6, FGFRi) (Fig. 2a), or co-treatment with BMP4, rescued chromosome missegregation in hiPSCs induced by FGF2 and Noggin, respectively (Fig. 2b). WNT3A and FGF2 trigger different molecular cascades downstream of their receptors[39,40]. Knockdown experiments highlighted a role for the canonical WNT co-receptor LRP6, but not the downstream effector β-catenin, in the control of chromosome segregation fidelity in hiPSCs, (Supplementary Fig. 2c, d). These results are in agreement with previous studies in somatic cells[27–29], and highlight a role for β-catenin independent WNT/GSK3 signalling—possibly through WNT/STOP[39,41]—in chromosome segregation fidelity. Rescue experiments with small molecule compounds targeting kinases downstream of FGF signalling supported a role for the MEK/ERK cascade in FGF2-driven chromosome missegregation (Supplementary Fig. 2c).

Morphogenetic signals often crosstalk epistatically to exert their functions in lineage specification[18,20,36]. We found that activation of the WNT pathway by WNT3A or GSK3i was sufficient to rescue not only DKK1, but also FGF2 and Noggin effects in chromosome missegregation (Fig. 2c, d and Supplementary Fig. 2e). On the other hand, additional epistasis experiments revealed that BMP4 was not able to rescue inhibition of endogenous WNT signalling activity (Fig. 2c and Supplementary Fig. 2f). Taken together, these results indicate that WNT, BMP, and FGF signalling functionally interact upstream of GSK3 to regulate chromosome segregation fidelity during human pluripotency (Fig. 2e).

Live cell imaging analyses of hiPSCs labelled with SiR-DNA confirmed that DKK1, FGF2, and Noggin treatments induce chromosome missegregation (Supplementary Fig. 2g, h), and revealed that cells undergoing chromosome missegregation completed their division and survived until G1 (Supplementary Fig. 2i). Noggin and FGF2 did not impact the length of mitosis (Supplementary Fig. 2j). However, DKK1 treatment increased by ~4 min the average time required for completion of mitosis in hiPSCs (Supplementary Fig. 2j), similarly as we have reported before in somatic cells[29,42].

In agreement with the immunofluorescence and live cell imaging studies, analyses of metaphase spreads of hiPSCs using multiplex FISH (M-FISH) and Giemsa staining revealed abnormal karyotypes after one cell cycle in the presence of DKK1, FGF2, or Noggin (Fig. 2f, g, and Supplementary Fig. 2k, l). These results show that, while exogenous activation of FGF signalling can induce aneuploidy, autocrine WNT and BMP signalling prevent chromosome instability in hiPSCs.

### WNT, BMP, and FGF regulate replicative stress during S-phase

Treatment of hiPSCs with DKK1, FGF2, or Noggin neither impacted cell cycle progression nor induced differentiation priming after 16 h (Supplementary Fig. 2m, n). Furthermore, triggering exit of pluripotency by placing cells in E6 or RPMI media for 16 h did not result in increased chromosome missegregation (Fig. 1f and Supplementary Fig. 2o), indicating that signalling roles in chromosome segregation in hiPSCs are not due to cell differentiation or proliferation. To get insights on the mode of action of WNT, BMP, and FGF, we decided to perform single-cell sequencing analyses aimed at identifying cellular processes that could be co-regulated by these pathways.

Gene ontology analyses of the combined differentially regulated genes upon endogenous WNT or BMP signalling inhibition, or FGF signalling activation, showed the highest enrichment score

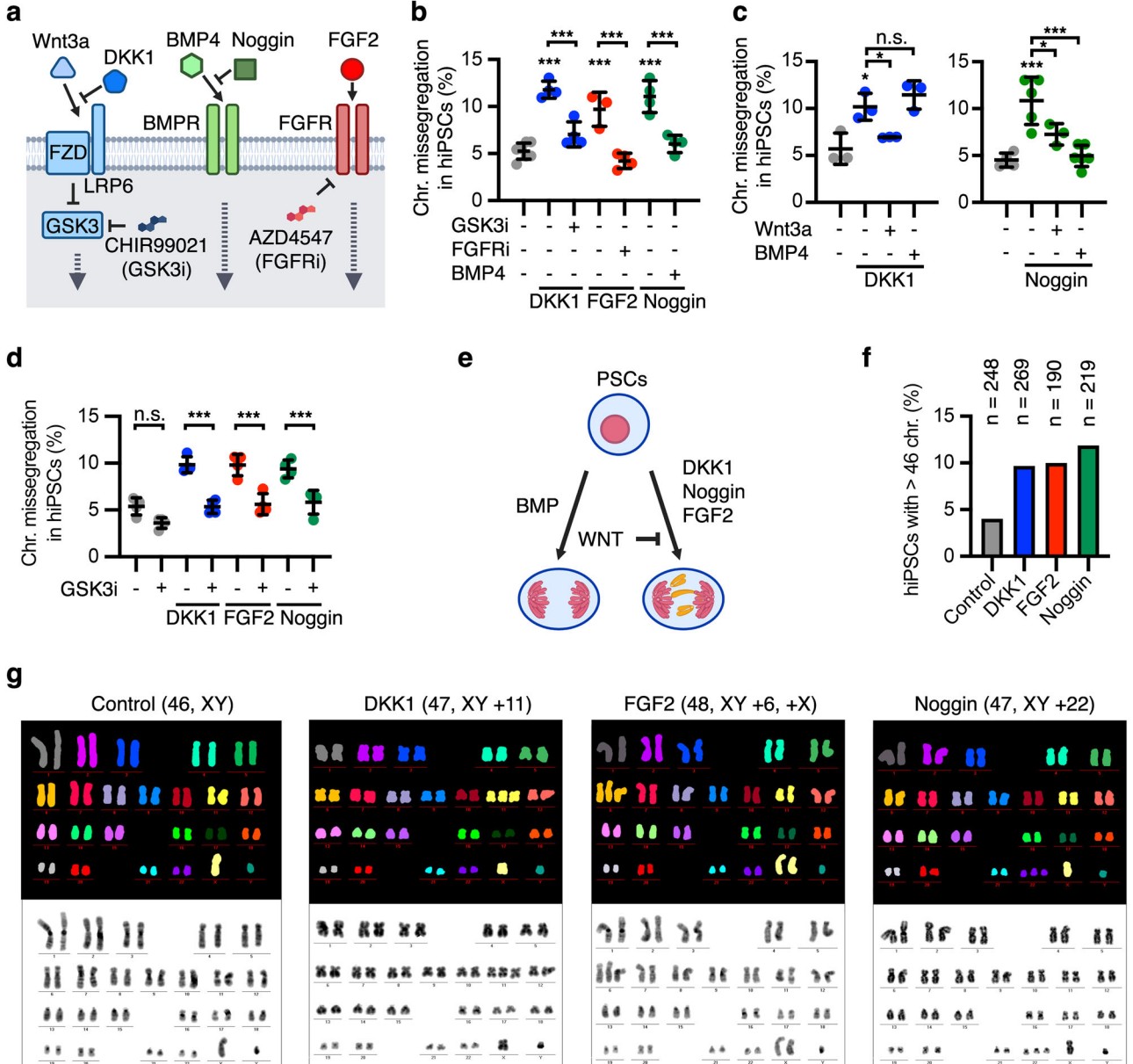

**Fig. 2 | WNT, BMP, and FGF signalling regulate chromosome segregation fidelity in PSCs. a** Simplified scheme of the WNT, BMP, and FGF signalling pathways highlighting the proteins and modulators used in this work. **b–d** Chromosome segregation analyses in hiPSCs treated as indicated for 16 h. Data are mean ± s.d. of **b** $n = 4$ biological replicates (except $n = 3$ in FGF2 condition), **c** $n = 3$ biological replicates (left panel) and $n = 3$–6 biological replicates (right panel), and **d** $n = 4$ biological replicates with >100 mitotic cells analysed per condition in each replicate. *P*-values from one-way ANOVA analyses with multiple comparisons and Dunnet corrections are indicated as *$P < 0.05$, **$P < 0.01$, ***$P < 0.001$, or n.s. ($P > 0.05$, not significant). **e** Schematic of the epistasis interactions between WNT,

FGF and BMP signalling in chromosome segregation fidelity. **f** Analyses of the chromosome gains in Giemsa staining and M-FISH experiment analyses are shown. Data represent a representative M-FISH experiment with $n = 40$ cells analysed per condition, and a total of $n = 3$ biological replicates of Giemsa staining with the following cells counted per condition (in total): Control ($n = 208$ cells), DKK1 ($n = 229$ cells), FGF2 ($n = 79$ cells) and Noggin ($n = 179$ cells). **g** M-FISH examples of hiPSCs treated as indicated. Source data for all experiments are provided as a Source data file. **a**, **e** Created with BioRender.com released under a Creative Commons Attribution-NonCommercial-NoDerivs 4.0 International license[139].

for factors promoting H3K4 methylation (7.5-fold, $P = 6.5 \times 10^{-07}$), which modulates gene expression and is associated with the mitigation of replication stress[43,44], as well as downregulation of DNA unwinding factors involved in DNA replication (7.4-fold, $P = 3 \times 10^{-06}$) and ATP-dependent chromatin remodelling factors (6.8-fold, $P = 3.8 \times 10^{-08}$) (Fig. 3a, Supplementary Fig. 3a, b, and Supplementary Data 2), suggesting a role of these pathways in chromatin remodelling and/or DNA replication. Intriguingly, pluripotent stem cells deal with elevated replicative stress under normal self-renewing conditions[45,46]. We have previously found that (i) mild DNA replication stress can lead to whole

chromosome missegregation through elevated microtubule dynamics in cancer cells[17], (ii) WNT regulates microtubule plus end dynamics through unknown mechanisms[28], and (iii) direct perturbation of WNT activity only during mitosis did not induce chromosome missegregation[42]. Furthermore, qRT-PCR analyses largely validated that 16-hour treatment with DKK1, FGF2, or Noggin downregulates the expression of the identified DNA unwinding factors involved in DNA replication in hiPSCs (Supplementary Fig. 3, c). As such, we decided to explore whether WNT, BMP, and FGF signalling function in S-phase during DNA replication. For these experiments, we performed acute

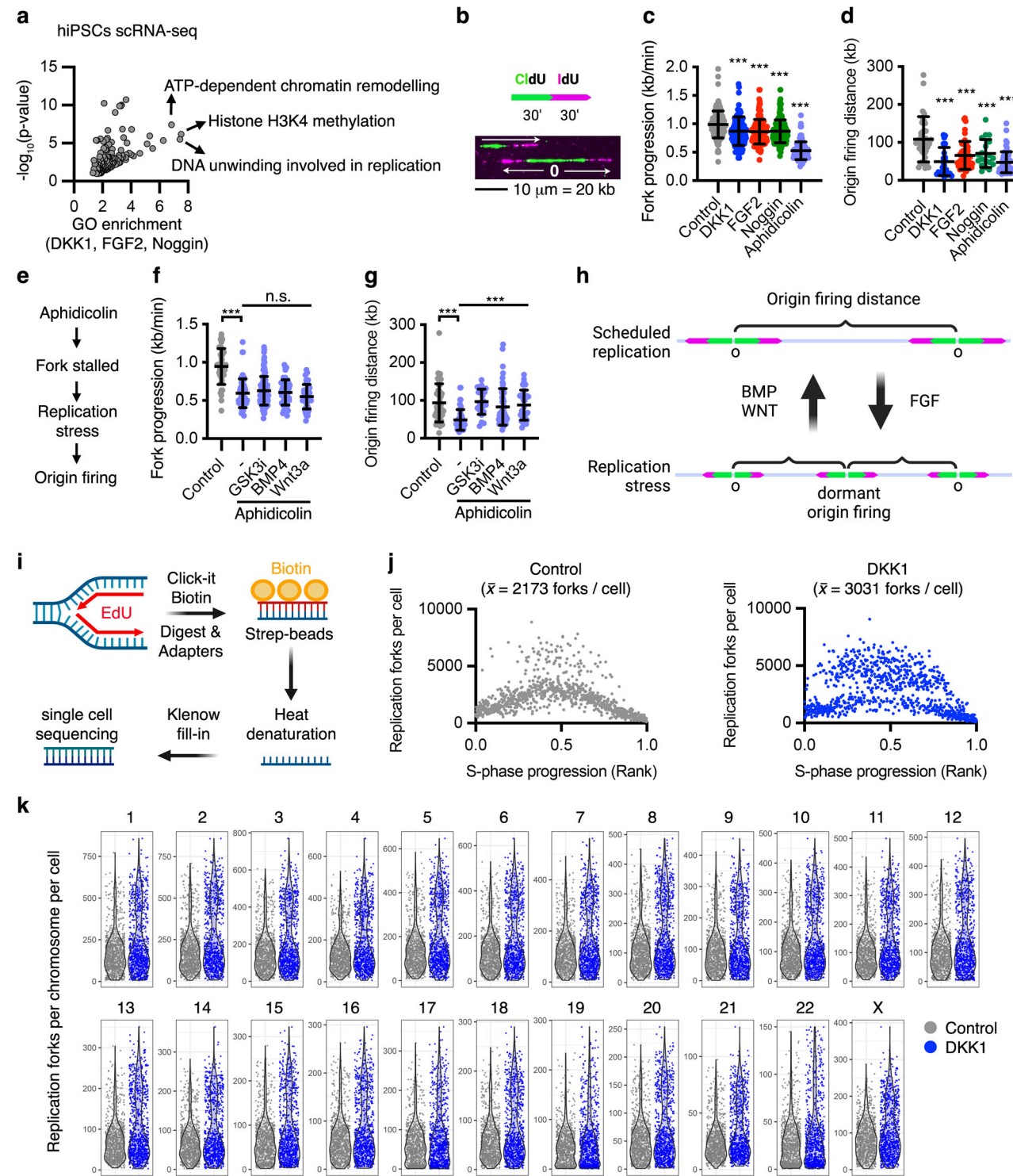

perturbations (3 h) aimed at identifying direct targets and functions of these pathways.

First, we performed single-molecule analyses of replication dynamics in hiPSCs by DNA combing (Fig. 3b). Inhibition of WNT and BMP signalling, as well as activation of FGF signalling for 3 h, modestly reduced replication fork velocity in hiPSCs between 13% and 15% (Fig. 3c). Slowed replication rates are associated with DNA replication stress response, which includes the firing of dormant origins as a compensatory mechanism to facilitate completion of DNA replication in eukaryotes[47]. We found that the distance between bi-directional forks, which serves as a readout for inter-origin

distance[48,49], was decreased from ~100 kb to 70–49 kb upon DKK1, FGF2, or Noggin treatments (Fig. 3d), similar to triggering mild DNA replication stress with a low dose of the DNA polymerase inhibitor aphidicolin (100 nM, Fig. 3d). Conversely, co-treatment with WNT3A, GSK3i, or BMP4 prevented compensatory origin firing upon 100 nM aphidicolin without affecting replication fork velocity (Fig. 3e–g), which indicates a role of WNT and BMP signalling in the mitigation of DNA replication stress downstream of stalled forks (Fig. 3h). In the absence of additional stressors, activation of WNT signalling with GSK3i did not significantly impact the inter-origin distance, but rescued DKK1 distance reduction between firing origins, suggesting that

**Fig. 3 | WNT, BMP, and FGF converge into the regulation of DNA replication stress. a** GO term enrichment analysis of common transcripts differentially upregulated by DKK1, FGF2, and Noggin in hiPSCs after 16 h treatment. Expression analyses were performed by single-cell RNA sequencing of >75 cells per condition. After normalizing the data for differences in library size, the 'FindMarkers' function with the ROC test was used to determine differentially expressed genes (DEGs) between the different treatment conditions. We selected DEGs with power > 0.25 for the subsequent GO analysis. $P$-values from GO enrichment analyses are provided directly from DAVID's EASE score: a one-sided Fisher Exact $P$-value for gene-enrichment analysis. Data is provided in Supplementary Data 2. **b–g** DNA combing experiments in hiPSCs treated as indicated for 3 h and labelled with consecutive pulses of CIdU and IdU as shown in (**b**). Each experiment of every subfigure was replicated at least twice, and the data displayed corresponds to a representative experiment, where is shown the mean ± s.d. of: control ($n = 127$), Aphidicolin ($n = 95$), DKK1 ($n = 150$), FGF2 ($n = 135$), and Noggin ($n = 153$) single forks measurements in (**c**) and control ($n = 42$), Aphidicolin ($n = 59$), Aphidicolin + WNT3A ($n = 44$), Aphidicolin + GSK3i ($n = 123$) and Aphidicolin + BMP4 ($n = 53$) single fork measurements in (**f**). In **d, g** inter-origin firing distance was assessed. In **d** control = 46, Aphidicolin = 60, DKK1 = 22, FGF2 = 48, Noggin = 19 inter-origin

distances were measured, and in **g** control ($n = 49$), Aphidicolin ($n = 49$), Aphidicolin + WNT3A ($n = 31$), Aphidicolin + GSK3i ($n = 26$) and Aphidicolin + BMP4 ($n = 59$) inter-origin firing distances were measured. In **b** examples of fork progression and origin (O) are shown. In **e** a scheme of the replication stress cascade targeted in (**f, g**) is shown. $P$-values from one-way ANOVA analyses with multiple comparisons and Dunnet corrections from the indicated groups, ***$P < 0.001$ or in (**f**) n.s. > 0.05. **h** Proposed roles of WNT, BMP, and FGF signalling in DNA replication. **i** Pipeline for single-cell EdU sequencing. Note that hiPSCs were treated first for 3 h, followed by two 15-min pulses of EdU separated by 1 h. **j** Replication forks per cell obtained by scEdU-seq in control or DKK1 (3 h) treated hiPSCs from a single biological experiment. Data corresponds to the number of forks per cell in Control ($n = 894$ cells) and DKK1 ($n = 888$ cells) after sequencing analysis. Single cells were ranked for their relative position in the S-phase ($x$-axis) according to their fork distribution pattern across different chromosomes, as previously described[50]. **k** Replication forks per chromosome per cell, as described in (**j**). Source data for all experiments are provided as a Source data file, with the exception of (**j, k**), which is included in the data repository GSE271478. **h, i** Created with BioRender.com released under a Creative Commons Attribution-NonCommercial-NoDerivs 4.0 International license[139].

---

basal WNT activity is sufficient to modulate DNA replication stress in hiPSCs (Supplementary Fig. 3d).

To further validate the roles of WNT signalling in the regulation of replication forks, we performed single-cell EdU sequencing (scEdU-seq)[50], which quantifies and assigns replicating forks across single-cell genomes (Fig. 3i). We identified an average of 2173 forks per hiPSC under basal conditions (Fig. 3j), with few cells doubling or tripling their fork numbers during mid-S-phase, possibly due to basal replicative stress[45,46]. In agreement with our inter-origin distance analyses, DKK1 treatment for 3 h increased the average number of replication forks in hiPSCs ($\bar{x} = 3031$ forks/cell). Notably, we found that the excessive number of forks largely concentrate in mid S-phase between two converging replication regions (Fig. 3j and Supplementary Fig. 4d–f). Further analyses showed that DKK1 increased the fork number in all chromosomes homogenously in the perturbed cells (Fig. 3k). These results suggest that endogenous WNT activity prevents excessive origin firing and/or facilitates termination of ongoing forks in pluripotent stem cells during DNA replication.

Previous research has shown that β-catenin-independent WNT/GSK3 signalling largely functions post-translationally by modulating protein phosphorylation[39]. To get additional functional information on how WNT, BMP, and FGF signalling impacts DNA replication, we performed phospho-proteomics and epistasis experiments upon acute perturbation of these pathways. We treated hiPSCs for 3 h with DKK1, Noggin and FGF2 in 4 independent experiments and analysed their phospho-proteome using a label-free quantification approach[51] (Supplementary Fig. 5a and Supplementary Data 3). Among the most significantly regulated phospho-peptides across the treatments, we identified several factors associated with DNA replication and damage response (Supplementary Fig. 5b). Five phospho-peptides were differentially regulated in the three treatments (Supplementary Fig. 5c), including the key DNA replication stress-associated proteins RIF1[17,52–54] and BRCA1 at the CDK1-target serine 1191, which is triggered by DNA replication stress to stabilise it[55,56] (Supplementary Fig. 3b–e). In addition to RIF1 and BRCA1, (i) FGF activation differentially regulated factors directly controlling replisome progression (Supplementary Fig. 5b, d), including the downregulation of the phosphorylation of the replication initiation factor MCM2 at the CDC7-target serine 26[57] (Supplementary Fig. 5b, d, e); (ii) BMP signalling inhibition triggered phosphorylation of EEPD1 and the DNA chaperone HMGA2, which are associated with the stability of single-stranded DNA (ssDNA) at stalled forks[58,59], and affected several factors functioning at the DNA damage response (Supplementary Fig. 5b, d); and (iii) DKK1 differentially regulated the phosphorylation of XRNA2 and SF3B1, which function at the fork in resolving R-loops during DNA replication stress[60,61], as well

as additional factors involved in the DNA damage response, including TP53BP1 (Supplementary Fig. 5b, d). Taking into account these results, we hypothesised that FGF signalling may function during origin licensing, DNA replication initiation and/or progression, while BMP and WNT might have distinct roles in the response to DNA replication stress (Supplementary Fig. 5d), which we decided to explore through (i) molecular studies of DNA replication initiation/progression and (ii) further epistasis studies using specific small molecule compounds and analyses of replication-associated DNA damage.

Failure to load MCM complex in G1 to license origins, as well as depletion of dNTP pools, are common sources of replicative stress. However, FACS analyses of MCM2 loading in G1 suggested that 3-h treatments with DKK1, FGF, and Noggin did not modulate origin licensing (Supplementary Fig. 5f, g). In addition, HPLC analyses showed that these morphogenetic signals do not affect the total cellular pools of nucleotides in unsynchronised hiPSCs (Supplementary Fig. 5h). We also did not observe a delay in S-phase progression with either treatment after synchronizing the cells (Supplementary Fig. 5i, j), confirming previous results from unsynchronized cells (Supplementary Fig. 2n). These results are consistent with our inter-origin analyses showing that WNT and BMP function downstream of stalled forks in S-phase (Fig. 3).

Unresolved DNA replication stress triggers subsequent errors, including double-strand breaks (DSBs). Phosphorylation of histone H2AX (γ-H2AX) is an early ATM-dependent response to DSBs[62,63], including after DNA replication stress in pluripotent stem cells[64]. Inhibition of autocrine WNT and BMP signalling with DKK1 and Noggin, respectively, as well as FGF activation with FGF2 for 3 h induced the formation of γ-H2AX foci in hiPSCs specifically during S-phase (Fig. 4a, b EdU⁺ cells), which is consistent with replication stress-associated DNA damage. In agreement with our inter-origin analyses (Fig. 3c–g), WNT and BMP activation—but not FGF signalling inhibition by FGFRi (Supplementary Fig. 5k)—prevented the formation of γ-H2AX foci upon partial inhibition of the DNA polymerase using 100 nM aphidicolin (Fig. 4c). Intriguingly, additional rescue experiments indicated that WNT, but not BMP signalling, protects hiPSCs in S-phase from DSBs upon acute dNTP depletion using hydroxyurea (HU) (Fig. 3d). Topoisomerase I inhibition by camptothecin (CPT) triggers recruitment and phosphorylation of the ssDNA break proteins RPA1-3 (p-RPA) at the stalled forks[65], which was rescued by BMP signalling activation, but not by WNT signalling (Fig. 4e, f). These results, together with our phospho-proteomics (Supplementary Fig. 5a–d) and previous epistasis analyses (Fig. 2a–e) support a role of FGF signalling interfering with fork progression—but not origin licensing or S-phase progression—a function of BMP signalling in protecting ssDNA at the

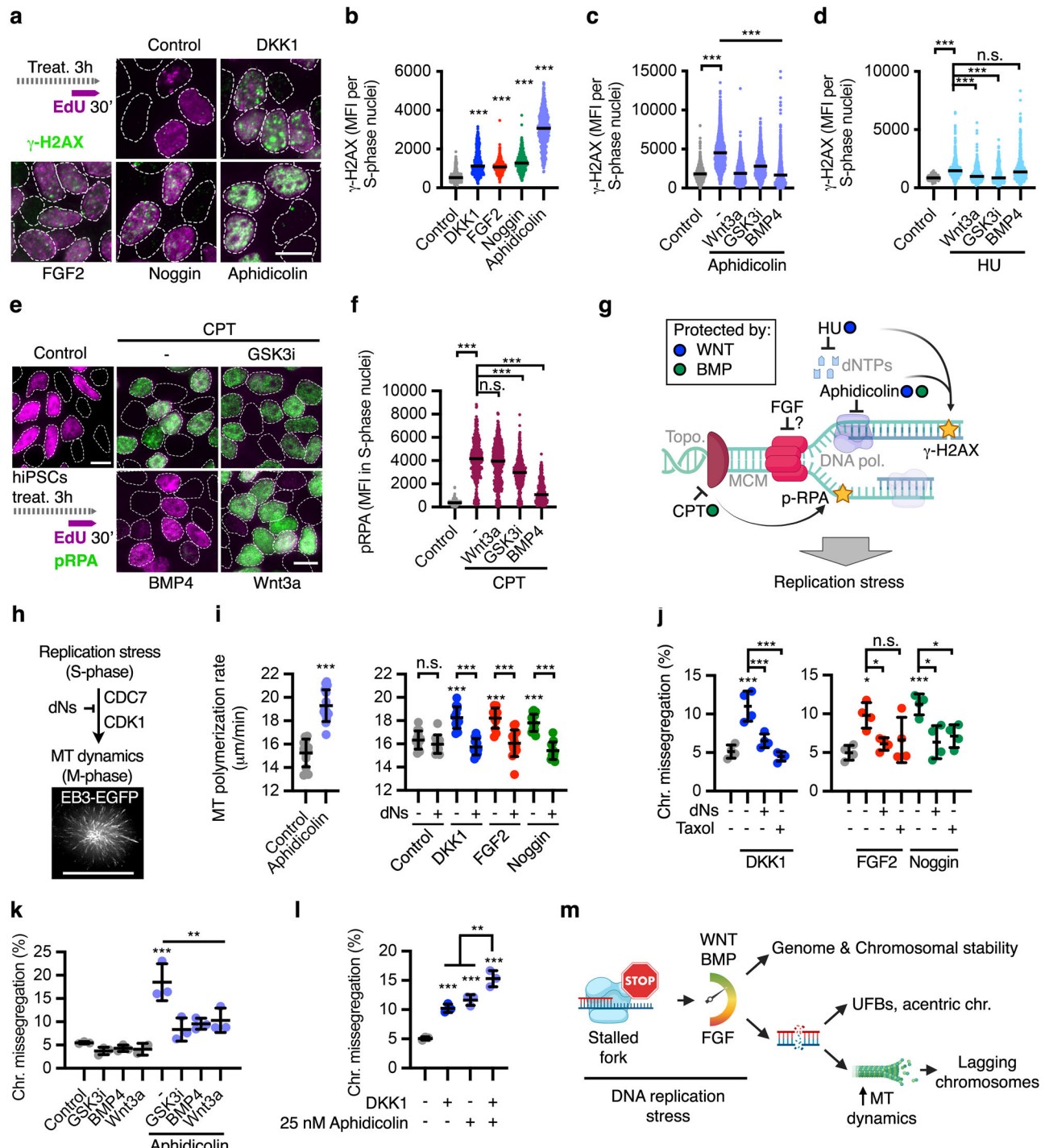

stalled replication forks, and a downstream role of WNT signalling in resolving DNA replication stress (Fig. 4g). Future molecular studies are required to disentangle the many targets and functions of either pathway across the different steps associated with replicative stress.

## WNT, BMP, and FGF signalling regulate chromosome segregation in mitosis through the modulation of DNA replication stress in the previous S-phase

Unresolved severe DNA replication stress leads to cell cycle arrest, and cells that override the DNA damage checkpoint often present specific mitotic phenotypes, including acentric chromosomes and ultra-fine bridges during anaphase[14,15]. Mild replicative stress can escape the checkpoint control[14,17], and triggers CDC7-dependent elevated

microtubule dynamics in mitosis and subsequent whole-chromosome missegregation[17]. DKK1, FGF2, and Noggin did not induce cell cycle arrest in hiPSCs (Supplementary Fig. 2n), but instead promoted similar effects during DNA replication as 50 nM aphidicolin (Fig. 4b), which triggers mild/physiological levels of replication stress[17]. Furthermore, loss of WNT signalling upregulates polymerisation of microtubule plus-ends in somatic cells[28]. As such, we decided to examine whether WNT, BMP, and FGF signalling roles in chromosome segregation are triggered by mild DNA replication stress and subsequent elevated microtubule dynamics.

Live cell imaging analyses of EB3-eGFP in hiPSCs revealed that DKK1, Noggin, and FGF2 increased microtubule polymerisation dynamics during mitosis, similarly to inducing replicative stress with

**Fig. 4 | WNT, BMP, and FGF affect DNA damage in S-phase and chromosome segregation fidelity in mitosis through their roles in DNA replication.**
**a–f** Accumulation of γ-H2AX foci (**a–d**) and phospho-RPA (**e, f**) during S-phase in hiPSCs treated for 3 h as indicated. HU, hydroxyurea; CPT, camptothecin. Data are median fluorescence intensity (MFI) of EdU⁺ nuclei of a representative experiment, performed at least three independent times ($n = 3$ biological replicates) with >250 cells per condition in each experiment. MFI at EdU⁺ nuclei of: control ($n = 616$ cells), DKK1 ($n = 571$ cells), FGF2 ($n = 526$ cells), Noggin ($n = 478$), and Aphidicolin ($n = 460$) in (**b**), control ($n = 1824$ cells), Aphidicolin ($n = 2394$ cells), Aphidicolin + WNT3A ($n = 1523$ cells), Aphidicolin + GSK3i ($n = 1549$ cells), and Aphidicolin + BMP4 ($n = 1743$ cells) in (**c**), control ($n = 250$ cells), HU ($n = 355$ cells), HU + WNT3A ($n = 529$ cells), HU + GSK3i ($n = 373$ cells), and HU + BMP4 ($n = 448$ cells) in (**d**), control ($n = 414$ cells), CPT ($n = 417$ cells), CPT + WNT3A ($n = 689$ cells), CPT + GSK3i ($n = 582$ cells), and HU + BMP4 ($n = 657$ cells) in (**f**) were analysed. **g** Schematic summarising the signalling functions of WNT, BMP, and FGF upon DNA replication stress. *P*-values from one-way ANOVA analyses with multiple comparisons with Dunnet corrections from the indicated groups, \*\*\**P* < 0.001, in (**d**) n.s. = 0.993, in (**f**) n.s. = 0.217. **h, i** Mitotic microtubule plus-end assembly rates measured by EB3-GFP tracking during prometaphase in hiPSCs. Cells were treated as indicated (16 h of total treatment) and arrested in mitosis using dimethylenastrone for 2 h prior to

imaging. Experiments were replicated at least two times and the data shown corresponds to a representative experiment. Data are mean ± s.d. of average growth rates of 20 microtubules/per cell, where microtubule polymerization rate was measured for at least four consecutive time points (every 2 s). Each dot in the figure corresponds to one cell; control = 12 cells, Aphidicolin = 15 cells (left panel), control = 14 cells, control + dNs = 11 cells, DKK1 = 13 cells, DKK1 + dNs = 12 cells, FGF2 = 13 cells, FGF2 + dNs = 14 cells, Noggin = 13 cells, Noggin + dNs = 12 cells. *P*-values from one-way ANOVA analyses with multiple comparisons with Dunnet corrections are indicated as \*\*\**P* < 0.001, or not significant (n.s. = 0.974), for single cells of a representative experiment. **j–l** Chromosome segregation analyses in hiPSCs treated as indicated for 16 h. Data are mean ± s.d. of $n = 4$ biological replicates (**j**), $n = 3$ biological replicates (**k**), and $n = 3$ biological replicates (**l**) with >100 mitotic cells per condition and per replicate. *P*-values from one-way ANOVA analyses with multiple comparisons with Dunnet corrections are indicated as \**P* < 0.05, \*\**P* < 0.01, \*\*\**P* < 0.001, or in (**j**) not significant (n.s. = 0.07) for independent experiments. **m** Proposed model for cell signalling regulation of DNA replication and chromosome segregation fidelity. Scale bars = 10 μm. Source data for all experiments are provided as a Source data file. **g, m** Created with BioRender.com released under a Creative Commons Attribution-NonCommercial-NoDerivs 4.0 International license[139].

aphidicolin (Fig. 4h, i). To determine whether cell signalling effects in the mitotic spindle were indeed caused by their roles in DNA replication, we co-treated cells with nucleosides (dNs), which alleviate DNA replication stress and prevent DNA replication-associated damage[66,67], including upon consumption of endogenous nucleotide pools after the firing of too many origins[17,68,69]. Treatment of hiPSCs with nucleosides indeed rescued the upregulation of microtubule polymerisation by DKK1, Noggin, and FGF2 (Fig. 4i), supporting a mechanistic link between WNT, BMP, and FGF roles connecting S- and M-phase.

Critically, attenuation of replicative stress with nucleosides was sufficient to rescue chromosome missegregation induced by DKK1, FGF2, or Noggin in hiPSCs (Fig. 4j), as shown before for other chemical and genetic perturbations inducing mild replication stress-associated chromosome segregation errors[14,17]. In agreement with a mechanistic link between replication stress, elevated microtubule dynamics and chromosome missegregation, stabilisation of microtubule dynamics with 10 nM taxol also rescued cell signalling induced chromosome missegregation (Fig. 4j). Furthermore, triggering mild DNA replication stress with 50 nM aphidicolin for 16 h increased from 5% to 18% the proportion of mitotic hiPSCs displaying lagging chromosomes, which could be rescued by exogenous activation of WNT or BMP signalling (Fig. 4k). In addition, WNT inhibition by DKK1 cooperated with 25 nM aphidicolin to induce chromosome missegregation (Fig. 4l).

In light of these results, we decided to re-examine our chromosome segregation experiments for the presence of CENPC-negative missegregated chromosomes, which is typical for acentric chromosomes[14,70]. DKK1, Noggin or FGF2 treatment increased the proportion of anaphases with CENPC negative chromosomes from 0.7% to 1.25–2.5% (Supplementary Fig. 6a), representing ~15% of the total missegregated chromosomes across experiments (Supplementary Fig. 6a). In agreement with a more severe effect during DNA replication (Fig. 3d), treatment with 50 nM Aphidicolin resulted in 5% of anaphases displaying acentric chromosomes (Supplementary Fig. 6b; ~35% of the total missegregated chromosomes). Furthermore, follow-up analyses revealed that 10–15% of hiPSCs treated with DKK1, Noggin or FGF2, presented ultra-fine bridges at late anaphase (Supplementary fig. 6c, d), similarly to treating cells with 50 nM aphidicolin. These results further support the role of WNT, BMP, and FGF in regulating DNA replication stress, reveal a richness of replicative stress-derived phenotypes in hiPSCs, and suggest that future genomic analyses are required to evaluate the contribution of cell signalling and different replicative stress severity to structural chromosomal defects and copy-number-variations in pluripotent stem cells.

Taken together, our data indicate (i) that FGF signalling induces replicative stress-dependent chromosome missegregation in pluripotent stem cells, and (ii) that autocrine BMP and WNT signalling promote chromosome segregation fidelity in pluripotent stem cells by preventing excessive origin firing and DNA damage downstream of stalled forks during S-phase from different sources of replicative stress (Fig. 4m). In the case of WNT signalling, our findings further highlight a complex programme driven by this pathway to promote faithful G1, S- and M-phase progression[28,29,42,71,72]. The exact molecular mechanisms underlying WNT, BMP, and FGF signalling control of DNA replication and how other signalling pathways, including TGF-β (Fig. 1b), epistatically impact chromosome segregation fidelity, remain to be characterised.

## The limited function of morphogens in chromosomal stability after germ layer specification

During human embryonic development, WNT3A and NODAL direct epiblast cells into a transient posterior structure named primitive streak prior to commitment to the mesoderm and the definitive endoderm, a process that can be mimicked in vitro using WNT3A and Activin A (Supplementary Fig. 7a, b)[32,34]. In contrast to pluripotent stem cells, inhibition of WNT signalling did not result in chromosome missegregation in human primitive streak-like cells (Fig. 5a and Supplementary Fig. 7a), and had a limited effect in the definitive endoderm-like cells (Fig. 5a and Supplementary Fig. 7a, d, e). Upon FGF induction, WNT and BMP signalling display antagonistic functions in the primitive streak differentiation into paraxial and lateral mesoderm (Supplementary Fig. 7a, f)[32]. Perturbation of WNT, BMP or FGF signalling during the last 16 h of the specification of these lineages did not impact chromosome segregation fidelity (Fig. 5a and Supplementary Fig. 7g). To get insight into signalling roles in the ectoderm lineage, we turned into neuroectoderm specification, which is the first step in the development of the nervous system[20]. We performed Noggin-directed differentiation of hiPSCs into neuroectoderm-like cells (Supplementary Fig. 7a, h)[33,73]. Neuroectoderm cells displayed slightly higher levels of chromosome missegregation (6%) compared to primitive streak, endoderm, or mesoderm (~3%) (Supplementary Fig. 7g, i). Although treatment with BMP4 slightly reduced chromosome missegregation in neuroectoderm-like cells, Noggin withdrawal or DKK1 treatment had no effect (Fig. 5a and Supplementary Fig. 7i). We also generated neural crest-like cells and early neural stem cells (NSCs) by a pairwise decision through activation or inhibition of WNT activity, respectively (Supplementary Fig. 7a, j)[74]. Early neural progenitors displayed

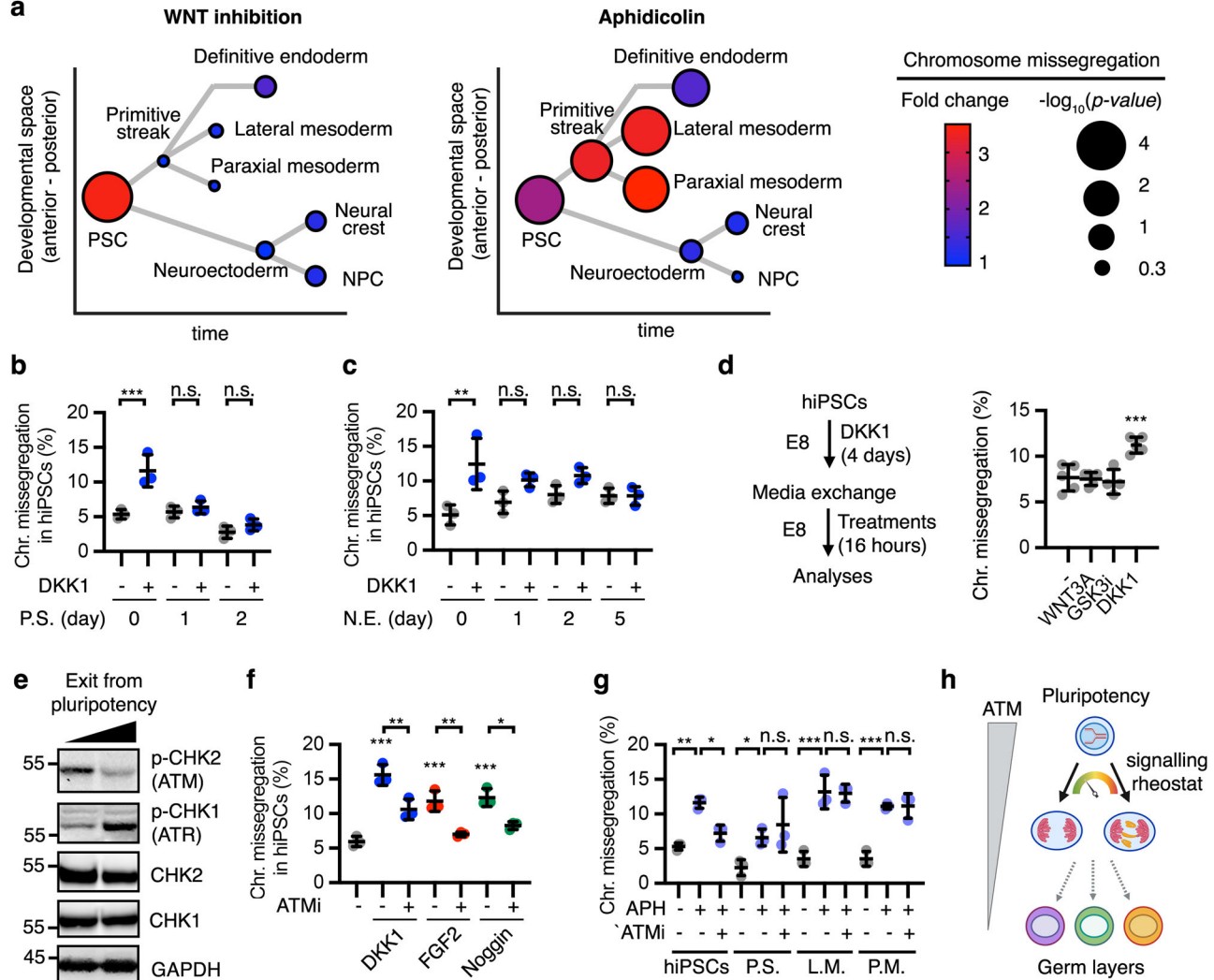

**Fig. 5 | WNT, BMP, and FGF do not impact chromosomal stability after human early lineage specification. a** Summary of chromosome segregation analyses across different embryonic lineages obtained upon differentiation of hiPSCs as further described in Supplementary Fig. 5a. Data show fold changes in chromosome missegregation rates upon DKK1 treatment (WNT inhibition) or aphidicolin versus control untreated conditions, and *P*-values from two-tailed t-tests (treated vs control) of 3–7 biological replicates with >100 anaphases analysed in each condition, also shown in Supplementary Fig. 5. Data are displayed in a lineage tree depicting relative developmental position (A–P) in time of each lineage. NPC, neural progenitor cell. **b–d** Chromosome segregation analyses in hiPSCs undergoing differentiation towards primitive streak (P.S.) (**b**) or neuroectoderm (**c**) as indicated in Supplementary Fig. 6a, and treated for 16 h as indicated. Data are mean ± s.d. of **b** *n* = 3 biological replicates, **c** *n* = 3 biological replicates and **d** *n* = 4-5 biological replicates with >than 100 anaphases quantified per condition and per replicate). Note that GSK3i/WNT3A were exchanged to DKK1 during the differentiation towards primitive streak to ensure proper WNT inhibition. In **d** hiPSCs were cultured with DKK1 in E8 media for 4 days, and either kept for another 16 h with DKK1 or exchanged towards control (−), WNT3A or GSK3i in E8 media. *P*-values

from two-way with Tukey corrections (**b**, **c**) or one-way with Dunnet corrections (**d**) ANOVA analyses with multiple comparisons are indicated as \*\**P* < 0.01, \*\*\**P* < 0.001, or n.s. (*P* > 0.05, not significant). **e** Representative Western blot analyses (of *n* = 2 biological replicates) of hiPSCs undergoing differentiation towards primitive streak (Left = day 0, right = day 2 of differentiation as shown in Supplementary Fig.7a). The molecular weight markers are indicated in kDa. **f, g** Chromosome segregation analyses in hiPSCs, primitive streak-like (P.S.), lateral mesoderm (L.M.) and paraxial mesoderm (P.M.) like cells treated as indicated for 16 h. Data are mean ± s.d. of **f** *n* = 3 biological replicates and **g** *n* = 3 biological replicates with >100 anaphases quantified per condition and per replicate) APH, 50 nM aphidicolin. ATMi, 3 µM AZD0156. *P*-values from one-way ANOVA analyses with multiple comparisons and Dunnet corrections are indicated as \**P* < 0.05, \*\**P* < 0.01, \*\*\**P* < 0.001, or n.s. (*P* > 0.05, not significant). **h** Schematic of the functional interaction between signalling roles in chromosome segregation and cell fate. Source data for all experiments are provided as a Source data file. **h** Created with BioRender.com released under a Creative Commons Attribution-NonCommercial-NoDerivs 4.0 International license[139].

higher basal chromosome missegregation than neural crest-like cells (Supplementary Fig. 7k), but perturbation of WNT signalling had no significant impact in chromosome segregation in either lineage (Fig. 5a and Supplementary Fig. 7k). These results indicate that WNT, BMP, and FGF roles in chromosome segregation fidelity are largely impaired after exiting from pluripotency, regardless of their critical signalling functions in cell fate determination in the three human germs layers[32,33,73] (Supplementary Fig. 7a). In contrast to FGF2, DKK1,

or Noggin—and despite the presence of WNT3A (Supplementary Fig. 7a)—aphidicolin treatment-induced DNA damage and chromosome missegregation in primitive streak, endoderm, and mesoderm-like cells (Fig. 5a and Supplementary Fig. 7l, m). This indicates that specified cells display a functional DNA replication stress response but it is uncoupled of extracellular signalling activity. Aphidicolin did not induce high levels of γ-H2AX foci or chromosome missegregation in neuroectoderm, early neural progenitor cells, or neural crest cells

(Fig. 5a), possibly due to a higher basal replicative stress in these cells (Supplementary Fig. 7l, m).

To get additional insights on the timing for cell signalling withdrawal from chromosome segregation fidelity during early lineage specification, we followed the differentiation of hiPSCs into primitive streak or neuroectoderm and analysed the effect of WNT inhibition by DKK1 during the early steps of the specification process. Differentiation into primitive streak resulted in lower basal chromosome missegregation rates (Fig. 5b), while differentiation into neuroectoderm steadily increased them (Fig. 5c). However, WNT inhibition had no significant effect in chromosome segregation fidelity once cells committed to either lineage for one day (Fig. 5b, c). On the other, hiPSCs continuously cultured with DKK1 for 5 days under pluripotency conditions (E8 media, Supplementary Fig. 8a) partially retained their response to WNT activation or inhibition towards chromosome segregation fidelity (Fig. 5d). In addition, hiPSCs treated for 1 or 5 days with DKK1 in E8 media displayed similar levels of aneuploidy (Supplementary Fig. 8b, c). These results suggest that lineage specification and not the duration of the signalling perturbations underlies the loss of function of cell signalling in chromosome segregation upon exit from pluripotency.

Next, we interrogated how different in vitro pluripotency stages respond to signalling-dependent DNA replication stress. To get comparative analyses between the impact of WNT signalling in naive, formative and primed pluripotency, we turned into mESCs (naive), and specified them into formative-like stem cells (mFSCs, formative, Supplementary Fig. 8d, e), Epiblast-like cells (mEpiLCs, primed, Supplementary Fig. 8f, g), or differentiated them into the 3 germ layers by removing LIF (differentiated). Treatment with DKK1 induced chromosome missegregation in mESCs, mFSCs, and mEpiLCs—but not in differentiated cells—(Supplementary Fig. 8e). Interestingly, WNT regulation of chromosome segregation fidelity peaked in primed mEpiLCs (corresponding to hiPSCs in human and the pre-gastrulating epiblast in the embryo), which have been recently shown to display a higher tolerance to replicative stress compared to naive mESCs[75].

ATM signalling regulates the DNA replication stress response in pluripotent stem cells[46], in addition to the most common ATR response. We observed that exit from pluripotency towards primitive streak in human cells is accompanied by reduced phosphorylation of the ATM target CHK2, and increased phosphorylation of the ATR target CHK1 (Fig. 5e). We hypothesised that a switch between ATM and ATR signalling could mediate the cell fate-dependent roles of morphogens in chromosomal stability. Indeed, inhibition of ATM signalling using AZD0156 (ATMi) rescued chromosome missegregation induced by DKK1, Noggin, and FGF2, as well as aphidicolin, in hiPSCs (Fig. 5f, g). However, ATMi did not rescue chromosome missegregation induced by aphidicolin in human primitive streak, lateral mesoderm or paraxial mesoderm (Fig. 5g). Taken together, these results indicate that the effects of WNT, BMP, and FGF signalling on chromosome segregation fidelity decline after lineage specification in the three human germ layers following the withdrawal of ATM signalling as first responder to DNA replication stress (Fig. 5h).

## A tug-of-war between WNT and FGF impacts chromosomal stability during neurogenesis

Beyond peri-implantation and gastrulating embryos, only the neocortex displays high levels of genomic and chromosomal mosaicism later during embryonic development[5,12,76–79]. To study the roles of cell signalling in chromosome segregation fidelity across more differentiated lineages, we generated cardiomyocyte-like (mesoderm), hepatocyte-like (endoderm), and neuronal progenitor cells (ectoderm) from hiPSCs (Supplementary Fig. 9a–l). Despite their response to chemically-induced replicative stress, neither cardiomyocyte nor hepatocyte-like cells at different stages of differentiation showed increased chromosome missegregation upon WNT inhibition

(Supplementary Fig. 9c, g), in sharp contrast with neural progenitors, as further explained below.

To model human cortical neurogenesis in vitro, we differentiated hiPSCs into neural progenitors (hiNPCs) during a 16-day course (Fig. 6a, b and Supplementary Fig. 9h)[80,81]. Neural progenitor proliferation and neurogenesis are promoted by paracrine FGF and autocrine WNT signalling, among other pathways[82–86]. In vitro generated hiNPCs expressed primarily *WNT8B*, in addition to *WNT1/3A/10B* (Supplementary Fig. 9i). Inhibition of autocrine WNT signalling activity by DKK1, as well as activation of FGF signalling with FGF2, strongly increased chromosome missegregation in Nestin⁺ hiNPCs during neurogenesis (day 16), but had no effect during hiNPC expansion (day 10) (Fig. 6c, d and Supplementary Fig. 9j–l).

DNA combing and γ-H2AX foci analyses confirmed that both FGF activation and WNT inhibition promoted DNA replication stress and DNA damage in hiNPCs at the onset of neurogenesis (Fig. 6e–g). Alleviation of replicative stress in hiNPCs using nucleosides rescued chromosome missegregation induced by DKK1 and FGF2 (Fig. 6h, i). Of note, hiNPCs displayed higher replication speed compared to hiPSCs, possibly due to the shortening of S-phase during their commitment to cortical neurons[87]. Furthermore, exogenous activation of the WNT pathway by GSK3i or WNT3A during in vitro neurogenesis rescued FGF2- and aphidicolin-induced chromosome missegregation (Fig. 6i, j).

Next, we used mouse embryonic brains and primary neural progenitors (NPCs) to determine whether our findings translate to in vivo and ex vivo conditions. In the developing mouse neocortex, NPCs expand at the ventricular zone before E13.5, followed by direct and indirect neurogenesis during E13.5–E15.5[82,88]. Expression of WNT ligands, as well as phosphorylation of the WNT receptor LRP6, remained mostly constant in the developing ventricular zone during E12.5–E16.5 (Fig. 7a, b and Supplementary Fig. 10a, b). *Fgf2* and *Fgfr1* are expressed in the ventricular zone of rodents before the onset of neurogenesis[89]. Accordingly, we found that FGFR1 receptor activation (via phosphorylation) peaked in the apical progenitors during E14.5 (Fig. 7a, c). We hypothesised that differential FGF and WNT activities during neurogenesis might contribute to the observed high levels of mosaicism in the developing brain[5,12,76–79].

To study the functional interaction between WNT and FGF signalling, we isolated NPCs from E12.5 and E14.5 mouse embryonic cortices, which mainly expressed *Wnt7a/b* (Supplementary Fig. 10a, b). Inhibition of endogenous WNT signalling by DKK1 or activation of FGF signalling using FGF2 induced chromosome missegregation in E14.5-derived NPCs, but not in E12.5-derived NPCs (Fig. 7d). These results echoed our finding in expanding vs. differentiating human iNPCs (Fig. 6c, d), further supporting a selective conserved role of WNT and FGF signalling in controlling chromosome segregation fidelity during cortical neurogenesis. WNT and FGF antagonising roles in chromosome segregation fidelity were also tracked back to S-phase in mouse primary NPCs: first, alleviation of DNA replication stress using nucleosides rescued chromosome segregation defects induced by DKK1 and FGF2 (Fig. 7d). Second, both FGF2 and DKK1 increased the number of γ-H2AX foci during S-phase (Fig. 7e, f). As in the case of hiNPCs (Fig. 6i), exogenous activation of WNT signalling with WNT3A or GSK3i rescued FGF2-induced chromosome missegregation in primary NPCs isolated from E14.5 mouse embryos (Fig. 7g).

To characterise WNT signalling roles in chromosome segregation directly in vivo, we performed in utero intra-ventricular injections of recombinant DKK1 in E13.5 embryos and analysed their ventricular zones at E14.5 (Fig. 7h) during the peak of FGF signalling activity. We observed high levels of chromosome missegregation (9%) in the control embryos (Fig. 7h–k, PBS injected), similar to previous estimates[11,12,25,76,90] and to mNPCs cultured ex vivo with FGF2 (Fig. 7d, g). Critically, in vivo inhibition of WNT signalling further induced chromosome missegregation in 19% of dividing NPCs during embryonic neurogenesis (Fig. 7j, k).

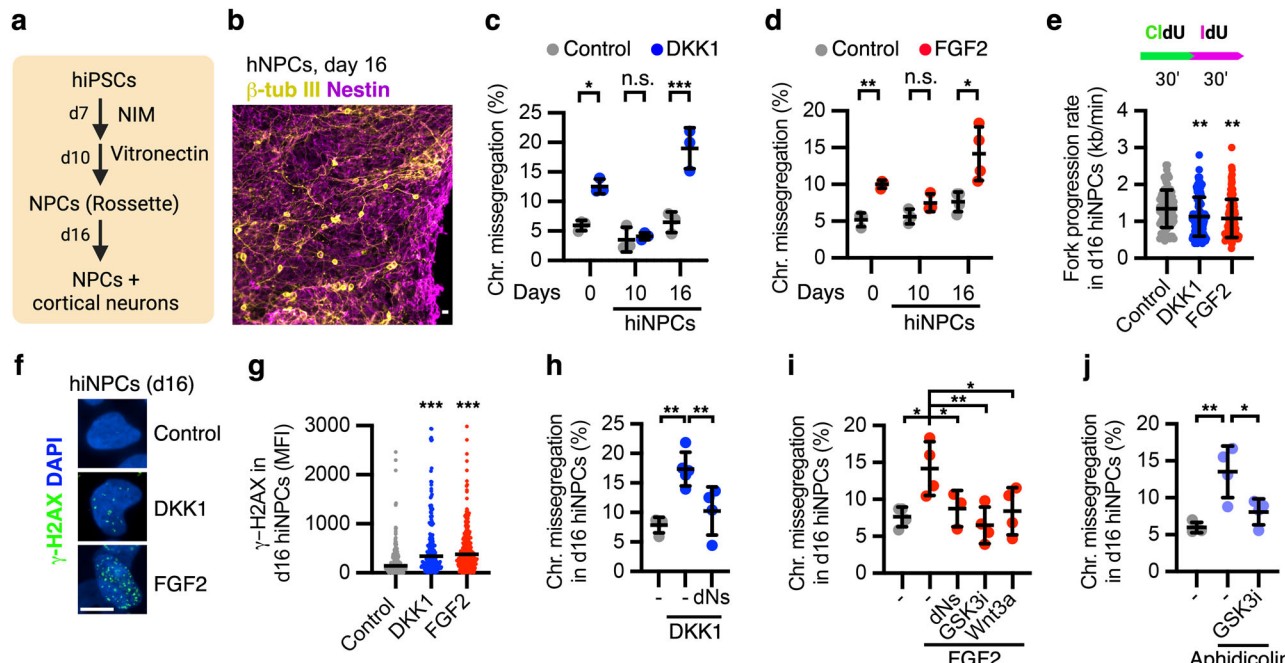

**Fig. 6 | WNT and FGF display antagonistic roles in genome stability during human in vitro neurogenesis. a**, **b** In vitro specification of hiPSCs into cortical neural progenitors and neurons. NIM neural induction media, NPC neural progenitor cell, B-tub III beta tubulin III. **c**, **d** Chromosome segregation analyses in hiPSCs, expanding hiNPCs (10 days) and differentiating hiNPCs (16 days) as indicated in (**a**), and treated with DKK1 or FGF2 16 h before harvesting. Data are mean ± s.d. of (**c**) $n = 3$ biological replicates and (**d**) $n = 3$-4 biological replicates with >50–100 anaphases quantified per condition in each replicate). *P*-values from two-way ANOVA analyses with multiple comparisons and Tukey corrections are indicated as *$P < 0.05$, **$P < 0.01$, ***$P < 0.001$, or n.s. ($P > 0.05$, not significant). **e** DNA combing experiments in differentiating hiNPCs treated as indicated for 3 h, and labelled with consecutive pulses of CldU and IdU. Data are mean ± s.d. of the fork speed measurement in Control=83 forks, DKK1 = 112 forks and FGF2 = 119 forks, corresponding to a representative experiment after biological replication ($n = 3$). *P*-

values from one-way ANOVA analyses with multiple comparisons with Dunnet corrections are indicated, from left to right as **$P = 0.0052$ and **$P = 0.0011$. **f**, **g** Accumulation of γ-H2AX foci in differentiating hiNPCs treated as indicated. A representative experiment from three independent experiments is shown. Data are median of the fluorescence intensity (MFI) in control 267 nuclei, DKK1 = 206 nuclei and FGF2 = 247 nuclei. *P*-values from one-way ANOVA analyses with multiple comparisons with Dunnet corrections are indicated, from left to right as **$P = 0.0017$ and **$P = 0.002$. **h**–**j** Chromosome segregation analyses in differentiating hiNPCs (16 days) treated as indicated for 16 h. Data are mean ± s.d. of $n = 3$-4 biological replicates (**h**–**j**) with 50–100 mitotic cells analysed per condition in each replicate. *P*-values from one-way ANOVA analyses with multiple comparisons with Dunnet corrections are indicated as *$P < 0.05$, **$P < 0.01$, ***$P < 0.001$. Scale bars = 10 μm. Source data for all experiments are provided as a Source data file.

Finally, we enquired whether signalling-induced chromosome missegregation in NPCs impacts their viability and fate. Tracking analyses of live cell imaging showed that chromosome missegregation in mouse E14.5-derived NPCs increased 2-fold the rate of both cell death (14% after normal mitosis vs 30% after chromosome missegregation) and of additional segregation errors (5% after normal mitosis vs 12% after chromosome missegregation) in the daughter cells, but surprisingly did not substantially impact the ratio of dividing/non-dividing daughter cells (Supplementary Fig. 10d–g). These results indicate that chromosome missegregation is largely tolerated in primary NPCs committed to neurogenesis and could contribute to chromosomal mosaicism in the brain[91].

In summary, our findings show that a tug-of-war between the WNT and FGF pathways controls chromosome segregation fidelity during mouse and human neurogenesis (Figs. 7k, 8). In particular, our ex vivo and in vivo analyses suggest that paracrine FGF signalling activation, as it occurs during E14.5, is sufficient to induce chromosome missegregation in mNPCs, which cannot be fully countered by basal, autocrine WNT activity. It remains to be investigated whether other spatiotemporal signalling gradients regulating neurogenesis, neural specification, and brain patterning—including SHH, EGF, BMP, and TGF-β[20,92,93]—also modulate genome stability in neural progenitors.

## Discussion

Morphogens, patterning signals and growth factors direct cells to build and maintain tissues, organs and organisms by interpreting

genotypes into gene-regulatory networks. In this study, we show that these developmental signals not only *read* genetic blueprints, but also play a role in their maintenance. We reveal a dichotomy between signals inducing anteriorisation or posteriorisation during gastrulation in inducing or protecting pluripotent stem cells from chromosome missegregation, respectively. We identified that these patterning signals regulate DNA replication dynamics and integrity in S-phase thereby affecting microtubule dynamics and chromosome segregation fidelity in the subsequent mitosis. We show that WNT and BMP signalling sit at the helm of this regulatory cascade by protecting cells from different sources of replicative stress. Specifically, we found that BMP signalling protects ssDNA during DNA replication stress, and that WNT signalling functions downstream alleviating replicative stress and preventing DSBs induced by stalled forks and depletion of nucleotides (Fig. 8). In light of our results, we think it would be important to revisit different hiPSC/hESC culture conditions to reduce the risk of aneuploidy derived of long-term culture.

Our results indicate that WNT control of chromosome segregation fidelity peaks during primed pluripotency. By performing in vitro lineage specification experiments with mouse and human pluripotent stem cells, we showed that cells rely on extracellular signals for faithful chromosome segregation in naive, formative and primed pluripotency and during neurogenesis, but not in other early specified human lineages (e.g. Primitive streak, mesoderm, and endoderm) or subsequent differentiated lineages such as hepatocyte- and cardiomyocyte-like cells.

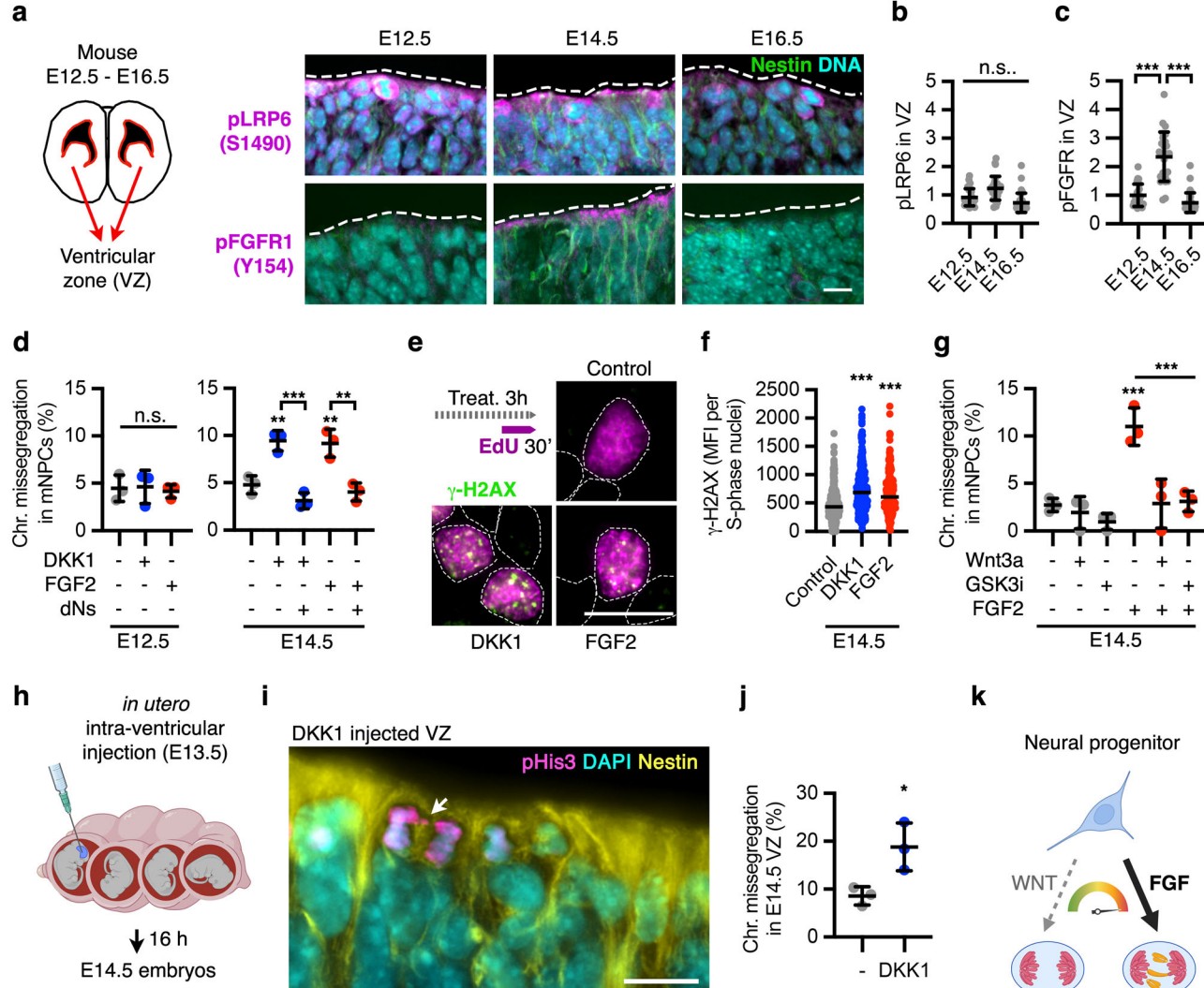

**Fig. 7 | A tug-of-war between WNT and FGF regulates chromosome segregation fidelity in mouse neural progenitors. a–c** Immunofluorescence analyses of the activated WNT (LRP6) and FGF (FGFR1) receptors in the ventricular zone of E12.5-16.5 embryonic mouse brains. Quantification of phospho-LRP6 (S1490) (**b**) and phospho-FGFR1 (Y154) (**c**) in NPCs from the ventricular zone of mouse embryos is shown for three developmental stages of cortical neurogenesis. Data are mean ± s.d. of the relative fluorescence intensity normalised to Nestin in apical neural progenitors of $n > 5$ brain cryosections from $n = 3$ embryos of each condition. *P*-values from one-way ANOVA analyses with multiple comparisons with Dunnet corrections are indicated as ***$P < 0.001$, or n.s. ($P > 0.05$, not significant). **d** Chromosome segregation analyses in ex vivo cultured NPCs from E12.5 and E14.5 mouse embryos, treated as indicated for 16 h. Data are mean ± s.d. of chromosome missegregation rates from $n = 3$ biological replicates with >50 anaphases per condition in each replicate. In each experiment, mNPCs dissociated from $n > 3$ mouse embryo brains were pooled together for seeding. *P*-values from two-way ANOVA analyses with multiple comparisons and Tukey corrections are indicated as **$P < 0.01$, ***$P < 0.001$, or n.s. ($P > 0.05$, not significant). **e**, **f** Accumulation of γ-H2AX foci in S-phase of E14.5-derived NPCs treated for 3 h as indicated. Data are the

MFI in $n > 200$ EdU+ nuclei from a representative experiment out of three biological replicates. *P*-values from one-way ANOVA analyses with multiple comparisons with Dunnet corrections are indicated as ***$P < 0.001$. **g** Chromosome segregation analyses in ex vivo cultured NPCs from E14.5 mouse embryos, treated as indicated for 16 h. Data are chromosome missegregation rates plotted as mean ± s.d. from $n = 3$ biological replicates with 50–100 anaphases analysed per condition and per replicate. *P*-values from one-way ANOVA analyses with multiple comparisons with Dunnet corrections are indicated as ***$P < 0.001$. **h** Schematics of in utero ventricular injections of PBS (control) or recombinant DKK1 in E13.5 mouse embryos, later sacrificed at stage E14.5. **i**, **j** Chromosome segregation analyses in the ventricular zone of PBS (Control) or DKK1 injected mouse E14.5 embryos. Data are mean ± s.d. of three injected embryos per condition (>10 cryosections per embryo). *P*-values from a two-tailed *t*-test are indicated as *$P = 0.029$. **k** A tug-of-war between WNT and FGF controls chromosome segregation fidelity in NPCs, and might underlie the high levels of chromosome missegregation occurring during neurogenesis. Scale bars = 10 μm. Source data for all experiments are provided as a Source data file. **h**, **k** Created with BioRender.com released under a Creative Commons Attribution-NonCommercial-NoDerivs 4.0 International license[139].

These results highlight that the exit of pluripotency is a boundary for cell signal control of chromosome segregation fidelity. Using mouse and human neural progenitors, we show that FGF and WNT signalling display opposite roles in chromosome segregation fidelity during cortical neurogenesis (Fig. 8). In particular, our results suggest that DNA replication stress triggered by elevated FGF2 activity during the onset of neurogenesis could contribute to the notoriously high levels of chromosomal mosaicism of the developing brain.

Together with previous studies[27–29,94], our work highlights a critical role of β-catenin-independent WNT/GSK3 signalling (Supplementary Fig. 2c), possibly through the post-translational cascade WNT/STOP[41], in reducing DNA replication stress, modulating microtubule polymerisation dynamics and promoting chromosome segregation fidelity. Intriguingly, alterations on key Wnt components that increase Wnt activity, including GSK3, have also been shown to induce chromosome segregation defects in other contexts[95–102]. Furthermore, loss

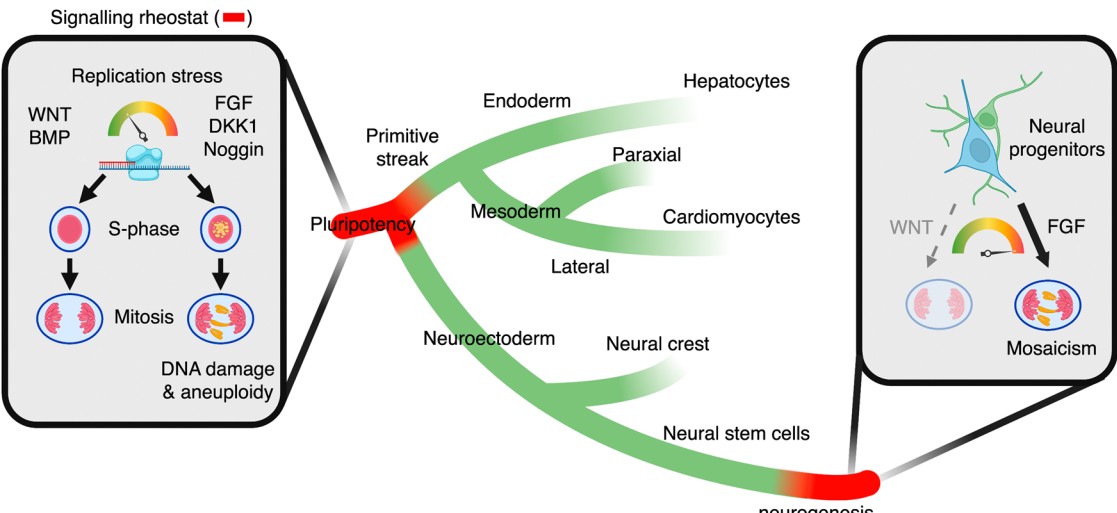

**Fig. 8 | Model of proposed roles of patterning signals in chromosome segregation fidelity during human lineage specification.** In pluripotent stem cells, the WNT, BMP, and FGF pathways form part of an ATM-dependent signalling rheostat that modulates DNA replication stress during S-phase, which in turn regulates microtubule dynamics and chromosome segregation fidelity in the subsequent mitosis. WNT signalling sits at the helm of this regulatory network by protecting pluripotent stem cells from chromosome missegregation upon different sources of DNA replication stress, including by other patterning signals. The capacity of investigated extracellular signals to influence chromosome segregation fidelity is largely lost after exit from pluripotency and specification into the three germ layers following the withdrawal of ATM signalling as a first responder during DNA replication stress, but remerges during neurogenesis. In particular, we find that FGF signalling induces high levels of chromosome missegregation of neural progenitors committed to neurogenesis. Figure 8 was created with BioRender.com and released under a Creative Commons Attribution-NonCommercial-NoDerivs 4.0 International license[139].

of β-catenin can increase astral microtubule dynamics[103]. Although many of these phenotypes are caused by WNT signalling independent mechanisms[29], it remains to be characterised whether both excess or depletion of WNT activity outside of particular parameters might risk genome maintenance, similarly as it occurs with the up- and down-regulation of replication fork speed[104].

A standing question is why patterning signals would regulate chromosome segregation fidelity during early lineage specification and neurogenesis, but not in the other analysed lineages. Pluripotent stem cells and neural progenitors share specific features compared to the other embryonic lineages studied here: First, both are bona fide stem cells with the capacity to self-renew and differentiate in various cell types. Second, our data links the ATM response to the capacity of WNT signalling to regulate chromosome segregation fidelity (Fig. 5f, g). While ATM is not essential for embryonic development[105,106], loss of ATM sensitises embryonic stem cells[46] and gastrulating embryos[107], as well as neural progenitors in the developing brain[108], to genotoxic stress. Together, these results suggest that ATM activity is tightly linked to pluripotent and NSCs although further studies are required to determine the underlying reasons. Third, mutations in genes related to replication stress in both humans and mice often result in phenotypes at the peri-implantation stage[109] and in the developing brain[110,111]. However, it is important to note that pluripotent stem cells and neural progenitors strikingly differ in their cell cycle length and origin licensing strategies[112], among other features. Taken together this and previous studies[5,7,8,10,11] suggest that early lineage specification and neurogenesis represent developmental bottlenecks where chromosomal stability is at risk. However, further analyses in other stem cell models should provide additional molecular and cell identity insights allowing to unravel how and why patterning signals impact chromosome segregation fidelity in a lineage-specific manner.

Predictions from our study are as follows: **1)** Morphogens and patterning signals should interlink DNA replication dynamics, DNA damage, and chromosomal stability with cell fate. Indeed, mutational and aneuploidy rates largely overlap during human development, peaking before gastrulation and during neurogenesis[5,9,11]. Furthermore, DNA replication speed, ATM, γ-H2AX, and aneuploidy have recently emerged as key regulators of cell fate[24,113–115], supporting a bidirectional relationship between them. Hence, it would be important to explore whether signalling roles in DNA replication provide an additional layer of control for lineage specification. **2)** Spatio-temporal signalling gradients driving the anterior-posterior axis in the gastrulating embryo should also generate a mosaicism gradient overlapping with the former. Human blastocysts do indeed show high levels of aneuploidy[9] and spatial chromosomal mosaicism[116], but it remains to be determined (i) whether DKK1, Chordin, Cerberus, Noggin, and/or LEFTY-A also induce higher levels of CIN in the anterior part of the embryo during gastrulation, (ii) if NODAL, BMP4 and WNT3A[34] protect the organiser from replication stress and chromosome missegregation, and (iii) whether these gradients trigger BMP4- and lineage-specific depletion of aneuploid cells[7,117]. **3)** Genomic alterations in the brain should be largely traced to neurogenesis, and not to e.g. neuroectoderm or expanding neural progenitors. This is the case[5,10,11,76]. Of note, genomic mosaicism has been proposed to foster neuronal specialisation[118,119]. Beyond its detrimental effects, it remains to be tested whether FGF-driven DNA replication stress and damage is just tolerated during proliferation or, in addition, could render diversification in neuronal lineages at the cost of stochastic aneuploidy. In that respect, microglia are highly active during neurogenesis to remove aneuploid cells[25], and specific mitotic gene variants have evolved to partially reduce chromosome missegregation in the human developing brain[120].

Given the technical and ethical challenges in testing these predictions in human embryos, the development of in vitro models of human gastrulation[30,117] and brain development[120–122] opens now the opportunity to address these questions.

Finally, our results, together with recent insights on SHH signalling[123], hypoxia[124], pathogens[125], and tissue architecture[126], highlight the importance of extracellular cues, and cell fate for the correct replication, maintenance and segregation of chromosomes in mammalian cells.

In regards to the limitations, this study largely focuses on 2D stem cell models and molecular perturbations to disentangle cell fate and cell signalling effects in DNA replication stress and chromosome

segregation fidelity. Given the tight relationship between patterning signals and cell lineage specification, future molecular and genetic studies using in vivo and 3D stem cell models are required to understand the spatio-temporal roles of signalling gradients in chromosomal stability during development. Of note, chromosomal mosaicism, as well as problems derived from it (i.e. spontaneous miscarriage) are not as prevalent in mouse pre-implantation embryos compared to humans. Furthermore, several mechanisms exist to reduce and cleanse aneuploid cells in embryos, including in blastocysts and in the developing brain[7,25,117]. Critically, the use of recently developed mouse and human 3D stem cell-embryo models (i.e. Blastoids, gastruloids, and EiTiX embryoids) could serve as valuable models to test our predictions, and should be taken in account for further follow-up studies. This study provides limited information about the roles of FGF signalling in vivo during developmental neurogenesis, which might require precise perturbations disentangling the role of this pathway in cell cycle progression, cell fate commitment and genome stability at this critical phase of embryonic development. Further studies are required (i) to unravel the complexity of the replication mechanisms controlled by each signalling cascade–especially considering the multiple functions of the co-regulated proteins RIF1 and BRCA1[17,52–56], (ii) to determine the contribution of other cues to this signalling hub, as well as (iii) to understand their dependency on ATM activity. Finally, future genome sequencing analyses are required to determine the consequences of cell signalling perturbation, especially upon replicative stress, towards mutations, copy-number-variations and other structural chromosomal alterations.

## Methods

### Ethics statement

Work with hESCs was conducted at the MRC Laboratory of Molecular Biology (LMB) under approval from the UK Stem Cell Bank Steering Committee, and in accordance with the regulations of the UK Code of Practice for the Use of Human Stem Cell lines. H9 hESCs were kindly provided by M. Lancaster (LMB) under an agreement with WiCell.

Time-mate pregnant C57BL/6N wild-type mice for cortical isolations were purchased from Janvier. Animals had *ad libitum* access to food and water and were kept under a 12 h light–2 h dark cycle. All animal experiments were approved by Regierungspräsidium Karlsruhe, Germany and reported in compliance with the ARRIVE guidelines. All methods were conducted following German Animal Welfare Act regulations under A.P. supervision. C57BL/6N wild-type female mice used for in utero injections were maintained and bred at the DKFZ central mouse facility. In utero injection, experiments were approved by the local animal welfare committee *Regierungspräsidium Karlsruhe* following the guidelines from GV-SOLAS (AZ 35-9185.81/G-94/18 from J.A.). For terminal tissue harvesting procedures, pregnant mice were euthanized using cervical dislocation and embryos were decapitated, following the approved animal facility procedures.

### Cell culture

The hiPSCs were a gift from Kyung-Min Noh (EMBL). Cells were seeded in wells coated with Vitronectin 1 h at 37 °C (VTN 1:100 diluted in PBS), and grew in hiPSC culture medium. In detail, cells were cultured in Essential E8 medium (Thermo Scientific) supplemented with Penicillin/Streptomycin. We added Revitacell Supplement (Thermo Scientific) for the first 24 h after plating. Media were changed every day, and cells were split every 3–4 days using Versene (Gibco). All the experiments were carried out with hiPSCs between passages 10 and 20. For immunofluorescence experiments, 50.000 cells were seeded per each 12-well.

The mouse feeder-free embryonic stem cell line Sox1-GFP was a gift from A. Smith (University of Cambridge), and the line E14Tg2a was a gift from C. Niehrs (DKFZ). Cells were cultured in DMEM + 15% Panserum (PANSERA) supplemented with Leukaemia Inhibitor Factor (LIF)

and Penicillin/Streptomycin. Prior to seeding, well plates were coated with Gelatin 0.1% diluted in PBS for 10 min at room temperature. Routine passage was carried out every 2 days, after seeding 100,000 cells/well in 6-well plates. For immunofluorescence experiments, 50,000 cells were seeded per each 12-well.

H9 hESCs were cultured in 6-well plates pre-coated with 1.6% growth factor reduced matrigel (356231, Corning) diluted in DMEM F12 (21331-020, Thermo Fisher Scientific) for 30 min at 37 °C in mTeSR Plus (100-0276, StemCell Technologies). hESCs were routinely passaged with EDTA (produced in-house) at a splitting ratio of 1–5 every 3 days, with media change every day. All the cell lines tested negative for the presence of mycoplasma.

Human WNT3A, DKK1-FLAG (DKK1), and RSPO3-ΔC-AP (RSPO3)[127] conditioned media were generated in-house using the corresponding basal media of interest and tested regularly in WNT reporter assays, as previously indicated[42,128].

### Cell lineage specification experiments

mESCs were cultured in 2i (MEKi + GSK3i) + LIF medium prior to induction towards primed (mEpiLCs) and formative (mFSCs) pluripotency.

For induction of mouse epiblast-like cells (EpiLCs), mESCs were seeded at a density of $1.0 \times 10^5$ ESCs on a well of a 12-well plate coated with human plasma fibronectin (16.7 µg/mL) in N2B27 medium containing activin A (20 ng/mL), bFGF (12 ng/mL), and KSR (1%) for 2 days, as described[129].

For induction of mouse formative stem cells (mFSCs), mESCs were seeded at a a density of $1.0 \times 10^5$ ESCs on a well of a 12-well plate coated with human plasma fibronectin (16.7 µg/mL) in N2B27 medium containing low doses of Activin A (3 ng/mL), 2 µM XAV939 (WNT inhibitor) and 1.0 µM BMS493 (Retinoic Acid Receptor inhibitor), as described[130]. Cells were cultured in these conditions for a minimum of three times prior to the experiment. Note that for the assessment of DKK1-driven chromosome missegregation in mFSCs, cells were cultured without XAV939 during the last passage prior to harvesting, and ultimately treated with DKK1 for 16 h.

Primitive streak, mesoderm and endoderm lineages were generated as previously described[32]. Briefly, we plated hiPSCs in low confluency (1:25) for 2 days in a hiPSC culture medium prior to differentiation. During the differentiation procedures, we grew cells in A-RPMI media supplemented with GlutaMax and Penicillin/Streptomycin. To generate primitive streak-like cells, we treated hiPSCs with 5 µM CHIR99021 for 1 day, followed by overnight treatment with WNT3A-conditioned medium (1:5) and 50 ng/mL Activin A. For paraxial mesoderm, we treated hiPSCs with 4 µM CHIR99021, 30 ng/mL Activin A, 20 ng/mL FGF2 and 100 nM PIK90 for 1 day, followed by treatment with 3 µM CHIR99021, 1 µM A830, 250 nM LDN and 20 ng/mL FGF2 for another day. For lateral mesoderm, we treated hiPSCs with 6 µM CHIR99021, 30 ng/mL Activin A, 40 ng/mL BMP4, 20 ng/mL FGF2 and 100 nM PIK90 (PI3Ki), followed by 1-day treatment with 1 µM A8301 (TGFBRi), 30 ng/mL BMP4 and 1 µM C59 (PORCNi).

Neuroectoderm, neural crest and NSCs were generated as previously described[33]. Briefly, we plated hiPSCs in low confluency (1:35) for 1 day in hiPSC culture medium prior to differentiation. During the differentiation, we cultured cells in E6 media supplemented with Penicillin/Streptomycin. To generate neuroectoderm, we grew hiPSCs in the presence of 200 ng/mL Noggin and 10 µM SB431542 (TGFBi) for 5 days, changing media on days 2 and 4 of the differentiation protocol. To generate neural crest-like cells, we treated hiPSCs for 5 days with 200 ng/mL Noggin, 20 µM SB431542, and 1:5 WNT3A conditioned medium, changing the media on days 2 and 4. To generate NSCs, we treated hiPSCs for 5 days with 200 ng/mL Noggin, 20 µM SB431542, and 1:5 DKK1 conditioned medium, changing the media on day 2 and day 4. Please note that NSCs represent an earlier stage of neural differentiation compared to hiNPCs.

Hepatocyte-like cells were generated from hiPSCs as described before[131]. Briefly, we plated hiPSCs in low confluency (1:35) for 2 days in a hiPSC culture medium prior to differentiation. During the differentiation procedures, we grew cells in A-RPMI media supplemented with Gluta-Max and Penicillin/Streptomycin. To generate hepatocyte-like cells, we treated hiPSCs with 1:5 WNT3A 50 ng/mL Activin A, and 0.6% DMSO for 2 days, followed by 50 ng/mL Activin A, 0.6% DMSO for another 2 days to generate definitive endoderm. To further differentiate into hepatocyte-like cells, we treated the cells with 50 ng/mL BMP4 and 0.6% DMSO for 2 days, followed by 2 days with 20 ng/mL FGF1 and 0.6% DMSO. To generate more mature hepatocyte-like cells, cells were treated with 20 ng/mL HGF supplemented with 2% DMSO until day 20 of differentiation. Before changing the composition of the growth medium, cells were washed with DMEM/F12. The medium was changed every 2nd day.

Cardiomyocyte-like cells were generated as previously described[32]. Briefly, we plated hiPSCs in low confluency (1:25) for 2 days in a hiPSC culture medium prior to differentiation. During the differentiation procedures, we grew cells in A-RPMI media supplemented with Gluta-Max and Penicillin/Streptomycin. To generate cardiomyocyte-like cells, we treated hiPSCs with 30 ng/mL, 40 ng/mL BMP4, 6 μM CHIR99021, 20 ng/mL FGF2, and 100 nM PIK90 for 1 day, followed by another day culture with 1 μM A8301, 30 ng/mL BMP4, 1 μM + C59 to generate lateral mesoderm. To further differentiate into cardiac-mesoderm, we treated the cells with 1 μM A8301, 30 ng/mL BMP4, 1 μM C59, and 20 ng/mL FGF2 for 2 days, followed by 30 ng/mL BMP4, 1 μM XAV939, and 200 μg/mL Phospho-ascorbic-acid to reach cardiomyocyte-like cells. Before changing the composition of the growth medium, cells were washed with DMEM/F12. The medium was changed every 2nd day.

Human-induced neural progenitors (hiNPCs) were generated using previous protocols with few modifications[80]. In detail, hiPSCs were cultured in a hiPSC culture medium for 4 days until reaching 70–80% confluence. Cells were dissociated using Dispase (5 min at 37 °C), washed with PBS, and resuspended in T75 flasks medium consisting of 50% E8 Medium + 50% Neural Induction Medium (NIM, DMEM/F12 (Gibco) supplemented with 1× N2 (Gibco), 1× non-essential amino acids and 1× Penicillin/Streptomycin) in order to aggregate in non-adherent conditions (Day 0). Media was changed on Day 1 (50% E8 Medium + 50% NIM), prior culture from Day 2–7 in daily changed 100% NIM media by slowly centrifuging the cell aggregates (embryoid bodies) at ×3g. On Day 7, 12-well plates with glass coverslips were coated using poly-D-Ornithine (15 μg/mL) 3 h at 37 °C. Embryoid bodies were collected and seeded at a confluence of around 15–20 EBs/12-well in NIM media. Media was changed every two days, and cells were harvested on Day 10 (early NPCs) and 16 (late/mature NPCs).

In Fig. 6 and Supplementary Fig. 6j, signal molecules were modified during the last 16 h of treatment as indicated in each individual panel.

## Protein and small compound treatments

Where indicated, cells were treated for 16 h or 3 h with 1:3 human DKK1 conditioned medium, 1:3 human WNT3A conditioned medium, 40 ng/mL recombinant human FGF2 (R&D), 5 ng/mL recombinant human TGFB1 (Peprotech), 200 ng/mL recombinant human Noggin (R&D), 100 ng/mL recombinant human EGF (R&D), 100 ng/mL recombinant human BMP4 (Peprotech), 50-200 nM aphidicolin (Sigma-Aldricht), 3 μM AZD0156 (ATMi) (Selleckchem), 2.5 μM CPT (Selleckchem), 3 mM HU, 3 μM CHIR99021 (GSK3i) (Selleckchem), 20 μM nucleoside mix (dNs) (SCBT), 1–10 nM taxol (Sigma), 0.1 μM FGFRi (Merck), or 10 μM LGK-974 (Selleckchem). A complete list of morphogens, growth factors, and small compounds as well as the concentration they were used in Fig. 1b is shown in Supplementary Data 1.

## Chromosome segregation and DNA damage analyses

For chromosome segregation analyses, cells were treated for 16 h, as indicated. For DNA damage analyses (γ-H2AX) cells were supplemented with EdU for 30 min prior to treatment to visualize S-phase cells, and after treated for 3 h prior to collection. Culture cells were fixed in 2–4% PFA for 10–15 min, permeabilised with 0.5% Triton X100 in PBS (PBST) for 10 min, followed by a blocking step for 20 min and overnight incubation with primary antibodies in 2% horse serum in 0.1% PBST. We used 1:250 guinea pig anti-CENPC antibody (MBL International Corporation, USA, cat no PD030) to stain kinetochores (Chromosome segregation), or 1:250 mouse anti-phospho-H2AX (Ser139) (γ-H2AX) antibody (Millipore, clone JBW301) to detect DSBs, 1:250 mouse anti-phospho-RPA32/RPA2 (Ser4 + 8) (pRPA) antibody to detect single strand DNA breaks, 1:250 mouse anti-BLM (SantaCruz Biotech) to detect ultrafine bridges; and the secondary antibodies 1:500 anti-guinea pig Cy3 (Millipore, AP308P), 1:500 anti-mouse Alexa488 (ThermoFisher A21202), supplemented with 1 μg/ml DAPI. In all the figures where DNA damage is measured (gH2AX, pRPA), incorporated EdU was subjected to a click-it reaction (ThermoFisher, C10337), as indicated by the manufacturer.

H9 hESCs were platted at a ratio of 1:15 in 12 well plates on the top of 15 mm coverslips pre-sterilized in ethanol 100%, washed in sterile ddH2O and coated with 1.6% Growth Factor Reduced Matrigel diluted in DMEM F12. Two days after plating, cells were treated with 250 ng/mL of DKK1; 40 ng/mL of FGF2, or 200 ng/mL of Noggin for 16 h. Cells were fixed using 4% paraformaldehyde (PFA) (15710, Electron Microscopy Sciences) diluted in PBS for 20 min at room temperature and then washed three times with PBS−0.1% Tween. The permeabilisation step was performed in PBS containing 0.3% Triton X-100 and 0.1 M glycine for 30 min at room temperature. Samples were incubated with primary antibodies (CENPC−MBL, PD030 and α-tubulin−Sigma, T9026) diluted at 1:250 in blocking solution (3% BSA and 0.1% Tween in PBS) overnight at 4 °C. The day after, samples were washed with PBS−0.1% Tween three times, followed by incubation with secondary antibodies (donkey anti-guinea pig 647, AP193SA6 Millipore, and donkey anti-mouse 488, A-21202 ThermoFisher Scientific), diluted at 1:500 in blocking solution for 2 h at room temperature.

To quantify cells exhibiting chromosome missegregation, we analysed 100–200 anaphases (hiPSCs, mESCs) and 50–100 anaphases (mNPCs, hiNPCs, lineages) in each biological replicate using either a Nikon Eclipse Ti using a 60× objective with oil immersion; or an inverted SP8 confocal microscope (Leica Microsystems) with a Leica 63×/1.4NA Oil objective and an upright Olympus VS200 slide scanning microscope with a 40×/0.95NA Air objective (hESCs). In mESCs, hESCs, hiPSCs and their derived lineages, chromosomes clearly separated (Lagging) from the bulk of segregated DNA chromatids were considered as chromosome missegregation. In mouse NPCs, which display a lower distance between the two masses of separating chromosomes, we quantified chromosomes clearly separated from or bridging between the bulk of segregated DNA, as previously shown (Figs. 4a, d[25] and 5g[10]). Follow-up analyses in Supplementary Fig. 6 determined further classify the missegregated chromosomes in CENPC positive (centric) and negative (likely acentric) anaphases from human iPSCs. Basal levels of chromosome missegregation in hiPSCs and mouse NPCS (E14.5) were in accordance with previous estimates[10,45].

To characterise DNA damage, EdU stained nuclei were automatically selected, and the median fluorescence intensity (MFI) was calculated, using ImageJ Fiji 2.0.0-rc-69/1.52p, after background subtraction, similarly as described[64]. This approach was validated using different concentrations of aphidicolin from 25 nM to 500 nM and was utilised instead of foci counting, due to the high basal numbers of γ-H2AX foci in hiPSCs.

## DNA combing experiments

Single-molecule DNA combing assays in hiPSCs and hiNPCs were performed to determine replication fork progression speed and inter-origin firing distances (OFD). Cells were treated with proteins or small molecule compounds for 3 h as indicated before harvesting. To label the replication forks, cells were incubated with two 30 min consecutive

pulses of 100 µM 5-Chloro-2′-deoxyuridine (CldU; Sigma-Aldrich) and, 100 µM 5-iodo-2′-deoxyuridine (IdU; Sigma-Aldrich), respectively. Cells were harvested and processed using the FiberPrep DNA extraction kit (Genomic Vision, France), as indicated by the manufacturer. Isolated DNA was immobilized on vinylsilane engraved coverslips (Genomic Vision, France) using the Molecular Combing System (Genomic Vision, France) at 2 kb/µm. Coverslips were stained with 1:10 anti-BrdU/CldU (BU1/75 (ICR1), Abcam, ab6326), 1:10 anti-BrdU/IdU (B44, BD Biosciences, 347580), 1:5 anti-ssDNA (IBL, 18731), and secondary conjugated antibodies to Cy5 (1:25, Abcam, UK, cat no ab6565), Cy3.5 (1:25, Abcam, UK, ab6946), and BV480 (1:25, BD Biosciences, 564877). Images were acquired using a Nikon CREST microscope using a 60× objective with oil immersion and autofocus (ssDNA). Using NIS Elements software, we stitched together five wide-field images along the longitudinal axis of the combed DNA. We analysed at least 100 labelled unidirectional DNA tracks per sample to determine replication fork progression rates. To determine inter-OFD, the distance between two neighbouring origins on the same DNA strand was measured for at least 45 origin pairs per condition. Similarly as shown before[48], we selected bi-directional forks with both IdU and CldU signals with (1) a gap between the CldU segments (Fired before the beginning of the pulse) and (2) with a continuous CldU segment sandwiched by IdU signals (Fig. 3b), which were fired during the first pulse and, as such, might also include origins triggered by replicative stress during the acute treatments. We did not select single isolated IdU signals, which could include origins fired during the second pulse, as many of those could also correspond to improperly labelled forks. These analyses consistently rendered average inter-OFD for control hiPSCs of ~100 kb, similar to previous estimates (111 kb)[132].

## Karyotype analyses

Cells were treated for 16 h, as indicated, followed by 16 h arrest using 100 ng/mL nocodazole.

Giemsa staining and karyotype analysis were performed as described previously[28]. Briefly, cells were pelleted, washed with PBS, and incubated in a hypotonic medium (40% DMEM-F12, 60% H2O) at RT for 15 min. Cells were fixed in Carnoy's solution (methanol:acetic acid = 3:1). Chromosomes spread onto glass slides and stained with Giemsa solution, and imaged using a Nikon Eclipse Ti with a 60× oil immersion objective.

M-FISH was performed as previously described[133]. Briefly, seven pools of flow-sorted human whole chromosome painting probes were amplified and combinatorial labelled using DEAC-, FITC-, Cy3, TexasRed, and Cy5-conjugated nucleotides and biotin-dUTP and digoxigenin-dUTP, respectively, by degenerative oligonucleotide-primed (DOP) PCR. Prior to hybridisation, metaphase spreads fixed on glass slides were digested with pepsin (0.5 mg/mL; Sigma) in 0.2 N HCL (Roth) for 10 min at 37 °C, washed in PBS, post-fixed in 1% formaldehyde, dehydrated with a degraded ethanol series and air dried. Slides were denatured in 70% formamide/1x SSC for 2 min at 72 °C. Hybridization mixture containing combinatorial labelled chromosome painting probes, an excess of unlabelled cot1 DNA in 50% formamide, 2× SSC, and 15% dextran sulphate were denatured for 7 min at 75 °C, pre-annealed for 20 min at 37 °C, and hybridized to the denatured metaphase preparations. After 48 h incubation at 37 °C slides were washed three times at room temperature in 2× SSC for 5 min, followed in 0.2× SSC/0.2% Tween-20 at 56 °C two times for 7 min. For indirect labelled probes, an immunofluorescence detection was carried out. Therefore, biotinylated probes were visualized using three layers of antibodies: streptavidin Alexa Fluor 750 conjugate (Invitrogen), biotinylated goat anti avidin (Vector) followed by a second streptavidin Alexa Fluor 750 conjugate (Invitrogen). Digoxigenin-labelled probes were visualized using two layers of antibodies: rabbit anti-digoxin (Sigma) followed by goat anti-rabbit IgG Cy5.5 (Linaris). Slides were washed three times in 4× SSC/0.2% Tween-20 for 5 min, counterstained

with 4.6-diamidino-2-phenylindole (DAPI) and covered with anti-fade solution. Images of metaphase spreads were taken for each fluorochrome using highly specific filter sets (Chroma Technology, Brattleboro, VT) on a DM RXA epifluorescence microscope (Leica Microsystems, Bensheim, Germany) equipped with a Sensys CCD camera (Photometrics, Tucson, AZ). Camera and microscope were controlled by the Leica Q-FISH software and images were processed on the basis of the Leica MCK software and presented as multicolour karyograms (Leica Microsystems Imaging solutions).

## Single-cell RNA-sequencing

For single-cell sequencing analyses, hiPSCs were treated for 16 h with 1:3 DKK1 conditioned medium, 40 ng/mL recombinant FGF2 (R&D), or 200 ng/mL recombinant Noggin (R&D). Live hiPSCs were stained with DAPI to exclude death cells, Annexin V to exclude pre-apoptotic cells, and DRAQ-5 to sort G1 cells. Cells were FACS sorted in 5µL lysis buffer (Quiagen) in single wells of a 96-well plate, and the RNA was separated using Biotin-dT30-bound streptavidin beads (Thermo Fischer, 18064014). After cDNA synthesis and amplification using SuperScript III Reverse transcriptase kit (Thermo), samples were cleaned up with 0.6× SPRI beads and tagmented with homemade Tn5 at 55 °C for 3 min. We performed a final PCR amplification of 12 cycles using a KAPA HiFi kit (Roche) with unique pairs of i5/i7 adaptor index primer, proceeded with 0.8× SPRI bead purification, pooled samples after DNA concentration quantification in Bioanalyzer, and performed a final 0.75× SPRI clean up. Samples were sequenced by NextSeq 2000 (Illumina) and with 75 paired-end (PE) reads. Read alignments and the count tables of mapped read per gene were obtained using STAR version 2.6.0a with GRCh38 human reference genome and its gene model (GRCh38.84). For differential gene expression analysis, the R package Seurat v 4.1.1[134] was applied. We filtered out genes that were detected in less than three cells and selected high-quality cells that had less than 25% mitochondrial counts and more than 4000 and less than 12,000 detected genes. After normalizing the data for differences in library size, the 'FindMarkers' function with the ROC test was used to determine differentially expressed genes (DEGs) between the different treatment conditions. We selected DEGs with power >0.25 for the subsequent GO analysis (https://david.ncifcrf.gov/tools.jsp, 07/2022).

## qRT-PCR and FACS analyses

RNA was extracted and purified using a Quiagen RNAeasy Plus column kit, according to the manufacturer's instructions. The cDNA was produced with SensiFAST cDNA Synthesis kit (Bioline) starting from 300 ng to 1 µg mRNA. Real-time quantitative PCR reactions from 8.3 ng of cDNA were set up in technical triplicate using the SensiFAST SYBR Hi-ROX kit (Bioline) on a StepOne Plus (ThermoScientific) and/or qTower qPCR machine. The sequences of the oligonucleotides used in this study are provided on request. Expression levels were normalized to PCR amplification with primers for GAPDH. The list of mouse and human primers used can be found in Supplementary Data 4.

Cell cycle profiles were performed in hiPSCs treated as indicated for 16 h with the selected factors. Prior to harvesting, cells were pulsed for 1 h using BrdU (Sigma). Cells were collected in ice-cold PBS, fixed in ice-cold ethanol (final concentration 70%) and stained using an anti-BrdU antibody and propidium iodide as described before[128].

## MS sample preparation and analysis

hiPSCs were seeded in 10 cm dishes in E8 media and treated with control, DKK1, Noggin or FGF2 (See Supplementary Data 1) for 3 h before harvesting. Raw data files can be found in the PRIDE repository under the PXD054388 accession number. Tgfb1 (5 ng/mL) and GSK3 inhibition (CHIR99021 3uM) treatment conditions were not taken into account in this study. Four independent experiments (n = 4 biological replicates/condition) were processed for phospho-proteome analysis was performed similarly as described previously[42]. After lysis 400 µg of

protein was used as starting material. Peptide samples were prepared as indicated before[135]. In short, proteins were digested using an in-solution protocol using Lys-C and Trypsin. For full proteome analysis prior to the enrichment procedure a fraction of the peptide sample was separated (2 µg), vacuum dried and stored until the LC-MS/MS analysis. The resulting peptides were subjected to a phospho-peptide enrichment step based on Fe-IMAC in a column format. The phospho-peptide fraction was desalted (StageTip) and stored until the LC-MS/MS analysis.

The LC-MS/MS analysis of phospho-proteome and full proteome samples was carried out on an Ultimate 3000 UPLC system (Thermo Fisher Scientific) directly connected to an Orbitrap Exploris 480 mass spectrometer for a total of 150 min. Peptides were online desalted on a trapping cartridge (Acclaim PepMap300 C18, 5 µm, 300 Å wide pore; Thermo Fisher Scientific) for 3 min using 30 µl/min flow of 0.05% TFA in water. The analytical multistep gradient (300 nl/min) was performed using a nanoEase MZ Peptide analytical column (300 Å, 1.7 µm, 75 µm × 200 mm, Waters) using solvent A (0.1% formic acid in water) and solvent B (0.1% formic acid in acetonitrile). For 132 min the concentration of B was linearly ramped from 4% to 30% or 2$–28% (phospho-proteome), followed by a quick ramp to 78%, after two minutes the concentration of B was lowered to 2% and a 10 min equilibration step appended. Eluting peptides were analysed in the mass spectrometer using a data-dependent acquisition mode. A full scan at 120 k resolution (380–1400 $m/z$, 300% AGC target, 45 ms maxIT) was followed by up to 2 s of MS/MS scans. Peptide features were isolated with a window of 1.4 m/z (1.2 $m/z$, phospho-proteome), and fragmented using 26% NCE or 28% NCE (phospho-proteome). Fragment spectra were recorded at 15 k resolution (100% AGC target (200% for phospho-proteome, 54 ms maxIT). Unassigned and singly charged eluting features were excluded from fragmentation and dynamic exclusion was set to 35 s. Each sample was followed by a wash run (40 min) to minimize carry-over between samples. Instrument performance throughout the course of the measurement was monitored by regular (approx. one per 48 h) injections of a standard sample and an in-house shiny application.

Data analysis was carried out by MaxQuant (version 1.6.14.0, Tyanova, Stefka, Tikira Temu, and Juergen Cox. "The MaxQuant computational platform for mass spectrometry-based shotgun proteomics." *Nature Protocols* 11.12 (2016): 2301) using an organism-specific database extracted from Uniprot.org (human reference database, containing 79,038 unique entries from 3rd January 2022). Settings were set to default with the following adaptions. Match between runs (MBR) was enabled to transfer peptide identifications across Raw files based on accurate retention time and $m/z$. Fractions were set in a way that MBR was only performed within replicates of the full proteome or the phospho fraction, respectively. Separate parameter groups were assigned for full proteome and phospho fractions. For the phospho fractions PTM was set to *True* and in the parameter group settings, *Phospho (STY)* was added to the variable modifications. For the full proteome fraction label free quantification (LFQ) was enabled with default settings.

Quantification on the full proteome fraction was done using the LFQ approach based on the MaxLFQ algorithm[51]. A minimum of two quantified peptides per protein was required for protein quantification. In addition, iBAQ-values[136] were generated.

For the downstream analysis, only peptides with fold changes >2 and $P$-values <0.1 for the four experiments were considered differentially regulated and are shown in Supplementary Fig. 5. Each of the factors was manually analysed in STRING, PhosphoSitePlus, and other databases for (i) their role in DNA replication or/and damage and (ii) known kinases modulating the identified phospho-sites.

## dNTP and NTP measurements

After 3 h treatments, cells were washed with ice-cold Tris-buffered saline and harvested in 500 µl of 15% (w/v) trichloroacetic acid and 30 mM MgCl$_2$, snap-frozen in liquid N$_2$, and stored at −80 °C. Cell lysates were thawed on ice, vortexed, and mixed on an Intelli-Mixer for 10 min at 4 °C. The samples were centrifuged at 16,000×g for 3 min at 4 °C to obtain a clear supernatant, and the supernatants were processed as described[137,138]. The level of dNTP in pmol for each sample is calculated using the dNTP standard of 100 pmol, run at the same time as the samples. The level of dNTP is expressed as pmol relative to the protein (mg) of the TCA precipitate of the corresponding sample quantified using the Bradford reagent.

## Luciferase reporter assays

For the Wnt reporter (TOPflash) assays, hiPSCs were seeded on a 96-well plate and transfected with 50 ng DNA per well, including 5 ng Firefly luciferase and 3 ng Renilla luciferase, filled up with empty vector (pCS2+). The transfection was performed using Lipofectamine3000 transfection reagent (Invitrogen) following the supplier's protocol. To measure transcriptional Hippo activity, we followed the same procedure but used 10 ng of HOP-Flash plasmid (instead of Firefly). TOPflash and Renilla plasmids were kindly provided by C. Niehrs and HOPflash plasmid was obtained from Addgene (#83467).

## Small interfering RNA (siRNA)

Universal siControl and mission siRNAs against human LRP6 and CTNNB1 were obtained from SIGMA (For more information, please check Supplementary Data 4). They were validated by qPCR and TOPflash. In order to optimize transfection efficiency, hiPSCs were reverse-transfected prior to seeding with 50 nM siRNA using Dharmafect1 transfection reagent (Horizon Discovery) following the manufacturer's guidelines. Cells were grown for two days and harvested.

## Live cell imaging

For chromosome missegregation tracking, hiPSCs and mNPCs were cultured as described before in µ-Slide 8-well chambers, and incubated with 500 nM SiR-DNA (Spirochrome AG, SC007) 1 h prior to and during the experiment. In preliminary analyses, we validated that this concentration of SiR-DNA did not induce mitotic delay or phenotypes. Treatments with selected factors were applied overnight before starting the imaging. Live cell imaging was performed using an automated Nikon Eclipse Ti2 inverted microscope equipped with a 40× water immersion objective (Nikon Apo LWD, NA 1.15) and an NEO sCMOS camera (Andor). Multipoint acquisition was controlled by NIS Elements 5.1 software. 5 z-stacks with 2-µm intervals were recorded every 5 min for up to 5 h in a preheated chamber (STXG-WSKM, Tokai Hit) at 37 °C and 5% CO$_2$. Images were analysed using ImageJ 2.0.0 software.

For microtubule dynamics analyses, cells were seeded onto 8-well glass-bottom imaging chambers (Ibidi) and transfected the next day with pEGFP-EB3 plasmid (kindly provided by L. Wordeman, University of Washington) using Lipofectamine 3000. Cells were treated for 16 h, as indicated. Before imaging, cells were treated for 30 min with 2 µM Dimethylenastron to induce monopolar spindles, as previously described[28]. Cells were live-imaged using a 100 × 1.45 NA oil objective, in 4 × 0.4 µm Z-optical sections with an additional 1.5× magnification switch, every 2 s for over 30 s, in a humid chamber with 37 °C, 5% CO$_2$.

## Single-cell EdU sequencing analyses

scEdU-seq was performed as previously described by van den Berg et al. in ref. 50. Briefly, hiPSC cells were treated with indicated conditions and labelled with a double pulse labelling (15 min 10 µM EdU, 60 min. chase with full medium and 15 min. 10 µM EdU). Subsequently, cells were fixed in 75% ethanol for up to 24 h and stored in a storage buffer (42.5 mL, H$_2$O RNAse free, 5 mL DMSO, 1 mL 1 M HEPES pH 7.5, 1.5 mL 5 M NaCl, 3.6 µl pure spermidine solution, 0.05% Tween and 200 µL 0.5 M EDTA) up to 1 year. Prior to sorting, cells Biotin-PEG3-Azide were clicked onto EdU molecules and cells were stained with

DAPI. Single cells were sorted into 384-well plates using the Cytoflex SRT. Libraries were generated by protein digestion using Proteinase K, NlaIII restriction enzyme digestion, end repair of fragments, dA-tailing and finally adaptor ligation. Libraries were pooled and Streptavidin MyOneC1 (Invitrogen) beads were used to affinity purify EdU-containing DNA molecules. These molecules were subjected to heat denaturation, after which the non-EdU biotin single strand was made double-stranded by the Klenow Large Fragment fill-in reaction. The restored fragments were amplified by in vitro transcription, reverse transcription and finally PCR. Libraries were sequenced on Illumina Nextseq2000 P2 2x100bp. Scripts to process and analyse the data are available at https://github.com/vincentvbatenburg/scEdU-seq.

### Western blotting

For Western blotting, cells were lysed in full lysis buffer (50 mM Tris·HCl, pH 7.5, 150 mM NaCl, 1% Nonidet P-40, 0.05% SDS, 1 mM β-mercaptoethanol, 2 mM EDTA, 1× protease phosphatase inhibitor mixture [Thermo Scientific]). The cleared lysates were mixed with 4× NuPAGE LDS sample buffer (Thermo Scientific) containing 50 mM DTT, resolved on 10% NuPAGE gels and transferred to nitrocellulose membranes. For Western blot experiments, the following antibodies were used: rabbit anti-GAPDH (14C10 mAb, Cell Signalling, #2118), rabbit anti-phospho-Chk2 Thr68 (C13C1, Cell Signalling #2197), rabbit-anti-phospho-Chk1 Ser345 (133D3, Cell Signalling, #2348), mouse anti-Total Chk1 (2360, Cell Signalling), and rabbit-anti-Total Chk2 (2662, Cell Signalling).

### Antibodies

The following primary antibodies were used for immunofluorescence/FACS at a dilution 1:250 (otherwise stated): guinea pig anti-CENPC antibody (MBL International Corporation, USA, cat no PD030), mouse anti-phospho-H2AX (Ser139) (γ-H2AX) antibody (Millipore, clone JBW301), mouse anti-phospho-RPA32/RPA2 (Ser4 + 8) (pRPA), mouse anti-BLM (SantaCruz Biotech), mouse anti-BrdU diluted 1:5 (BD Biosciences, 556028), mouse anti-Tubulin, beta III isoform, CT, clone TU-20 (Millipore, MAB1637), rabbit anti-Nestin (Millipore, ABD69), chicken anti-Nestin (Novus, NB100-1604), rabbit anti-phospho-FGFR1 (Merck, 06-1433), rabbit anti phospho-LRP6 S1490 (Cell Signalling Technology, 2568S), and rabbit anti-phospho-Ser10-Histone 3 (pHis3) (Millipore, 06, 570).

The following primary antibodies were used for DNA combing experiments: 1:10 anti-BrdU/CldU (BU1/75 (ICR1), Abcam, ab6326), 1:10 anti-BrdU/IdU (B44, BD Biosciences, 347580), and 1:5 anti-ssDNA (IBL, 18731).

The following primary antibodies were used for western blotting: rabbit anti–GAPDH (14C10 mAb, Cell Signalling, #2118), rabbit anti-phospho-Chk2 Thr68 (C13C1, Cell Signalling #2197), rabbit-anti-phospho-Chk1 Ser345 (133D3, Cell Signalling, #2348), mouse anti-Total Chk1 (2360, Cell Signalling), and rabbit-anti-Total Chk2 (2662, Cell Signalling).

The following secondary antibodies were used at a dilution 1:500 (otherwise stated): anti-guinea pig Cy3 (Millipore, AP308P), anti-mouse Alexa488 (ThermoFisher A21202), donkey anti-guinea pig 647 (AP193SA6 Millipore), donkey anti-mouse 488 (A-21202 ThermoFisher Scientific), donkey anti-rabbit 594 (A-21207 ThermoFisher Scientific), goat anti-chicken 488 (Thermo, A-11039), anti-Cy5 (dilution 1:25, Abcam, UK, cat no ab6565), Cy3.5 (dilution 1:25, Abcam, UK, ab6946), and BV480 (1:25, BD Biosciences, 564877). For western blotting, anti-mouse IgG HRP (Millipore, AP308P) and anti-rabbit IgG HRP (Millipore, AP307P) were used at dilutions 1:5000.

### Mouse studies

E13.5 C57BL/6N wild-type embryos were injected in utero either with 5 ng/μL DKK1 or with PBS + 0,1% BSA control solution. In detail, pregnant mice were anaesthetized with isoflurane, the uterine horns were exposed, and 1 μl of the solution was injected into the lateral ventricle of each embryo using glass micropipettes. Animals were sacrificed 16 h later, and embryonic heads were isolated in cold PBS, followed by fixation with 4% PFA for 3 days. Afterwards, embryonic heads were cryoprotected in 30% sucrose solution and embedded in Tissue-Tek OCT. Embryonic coronal forebrain sections (18 μm) were prepared using a Leica CM1950 cryo-microtome at the DKFZ Light Microscopy Core Facility. Cryosections were subjected to antigen retrieval using 1% sodium citrate, blocked in 0.1% PBST, and incubated overnight at 4 °C with 1:250 anti-phospho-FGFR1, anti-phospho-LRP6 S1490, anti-Nestin or/and anti-phospho-Ser10-Histone 3 (pHis3).

Mouse neural progenitor cells were dissociated from the neocortex of E14.5 mouse embryo brains using the papain dissociation kit (LK003150, Worthington Biochemical Corporation) following the manufacturer's instructions. Prior to seeding, 12-well plates with glass coverslips were coated overnight at 4 °C using Poly-D-Ornithine (15 μg/mL), and rinsed three times in PBS prior to seeding. Dissociated NPCs were seeded at a 500,000 cells/cm$^2$ density in a medium consisting of: Neurobasal media supplemented with B27 (1×), Glutamax (1×) and Penicillin/Streptomycin. Media was changed daily and primary cultured cells were harvested after 48 h for experiments.

### Statistics and reproducibility

The chromosome segregation analysis has been performed by four different scientists resulting in consistent results, and in the case of hESCs, performed blindly by another research group. Unbiased analysis of data was carried out wherever possible. All the data and statistical significance were analysed using Prism 9 (GraphPad). Data are shown as mean with standard deviation of biological replicates after independent confirmation of the results, as indicated in the figures; except for γ-H2AX analyses in Figs. 4 and 7 and Supplementary Figs. 5 and 6, where median values are represented. Where indicated, Student's t-tests with two-tails (two groups) or ordinary one-way ANOVA analyses with Tukey correction (three or more groups) were calculated. In Fig. 5a, fold changes and statistical analyses were calculated from the experiments shown in Supplementary Fig. 7 comparing control vs DKK1 treatments. The significance of GO groups from DEGs from Fig. 3a was obtained from DAVID (https://david.ncifcrf.gov/tools.jsp, 07/2022). Significance is indicated as: *$P<0.05$, **$P<0.01$, ***$P<0.001$, or n.s.: not significant.

### Reporting summary

Further information on research design is available in the Nature Portfolio Reporting Summary linked to this article.

## Data availability

All data generated in this study are provided in the article file, Supplementary Information, and Supplementary Data. The relevant source data from each figure are provided in the Source Data files. Single-cell RNA-sequencing datasets generated during this study and disclosed in Fig. 3a and Supplementary Fig. 3 are available in the ENA database, with accession number PRJEB76601. Single cell EdU-sequencing datasets generated during this study and disclosed in Fig. 3j, k and Supplementary Fig. 4 are available at the GEO database, with accession number GSE271478. The phospho-MS data generated in this study and represented in Supplementary Fig. 5a–d are provided in the Supplementary Information, in the Source Data file and the raw data in the PRIDE repository with accession number PXD054388. Source data are provided in this paper.

## Code availability

For single-cell EdU-seq analysis, all scripts to process raw data and generate figures are available at https://github.com/vincentvbatenburg/scEdU-seq. For single-cell RNA-seq analysis, the standard pipeline was used to write the code.

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

## Acknowledgements

We thank A. Smith, K.M. Noh, G. Pereira, M. Lancaster, and C. Niehrs for sharing reagents and cells. We thank A. Ciprianidis and N. Giebel for experimental support. We thank the DKFZ proteomics facility, the Nikon Imaging Center, and the FACS facility at the University of Heidelberg for proteomics analyses, access to microscopes and cytometers, and technical help. We thank the EMBL sequencing facility for technical support and sequencing analyses. This work was funded by the Deutsche Forschungsgemeinschaft (DFG, German Research Foundation)—SFB 1324—Project number 331351713 (Project B03 to S.P.A. and H.B.). A.d.J.-S. held a Humboldt Research Fellowship for Postdoctoral Researchers and a Young Marsilius Kolleg fellowship. J.H. is the recipient of a Studienstiftung PhD fellowship. J.A. and A.P. were supported by the Chica and Heinz Schaller Foundation. M.S. was supported by the Medical Research Council, as part of UK Research and Innovation (grant reference MRC, MC_UP_1201/24). V.S.R is supported by a Milstein fellowship.

## Author contributions

H.B. and S.P.A. designed the research; A.d.J.-S., J.H., A.B., and S.P.A. conceived, performed and analysed experiments. A.H., B.d.M., N.B., J.J.M.L., B.S., V.S.R., L.V., U.E, Y.B., B.D., S.A., M.T., J.v.d.B., V.v.B., and A.J. performed and analysed experiments. A.C., V.B., M.S., A.J., A.P., R.S., A.v.O., J.A., and H.B. contributed new reagents/analytic tools and supervised aspects of the project. J.J.M.L, J.B., J.v.d.B., and V.v.B. analysed the single-cell sequencing data. S.P.A. supervised the project and wrote the paper with the input of the other authors.

## Funding

## Competing interests

The authors declare no competing interests.

## Additional information

¹Centre for Organismal Studies (COS), Heidelberg University, Heidelberg, Germany. ²Department of Molecular Oncology, Section for Cellular Oncology, University Medical Center Göttingen (UMG), Göttingen, Germany. ³Oncode Institute, Utrecht, The Netherlands. ⁴Hubrecht Institute, Utrecht, The Netherlands. ⁵KNAW (Royal Netherlands Academy of Arts and Sciences), Utrecht, The Netherlands. ⁶University Medical Center Utrecht, Utrecht, The Netherlands. ⁷Department of Medical Biochemistry and Biophysics, Umeå University, Umeå, Sweden. ⁸Department of Clinical Neurobiology, University Hospital Heidelberg and German Cancer Research Center (DKFZ), Heidelberg, Germany. ⁹Genomics Core Facility, European Molecular Biology Laboratory (EMBL), Heidelberg, Germany. ¹⁰Institute of Human Genetics, Heidelberg University, Heidelberg, Germany. ¹¹MRC Laboratory of Molecular Biology, Cambridge, UK. ¹²Schaller Research Group, German Cancer Research Center (DKFZ), Heidelberg, Germany. ¹³Nikon Imaging Center at the University of Heidelberg, Bioquant, Heidelberg, Germany. ¹⁴Division of Molecular Thoracic Oncology, German Cancer Research Center (DKFZ), Heidelberg, Germany. ¹⁵These authors contributed equally: Anchel de Jaime-Soguero, Janina Hattemer. ¹⁶These authors jointly supervised this work: Holger Bastians and Sergio P. Acebrón. ✉e-mail: sergio.acebron@cos.uni-heidelberg.de

