## [Peer Review File · Nature Communications]

REVIEWER COMMENTS

Reviewer #1 (Remarks to the Author):

In this manuscript, De Jaime-Soguero and colleagues describe a novel function of some important signalling pathways in DNA repair. The pathways, Wnt, BMP, Nodal and FGF have well established functions in cell fate decisions during development and the finding represents a novel and important contribution to our understanding of their function. The function appears to be restricted to stem cells during self renewal and, as such, provides insights into the mechanisms that create resilience to damage in populations of stem cells. The experiments are well performed and described and go beyond description of an effect, combining a number of different techniques to pin point specific points of each signalling pathway in the DNA repair pathway. This is an important piece of work, connecting many pieces of previous literature regarding development and DNA damage/chromatic missegregation.

Here we make a number of suggestions that the authors should consider to improve the manuscript.

In the abstract the authors speak of WNT, BMP, FGF as rheostat that “monitors replication stress”. This is an interesting notion (a variable resistance that allows an adaptation to changes in current) but we are not clear that this is what they observe. They never show that the signalling pathways are linked in the manner of a single device, or they that they respond to quantitative changes in damage in the manner a rheostat would. There is also no evidence that they ‘monitor’ damage, in the way say p53 does. The data shows that these signaling pathways induce or prevent replication stress. This in itself is important but they should not try to infer mechanisms beyond what the data shows. If they do, they should justify it.

1, In the screen an important pathway that is missing and that should be tested is Hippo. This should be checked.

2. There is a clear correlation between phenotypes of gain and loss of function in hiPSCs and mESC. Since hiPSCs are more comparable to mEpiSCs, one wonders whether this indicates that the exact level of pluripotency is not relevant for the effects they see. Would human naive cells or mouse EpiSC respond in the same manner?

3. The rescues of damage with the downstream agonists and antagonists e.g ChIR99021 and the FGFRi, make sense, but the upstream signaling rescues are less understandable. Thinking stoichiometrically, without having information of dosage, given that DKK1 is an antagonist of Wnt at the ligand to receptor interface, it is very surprising that Wnt3a administration rescues the phenotype caused by DKK1 treatment. A similar logic applies to the rescue of Noggin treatment by BMP4 administration. The authors later in the manuscript use recombinant DKK1, which means it would be possible to titrate both Wnt3a and DKK1 in this experiment and show that the rescue effect is truly a question of Wnt3a overcoming DKK1 in a concentration dependent manner.

4. In Figures 1 and 2, the percentage of missegregation under DKK1 treatment fluctuated around 10%, while in this experiment the percentage suddenly lies close to 30%. It appears that this experiment was conducted only once with 50 counted cells in the same biological repeat. Conducting this experiment in independent repeats and adding error bars would make it more rigorous.

5. In Figure 3A the authors choose to do single cell RNAseq on 75 cells instead of bulk RNAseq which would have been more informative, especially given that the only result shown is a pseudobulk analysis that mixes all treatment conditions into one. Wnt, BMP, and FGF are very distinct signaling pathways, and it would be great to show the results separately for the individual pathways, rather than compacting them all into one “treated” condition. This would answer the question whether there are points where they converge or differ in their downstream effects.

6. In Fig 3J, the authors observe that their DKK1 treatments produces an average of ~1,000 gamma-H2AX foci per S-phase nucleus, which seems very high. If they are using physiological levels of DKK1 this means that in the human embryo, at the start of gastrulation when there are high levels of DKK1 in the AVE, every dividing cell in the anterior epiblast has an average of 1,000 double strand breaks per division? Is this right? Would they expect the same in the mouse? If one were to stain mouse embryos at this stage against gamma-H2AX and DKK1 one would expect a clear correlation between the two.

Alternatively, does this strong phenotype only hold true in early pluripotent cells that are not supposed to be experiencing Wnt inhibition? In that case, the authors should show experimentally that there are distinct differences in how cells of different pluripotency levels (naive vs. primed vs. formative vs. AVE-like) respond to DKK1 in regard to chromatin missegregation.

7. The experiments on the differentiation pathways are important, However the authors should justify the focus on the last 16 hrs of differentiation when it is likely that the decisions have been taken. What happens if they test the effects of the signalling pathway during the early stages of differentiation? They don't need to do it for all fates, but it would be good to see it in a few.

8. Also, Affidicolin seems to have a differential effect on the endoderm relative to other fates of the primitive streak. Intriguingly the endoderm, like the anterior neural derivatives are subject, early on to Wnt inhibition. Would the authors like to discuss this? Also, the system might be sensitized during differentiation and it would be helpful to see if Wnt inhibition increases the effects of Affidicolin.

9. In Fig S5C, the authors claim that DKK1 treatment does not have an effect on primitive streak stage cells in respect to Chromosome missegregation. The protocol to derive primitive streak-like cells contains treatment with 5 micro-molar ChIR99021, a stable and potent small molecule, as well as treatment with Wnt3a. Given that the authors use an unknown amount of DKK1 from cell culture supernatant, how do they know their treatment actually results in Wnt-inhibition? ChIR has an IC50 of ~10nM, indicating that even small remainders in the well could overpower the effect of the unknown DKK1 concentration. Especially since ChIR acts downstream of DKK1. Similarly, the authors show in an earlier figure that Wnt3a outcompetes Dkk1 in their hands, meaning that Wnt3a remainders in the well might also prevent DKK1 from showing effects on signaling. Can the authors please include quality control experiments particularly here but also in other parts of the manuscript to demonstrate that their alterations of Wnt signaling actually becomes effective?

10. Back to the issue of differentiation v cell renewal. The protocol used to derive neural stem cells/NPCs that the authors use here includes 5 days of treatment with Noggin and DKK1. Wouldn't it be expected that according to their data in the process of differentiation 10-30% of all divisions are missegregated and thus the resulting neuronal cells should be highly aneuploid at the start of the experiment? Could it be that after 5 days of continuous treatment with DKK1, the cells have selected for clones that are genetically more stable and less likely to conduct chromosome missegregation? I believe the experiments on Wnt signaling following a 5 day culture in DKK1 are difficult to interpret and require a better discussion,

What would happen to hiPCs that were only DKK1 treated for 5 days? After this time, would they still be responsive to differences in Wnt levels in regard to Chr. missegregation, or would they become insensitive like primitive streak cells?

11. In Figure S5 it is difficult to separate the effect of the cells' lineage vs the signaling treatment the cells experiences leading up to the experiment. Are cells more or less responsive to the signaling stimuli because they are in a different differentiation state, or because they were treated with activators and inhibitors of the very signaling pathways in question for the days leading up to the experiment? This would be an important point to raise in the discussion or a short "Limitations of the study" section.

Given the points made above, I believe the sentence on page 8: “These results indicate that WNT, BMP, and FGF roles in chromosome segregation fidelity are largely impaired after specification into the three human germ layers, regardless of their critical signalling functions in cell fate determination in those lineages”, may be an overstatement and not sufficiently proven.

12. Figure 5B requires more explanation in terms of how and how long cells were differentiated. The Western blot anti p-CHK1, p-CHK2 should be complemented by WB anti total-CHK1 and total-CHK2 to demonstrate phosphorylation-specific differences and not total abundance differences.

13. According to Figure S5, hiNPCs are generated by 5 days treatment of DKK1, but then the assay in Figure 6C tests 16 hours of DKK1 treatment? The protocol to derive NPCs outlined in Figure 6A may be different from way the authors derive NPCs in Figure S5. The authors seem to use the term “NPC” for multiple different stages of differentiation. This needs to be clarified and the presence or absence of signaling factors in the differentiation protocol needs to be highlighted in Figure 6A. Furthermore, it would be good to show in more detail at what stage of differentiation the NPC are. E.g. Neural Rosette stage should be Hes5 positive, while later neural stem cells should be Hes5 negative again, this would be good to demonstrate with the used protocol.

14. In Figure S6J, the cultures look like a mix between different stages of differentiation. The Tuj1 staining (b-tubulin III) highlights differentiated neuronal lineages, rather than NPCs. The Nestin positive, b-tubIII negative cells are likely NPCs. Furthermore, especially in the FGF2 treated condition, there seems to be a third population that is double negative. What are these cells? GFAP staining would help to test whether these are astrocytic lineages. Given this diversity of the cells present, it would be great to take apart the results of Figure 6C-D by cell type.

15. The experiments in Figure 7 are not really able to inform about the ratio of Wnt and FGF signaling, but rather demonstrate that higher levels of WNT are preventing chromatin missegregation at E14.5 and lower levels of Wnt increase the amount of missegregation and DNA damage. Increased levels of FGF2 increase DNA damage and missegregation, but the beneficial effect of Wnt signaling is dominant over the effect of FGF. In short, WNT is protective against missegregation, FGF increases missegregation, and WNT is dominant over FGF. However, given the dominance of WNT in this, I am not sure that talking about ratios is the correct way of summarizing it.

16. In the discussion, what do the authors mean with “but not in specified human three germ layers or their subsequent differentiated lineages.”? Is neuroectoderm and neurogenesis outside of the realm of three germ layers?

17. In the discussion, the term 'moonlighting' is not appropriate as it might well be that there is no main role of these signalling pathways, simply that the focus has been on one of their functions. For example, one would not call the PCP function of Wnt signalling a moonlighting role. They may want to use 'additional', 'novel' or other related term that extends the function of the signals.

In summary, there is definitely something both interesting and important here but the authors should be careful with the interpretation. The comment contrasting the impact of anteriorizing and posteriorizing signals on DNA damage and chromosome segregation is warranted by the data but is weakened by the fact that what they are measuring is cellular responses in self renewal conditions, not differentiation. All signalling pathways tested have a role in the maintenance of pluripotency in hiPSCs (and EpiSCs, see above) and naïve hPSCs but not in naïve mouse PSCs. So, not sure what it means to talk about differentiation in the context of self renewal. A different thing is if they implied that during anterior differentiation cells have a higher susceptibility to damage. This would be interesting as it is clear that the initial phase of anterior neural development requires a maintenance of pluripotency.

One issue that is left outstanding is the effectors of the functions observed. We appreciate that this might belong to a different study but the authors might mention this in the discussion as something for the future,

Minor comments

In page 6 and others, The authors use the term "Epistatic studies" several times, but I believe it should be "epistasis studies".

In p4: "activation of Transforming Growth Factor (TGF) by TGF- β 1/2, inhibition of Nodal signalling by LEFTY-A, as well as Fibroblast Growth Factor (FGF) signalling activation by various ligands (FGF2/4/8A/8B/9), also promoted chromosome segregation errors in mESCs and hiPSCs (Figure 1B)" This might be a mistake and they mean "inactivation of TGF...".

In p5, when describing the results of the single cell RNAseq, they refer to Figure 2A, when it should be 3A.

Reviewer #2 (Remarks to the Author):

In this interesting manuscript, De Jaime-Soguero et al. introduce the intriguing premise that developmental signaling pathways such as BMP, FGF, and WNT may directly control chromosome segregation. This builds on the authors' past work suggesting that WNT signaling supports faithful chromosome segregation and therefore prevents aneuploidy (PMID 33257473, 25656539). One of the main questions prompted by this line of work is whether these signaling pathways simply direct differentiation into distinct cell-types, and that differing cell-types have different chromosome stability profiles (i.e., whether these signaling pathways indirectly impact chromosome segregation). However, multiple lines of data support the authors' argument that these signaling pathways act directly on chromosome segregation, including the fact that short treatment (16 hours) with various signaling modulators is sufficient to lead to chromosome segregation phenotypes in pluripotent stem cells (PSCs). The effects seen after short-term signaling manipulation suggests that these effects are less likely to reflect differentiation phenotypes observed after manipulating these various signaling pathways. Additionally, conserved chromosome segregation phenotypes are seen in both human and mouse PSCs treated with the same signal (Fig. 1B), suggesting that the observed effects are less likely to reflect differentiation phenotypes, since primed human PSCs and naïve mouse PSCs often differentiate into quite different cell-types after exposure to the same signal.

Multiple other aspects of the paper are also quite interesting. First, these signaling pathways have cell-type-specific effects in controlling chromosome segregation, and do not affect all cell-types similarly (Fig. 5). The authors focus on how these signaling pathways control neural progenitor chromosome segregation and perform an incredibly challenging intrauterine injection experiment to provide in vivo support for their work, which otherwise is largely performed in vitro (Fig. 7). Second, the authors assign specific, and different, mechanisms for how WNT, FGF, and BMP signaling pathways control chromosome segregation through varied mechanisms such as fork progression, origin firing, or microtubule polymerization.

Overall this is quite an intriguing manuscript. While previous studies have implicated developmental signaling pathways in cell cycle regulation, their specific involvement in chromosome segregation was less well known, with the notable exception of the authors' past work on WNT (PMID 33257473, 25656539). More broadly, their results also have implications regarding the future design of cell culture media to expand various types of stem cells and progenitor cells ex vivo, and what signaling pathway manipulations may be detrimental for the long-term genomic stability of these cultured cells.

Major comments:

1. Rigor: The authors should be applauded for their rigorous use of multiple different proteins and small molecules to activate or inhibit a given pathway in order to demonstrate reproducible effects (e.g., Fig. 1C-D). Fig. 1C-D are particularly striking, because they show that signaling pathway levels (as assayed by BMP and WNT target gene expression) closely correlate with the degree of chromosome segregation. However, the authors almost exclusively use extracellular proteins and small molecules throughout the paper, and they may want to consider verifying key conclusions using genetic manipulations, if feasible. However, this reviewer is cognizant that the authors have already performed a massive amount of work for this manuscript, and have backed up their conclusions with multiple independent extracellular protein and small molecule pathway modulators, so this is a suggestion (not a request).

Minor concerns:

1. Given that their work revolves around WNT and chromosome segregation, the authors should consider citing earlier work on how GSK3 inhibitors can induce chromosome instability (PMID 17697341).
2. The authors suggest that FGF pathway activation increases the rate of chromosome mis-segregation. Standard media to maintain primed hPSCs, such as mTeSR, contains high FGF (100 ng/mL of FGF2; PMID 16388305). Does the authors' data have any implications for chromosome stability of undifferentiated hPSCs?
3. It is great that the authors comprehensively investigated the impact of various signaling pathways agonists and antagonists on chromosome missegregation (e.g., Fig. 1B), but do doses matter? This question of pathway dosage is important when considering the authors' model on signaling gradients and tissue patterning, presented in Fig. 1A.
4. One of the key conclusions drawn in this manuscript is how WNT and BMP regulate DNA replication stress. However, if the reviewer understands correctly, the effects of WNT3A, GSK3i, and BMP4 on fork progression and origin firing were tested only in the presence of aphidicolin (Figure 3F, 3G, 3K-N). Do they have the same effects in the absence of aphidicolin?
5. The authors could consider adding DAPI or markers of different cell-cycle phases in Figure 3I and 3M. This additional information could provide insights into the effects of these various signaling pathway perturbations across various cell cycle phases.
6. Figure S5: The authors test whether treatment of small molecules or recombinant proteins perturbs chromosome missegregation during hPSC differentiation. However, many of the manipulations tested will significantly impact hPSC differentiation into the corresponding cell-types, as shown by the papers cited by the authors. This therefore brings up the possibility that the observed chromosomal phenotypes simply reflect perturbed differentiation into different cell-types, which in turn have their own chromosomal stability programs. The authors should include data on the expression of various differentiation markers after each of these perturbations.

7. “We have previously identified that WNT ligands promote chromosome segregation fidelity in somatic and cancer cells 22-24, although the underlying mechanisms and biological relevance remained uncharacterised.” Instead of “biological”, do the authors mean “physiological” here?
8. Fig. 1B: There are discrepant effects of various TGF β pathway modulators. LEFTY (TGF β inhibitor) and two TGF β agonists (TGF β 1 and TGF β 2) all promote chromosome mis-segregation, whereas another TGF β agonist (ACTIVIN) suppresses chromosome mis-segregation.
9. Fig. 1B: Several questions. First, the different colors used to label various circles are a bit unclear. PORCNi (a WNT pathway modulator) is labeled a “compound”, not an “antagonist”, but as the authors know, it is a WNT pathway antagonist. Second, “retinoid acid” should be corrected to “retinoic acid”. Third, should “Lefty A” be re-named to the official gene name (LEFTY2)?
10. Fig. 1F: Is there a negative control condition, where no signaling modulators were added?
11. Fig. 3A: Were hPSCs treated with DKK1, FGF2, and NOGGIN simultaneously? This should be clarified in the Figure Legend.
12. Fig. 3B-N: How long were hPSCs treated with the various signaling modulators? This should be clarified in the Figure Legend.
13. Fig. 6B: The “NPCs” appear heterogeneous, with NESTIN+ NPCs and TUBB3+ neurons co-existing. Does this cellular heterogeneity affect any of the authors’ conclusions regarding “NPCs”?
14. Figure 7I: Define what “pHis3” means, in the Figure Legend or elsewhere.
15. The discovery that the WNT signaling pathway is implicated in microtubule polymerization is intriguing. Perhaps in the Discussion, the authors might consider speaking about the mechanisms through which WNT acts on microtubules, as the authors’ own past work (PMID 25656539) suggested that it is independent of beta-catenin, whereas other previous studies have reported the importance of activation and phosphorylation of beta-catenin in microtubule growth (P-C Hsieh et al., 2007, Oncogene; P Huang et al., 2007, Oncogene).

Reviewer #3 (Remarks to the Author):

Summary:

Dr Jaime-Soguero and Hattemer et al investigate the link between lineage specification and chromosomal instability in this very interesting manuscript. Various literature supports the notion of somatic mosaicism of chromosomal alterations, and outstanding questions remain as to the mechanisms driving initial chromosome segregation abnormalities, and why or how this might vary

between lineages or temporally during development. To address these questions, the authors systematically analyse the impact on chromosome segregation fidelity in a set of models to mimic various cellular pluripotency states and lineages. They test the impact of adding exogenous lineage specification signals to these cellular models and carefully score segregation error rates using microscopy. They find that during pluripotency the addition of some of these signals, but not others, induces replication stress and mitotic defects, which they link mechanistically by the induction of aberrant microtubule dynamics caused by replication stress (shown previously by one of the senior authors). This effect diminishes after pluripotency but reemerges during the differentiation pathway to neural progenitors. Overall this is a very intriguing and well performed study, that is of great interest to the development and genomic stability fields. However, in places the wording is confusing, and for such a study that bridges two fields it may be more important than usual to ensure complete clarity for all readers to gain the full understanding. I have a few concerns that would need to be addressed in order to reassure me that the conclusions are properly supported by the data:

1. The rationale to investigate roles in DNA replication could perhaps be strengthened. H3K4 methylation (di/tri?) factors are identified in the RNA analysis. Did they directly detect increased H3H4me? At promoters? Seems primarily to be a mark of transcription and maybe a minor role in replication stress? Was total gene transcription unchanged? A stronger rationale to make the link to replication stress might be helpful.

2. Page 5: “Conversely, co-treatment with WNT3A, GSK3i, or BMP4 rescued origin firing induced by 100 nM aphidicolin without affecting replication fork velocity (Figure 3E, 3F and 3G), which indicates a role of WNT and BMP signalling in the mitigation of DNA replication stress downstream of stalled forks (Figure 3H).” This is confusing. ‘Rescued’ origin firing might be better rephrased as ‘prevented compensatory origin firing’ if I have understood their data correctly. I am also not sure this can be termed ‘mitigation of replication stress’, since if there is still slowed forks, and now no compensatory origin firing, then it is likely the downstream consequences on genomic stability would be even worse (but this is not their observation). So there seems to be a disconnect here.

3. It is also not clear what the ‘co-treatment’ is with; aphidicolin? Why not co-treat DKK1/FGF2/Noggin with WNT3A/GSK3i/BMP4 as they did when assessing chromosome segregation? Would these co-treatments rescue the replication stress? If not then the effect on chromosome segregation could be via a different mechanism. This is important to strengthen their conclusion that the pathway from lineage specification signals is between replication stress, microtubule dynamics, and whole chromosome mis-segregation.

4. Figure 2g: Only examples of whole chromosome gains are shown. Did they ever see any structural alterations? If the mechanism is as proposed (e.g. replication stress feeds only into aberrant microtubule dynamics and mis-segregation of whole, intact chromosomes, then only whole chromosomal alterations might be expected). Also, were there recurrent aneuploidies observed in any of the treatments? Depending on the timescale of the experiment this might reveal and positive or negative selection for specific aneuploidies which might shed light on the potential benefit of elevated chromosomal instability. How do the abnormalities they detect compare to known mosaicism where previously reported? I.e. could their proposed mechanism explain the study's initiating unexplained observation of this phenomenon?

Minor points:

5. No line numbers so difficult to make specific comments.

6. Confusing wording in places: e.g. in abstract: "Tug-of-war between FGF and WNT triggers chromosome mis-segregation". What does this mean? Chromosome mis-segregation is triggered when one (which one)? Is winning? Or the balance between them is varying?

7. Figure 3K,L,N: the numbers of H2AX/RPA foci are very high (reaching into the thousands). Can they show example segmented images? Are they sure these are real foci?

8. Figure 8 legend: "Cell signalling control of chromosome segregation fidelity is lost after specification into the three germ layers following the withdrawal of ATM signalling as first responder during DNA replication stress, but reemerges during neurogenesis" confusing. Figure looks like fidelity is lost again upon neurogenesis (why?)

9. Hard to follow the logic of paragraph starting on bottom of page 10. Could expand and clarify.

10. The overall larger picture and logic of this intriguing concept is a bit lacking or confusing. Particularly since this work combines two fields, and most readers will be from one field or the other, it is particularly important to explain this intersect in very clear simple language. They seem to allude to a greater concept combining cell fate signalling and extra cellular clues and development in the control of accurate replication and segregation (e.g. last sentence of discussion). However it is not really clear what the rationale/benefit for such an entanglement. Why is increased replication stress or aneuploidy beneficial? If this leads to an increase in cell death how does this play in the scheme of development or cell fate? Could this instead be an unintentional consequence of lineage specification that is under the tolerable limit? The discussion

seems to be hinting that diversification could occur at the cost of aneuploidy, but it is not clear what sort of diversification this means, if aneuploid cells are ultimately removed (as they also mention). Alternative (smaller) chromosomal alterations? A clearer and simpler discussion or speculation on these issues could help place this work in context. Perhaps even some of this literature could be explained in the introduction.

11. Bottom of page 4, used SiR-DNA to track chromosome mis-segregation which cannot distinguish between whole, or large fragments of chromosomes, so cannot be certain these are whole chromosome mis-segregation events. This would need to be determined using either a second live cell marker for the centromere (such as CENP-A-GFP) or with fixed cell immunofluorescence against centromere proteins.

12. During the study, the authors focus on whole chromosome mis-segregation events arising from replication stress-induced increases in MT dynamics. Is the incidence of RS-induced anaphase bridges and acentric lagging chromosome fragments negligible in these cell lines? It is not clear whether they scored these types of errors. If not, then presumably the levels of RS induced by DKK1/FGF2/Noggin are not severe enough to induce these errors?

13. Results section paragraph 1 – should “Table 1” refer to Supplementary Table 1?

14. Fig 1C – right side y axis label should read Relative AXIN2 expression.

15. There is no reference to Fig 1E in the text.

16. Fig 2F-G – is whole chromosome loss ever observed in these cells following DKK1/FGF2/Noggin treatment?

17. Fig 4C – 10 nM Taxol (16hrs) isn't a particularly “low dose” since much lower concentrations have been used in previous studies (e.g. 0.2 – 0.5 nM in Stolz et al. (2015) EMBO Reports). What is the reason for selecting the concentration used here? Are iPSCs inherently less sensitive to Taxol? Would a sub-nanomolar concentration of Taxol still rescue chromosome missegregation?

18. Fig 5B – drawing conclusions from phosphorylation blots in the absence of a blot for total levels of the protein in question should be discouraged. Total CHK1 and total CHK2 western blots should be included to confirm the authors' hypothesis regarding a switch from ATM to ATR signalling.

19. The fork progression rate in untreated hiPSCs is 0.75 kb/min (Fig 3C), but in d16 hiNPCs the control fork rate is ~1.5 kb/min (Fig 6E). Is this increased fork speed with differentiation expected? Could the authors speculate on the reasons for the observed increase?

20. Please ensure all IF images include scale bars with sizes clearly stated in the figure legends.

21. Methods section, Protein and small compound treatments heading, line 1 – please correct references to figures (e.g. there is no Fig 2J or 4G).

Reviewer #4 (Remarks to the Author):

I have a few comments regarding the phosphoproteomics work mentioned in the manuscript by De Jaime-Soguero et al.

1. In the methods section, the details of MS experiments are missing. The reader is guided to reference 36, which itself redirects to another reference. I was able to find some detail in the supplemental document of reference 36, however, we cannot expect the reader of the current manuscript to keep digging. Please include either in the Method section itself or as a supplement, the complete phosphoproteomic workflow, including peptide digestion, mass spectrometer method and parameters, and data analysis steps.

2. Was the phosphoproteomic analysis quantitative, in other words, was SILAC or another labeling incorporated in the samples? I assume it was, since differentially regulated phosphopeptides were shown in Figure S3. In that case, please add that information for the reader.

Dear Referees,

Since the initial review of the manuscript, we have comprehensively addressed the issues raised by you. A detailed point-by-point response to all comments follows below. We now provide additional experimental evidence (New Figures 1F, 3I-K, 4L-M, 5B-E, S1A,D, S2C-H, S3A-D, S4A-F, S5F-J, S6, S8, S9J,K and Table S3), which further support the proposed role of patterning signals in the regulation of chromosome segregation fidelity during pluripotency and neurogenesis. We have also edited and expanded the main text according to your suggestions.

Thank you for your critical input, which we believe has substantially improved our manuscript.

Reviewer #1 (Remarks to the Author):

In this manuscript, De Jaime-Soguero and colleagues describe a novel function of some important signalling pathways in DNA repair. The pathways, Wnt, BMP, Nodal and FGF have well established functions in cell fate decisions during development and the finding represents a novel and important contribution to our understanding of their function. The function appears to be restricted to stem cells during self-renewal and, as such, provides insights into the mechanisms that create resilience to damage in populations of stem cells. The experiments are well performed and described and go beyond description of an effect, combining a number of different techniques to pin point specific points of each signalling pathway in the DNA repair pathway. This is an important piece of work, connecting many pieces of previous literature regarding development and DNA damage/chromatic missegregation.

Thank you.

Here we make a number of suggestions that the authors should consider to improve the manuscript.

In the abstract the authors speak of WNT, BMP, FGF as rheostat that “monitors replication stress”. This is an interesting notion (a variable resistance that allows an adaptation to changes in current) but we are not clear that this is what they observe. They never show that the signalling pathways are linked in the manner of a single device, or that they respond to quantitative changes in damage in the manner a rheostat would. There is also no evidence that they ‘monitor’ damage, in the way say p53 does. The data shows that these signaling pathways induce or prevent replication stress. This in itself is important but they should not try to infer mechanisms beyond what the data shows. If they do, they should justify it.

Our data indicates that these pathways converge into the regulation of DNA replication stress. However, we agree with the referee that some of the terminology we used (Rheostat, monitor) could lead to confusion, especially considering that the pathways modulate replication-associated processes at different levels (Figure 4 and S5). We changed the text accordingly in the title, abstract and discussion.

For the title we now refer to “Developmental signals”, but we will be open to use *morphogenetic* or *patterning* signals.

1, In the screen an important pathway that is missing and that should be tested is Hippo. This should be checked.

Following the referee's advice, we have analysed the effect of Hippo pathway inhibition using Verteporfin, and activation using a recently described compound that blocks LAST1 kinase phosphorylation (TRULI) rendering nuclear YAP activation (Kastan *et al*, 2021). We did not detect a significant effect on chromosome segregation fidelity by modulating the Hippo pathway neither in mESCs nor in hiPSCs (We included the data in Figure 1B). We also included TEAD luciferase reporter assays on the activity of these compounds for your perusal (Referee figure 1).

Referee figure 1: Hippo pathway TEAD reporter assays upon inhibition with the YAP/TAZ inhibitor VTPFN, or activation with the LAST1 inhibitor TRULI.

2. There is a clear correlation between phenotypes of gain and loss of function in hiPSCs and mESCs. Since hiPSCs are more comparable to mEpiSCs, one wonders whether this indicates that the exact level of pluripotency is not relevant for the effects they see. Would human naive cells or mouse EpiSC respond in the same manner?

We now show that DKK1 induced chromosome missegregation in mESCs (naive), mouse epiblast-like cells (mEpiLCs, primed), and formative stem cells (mFSCs), but not in differentiated cells (LIF withdrawal) (New Figure S8D-H; including qRT-PCR analyses of the different cellular pluripotency levels, previously characterized as discussed in the methods). Interestingly, primed cells showed higher responsiveness (Figure S8H; discussed in Page 13, Lines 500-506). It would be interesting to investigate in the future whether this is due to their *bona fide* stem cell status or to differential replication stress tolerance compared to naive cells, as suggested in:

<https://www.biorxiv.org/content/10.1101/2022.05.12.491744v1.full>

3. The rescues of damage with the downstream agonists and antagonists e.g ChIR99021 and the FGFRi, make sense, but the upstream signaling rescues are less understandable. Thinking stoichiometrically, without having information of dosage, given that DKK1 is an antagonist of Wnt at the ligand to receptor interface, it is very surprising that Wnt3a administration rescues the phenotype

caused by DKK1 treatment. A similar logic applies to the rescue of Noggin treatment by BMP4 administration. The authors later in the manuscript use recombinant DKK1, which means it would be possible to titrate both Wnt3a and DKK1 in this experiment and show that the rescue effect is truly a question of Wnt3a overcoming DKK1 in a concentration-dependent manner.

We indeed titrated WNT3A and BMP4 to ensure that we overcome the stoichiometry of the targeted complexes as i) DKK1 and WNT3A compete for LRP6 and ii) Noggin binds and inhibits BMP4. We now show i) qRT-PCR and reporter assays demonstrating that the used WNT3A and BMP4 concentrations can prevent signalling inhibition by DKK1 and Noggin, respectively (Figure S2E,F). Furthermore, and building on our previous results showing WNT activity-dependent chromosome missegregation (Figure 1C), we now show that titration of WNT3A rescues DKK1 in concentration dependent manner (Figure S1A).

Finally, as further described below, we expanded our epistasis analyses for WNT and FGF pathways, for which we had tools available, to further understand at which level of the cascade they impact chromosome segregation fidelity (Figure S2C-D).

4. In Figures 1 and 2, the percentage of missegregation under DKK1 treatment fluctuated around 10%, while in this experiment the percentage suddenly lies close to 30%. It appears that this experiment was conducted only once with 50 counted cells in the same biological repeat. Conducting this experiment in independent repeats and adding error bars would make it more rigorous.

Across our immunofluorescence analyses in Figures 1-4, 4-6% hiPSC anaphases consistently show lagging chromosomes, which is increased to 10-15% upon DKK1. The referee refers to the live cell imaging analyses of hiPSCs cultured with SiR-DNA shown in Figure S2G-J. In these experiments, both control and treatments consistently showed higher levels of chromosome missegregation, likely due to a sum a reasons i) better temporal access to anaphases (2-4 frames per division, and iii) the presence of SiR-DNA, which has been reported to have a mild effect in chromosome missegregation (<https://www.life-science-alliance.org/content/6/12/e202302260>), but likely not iii) photo-toxicity, as imaging was optimized to ensure equal timing across mitoses during the recording (See (Bufe *et al*, 2021)).

We show now the mean of three independent experiments (Figure S2H), with >150 anaphases tracked per condition. Despite possible caveats, we believe that the experiment is useful for the readers to see that hiPSCs survive chromosome missegregation (Figure S2I-J). However, given that this notion is supported by our karyotype analyses, we could also remove these imaging experiments if the referee finds the discrepancy problematic, as they are not central to any conclusion of the manuscript.

5. In Figure 3A the authors choose to do single cell RNAseq on 75 cells instead of bulk RNAseq which would have been more informative, especially given that the only result shown is a pseudobulk analysis that mixes all treatment conditions into one. Wnt, BMP, and FGF are very distinct signaling pathways, and it would be great to show the results separately for the individual pathways, rather than compacting

them all into one “treated” condition. This would answer the question whether there are points where they converge or differ in their downstream effects.

We now also show the effects of each pathway separately in Table S2 and Figure S3B, and validated key differentially expressed factors by qRT-PCR (Figure S3C). We would like to note that these sequencing analyses were our entry point into focusing on DNA replication and highlight processes perturbed during the 16 hours of treatment. We believe that our experiments under acute treatments (3h) in Figures 3-4 and S4-5 are more suited to evaluate how these pathways converge during DNA replication.

6. In Fig 3J, the authors observe that their DKK1 treatments produces an average of ~1,000 gamma-H2AX foci per S-phase nucleus, which seems very high. If they are using physiological levels of DKK1 this means that in the human embryo, at the start of gastrulation when there are high levels of DKK1 in the AVE, every dividing cell in the anterior epiblast has an average of 1,000 double strand breaks per division? Is this right? Would they expect the same in the mouse? If one were to stain mouse embryos at this stage against gamma-H2AX and DKK1 one would expect a clear correlation between the two. Alternatively, does this strong phenotype only hold true in early pluripotent cells that are not supposed to be experiencing Wnt inhibition? In that case, the authors should show experimentally that there are distinct differences in how cells of different pluripotency levels (naïve vs. primed vs. formative vs. AVE-like) respond to DKK1 in regard to chromatin missegregation.

The DNA damage analyses (gamma-H2AX and p-RPA) are displayed as median fluorescence intensity (MFI) instead of foci number. This is because pluripotent stem cells basally display gamma-H2AX foci (For example: (Ahuja *et al.*, 2016)), which mask the effects of DNA replication stress-associated perturbations. For instance, titration of aphidicolin led to concentration dependent increase of gamma-H2AX fluorescence intensity per EdU⁺ cell, but did not consistently increase the number of visible and separated foci (Referee figure 2). Similarly, previous studies in mESCs and hiPSCs show fluorescence intensity of gamma-H2AX as proxy for DNA damage. We now provide a reference (Ahuja *et al.*, 2016) and more detail information in the methods section (Page 46, Lines 1427-1431):

*“To characterise DNA damage, EdU stained nuclei were automatically selected, and the median fluorescence intensity (MFI) calculated, using ImageJ Fiji 2.0.0-rc-69/1.52p, after background subtraction, similarly as described (Ahuja *et al.*, 2016). This approach was validated using different concentrations of aphidicolin from 50 – 500 nM and was utilised instead of foci counting, due to the high basal numbers of γ -H2AX foci in hiPSCs.”*

Referee figure 2: *A*, gamma-H2AX analyses in hiPSCs upon different doses of Aphidicolin. *B*, Median fluorescence intensity (MFI) analyses resolve the effects of sub micromolar concentrations of Aphidicolin. *C*, However, foci counting using Find Maxima in Fiji (as well as other tested approaches) fail to discern changes due to the high basal numbers of foci in pluripotent stem cells.

We believe that a comprehensive analysis of spatio-temporal roles of WNT in genome and chromosomal stability in peri-implantation embryos requires a study on its own, and falls out the scope of this manuscript, and acknowledge in the discussion that using them together with novel 3D stem cell-embryo models would be the next step to assess the impact of morphogens in chromosomal mosaicism (Page 14, Lines 569-5579).

As requested by the referee, we now provide chromosome segregation analyses in naive, formative and primed mouse pluripotent stem cells (New Figure S8D-H, described above), as well as in “anteriorized” and “posteriorized” *in vitro* hiPSCs (Figure 1E,F and S1C,D). These analyses indicate that i) primed cells are more sensitive than naive/formative cells to WNT inhibition and ii) hiPSCs subjected to anteriorising signals display higher levels of chromosome missegregation compared to posteriorising signals.

7. The experiments on the differentiation pathways are important, However the authors should justify the focus on the last 16 hrs of differentiation when it is likely that the decisions have been taken. What happens if they test the effects of the signalling pathway during the early stages of differentiation? They don't need to do it for all fates, but it would be good to see it in a few.

We focused on the last 16 hours (overnight treatments) to ensure that we compared cells committed to either fate. We would like to note that cells have to undergo both S-phase and mitosis to display the indicated mitotic phenotypes (See Figure 4), which precluded us from reducing the treatment time as much as we do in the S-phase analyses.

We followed the referee's advice and now provide chromosome segregation analyses upon WNT inhibition in cells differentiating towards primitive streak and neuroectoderm (Figure 5B,C). Initial commitment to either fate during the first hours reduced the effect of DKK1 in chromosome missegregation compared to pluripotency conditions, while commitment beyond the first day was sufficient to fully impair WNT control of chromosome segregation fidelity. Of note, and in agreement with our previous data (Figure S7), progressive transition towards neuroectoderm leads to increased

basal missegregation, while differentiation towards primitive streak reduced basal levels of chromosome missegregation. This data also aligns with recent findings showcasing differential DNA damage response in neuroectodermal derived lineages respect mesodermal ones (Orlando *et al*, 2021).

8. Also, Affidicolin seems to have a differential effect on the endoderm relative to other fates of the primitive streak. Intriguingly the endoderm, like the anterior neural derivatives are subject, early on to Wnt inhibition. Would the authors like to discuss this? Also, the system might be sensitized during differentiation and it would be helpful to see if Wnt inhibition increases the effects of Affidicolin.

The referee is correct. We see that several fates that rely on WNT inhibition (Figure S7: neuroectoderm, Definitive endoderm, Neural stem cells) generally display higher basal levels of chromosome missegregation (lateral mesoderm is the exception here). It would require further studies to disentangle the contribution of intrinsic and extrinsic factors to replicative stress in each of these lineages.

Definitive endoderm cells origin is evolutionary conserved in vertebrates and require Wnt inhibition during anterior endoderm specification (Davenport *et al*, 2016; McLin *et al*, 2007). However, the endoderm referred by the reviewer corresponds to the anterior foregut endoderm (E9.5), a later stage compared to the one we assessed (definitive endoderm formation from the primitive streak-cells) in mouse developmental time.

As suggested by the referee, we have analysed the functional interaction between DKK1 and aphidicolin. We did not detect a significant functional interaction between DKK1 and Aphidicolin in Definitive Endoderm cells (Referee figure 3). In contrast, we now show that DKK1 cooperated with 25 nM aphidicolin to inducing chromosome missegregation in hiPSCs (New figure 4L).

Referee figure 3: Chromosome segregation analyses in hiPSCs specified towards definitive endoderm upon the indicated treatments.

9. In Fig S5C, the authors claim that DKK1 treatment does not have an effect on primitive streak stage cells in respect to Chromosome missegregation. The protocol to derive primitive streak-like cells contains treatment with 5 micro-molar ChIR99021, a stable and potent small molecule, as well as treatment with Wnt3a. Given that the authors use an unknown amount of DKK1 from cell culture supernatant, how do they know their treatment actually results in Wnt-inhibition? ChIR has an IC50 of ~10nM, indicating that even small remainders in the well could overpower the effect of the unknown

DKK1 concentration. Especially since ChIR acts downstream of DKK1. Similarly, the authors show in an earlier figure that Wnt3a outcompetes Dkk1 in their hands, meaning that Wnt3a remainders in the well might also prevent DKK1 from showing effects on signaling. Can the authors please include quality control experiments particularly here but also in other parts of the manuscript to demonstrate that their alterations of Wnt signaling actually becomes effective?

While CHIR99021 (CHIR) IC50 is indeed in the nanomolar range, in cell culture experiments only > 0.5 micromolar CHIR has a significant impact in WNT activity (See Referee figure 4). We have in the current figures several examples in which we switch between WNT OFF and ON producing the expected lineages (e.g., endoderm, lateral and paraxial mesoderm and neural crest cells), similarly as done by many others before to induce lineage specification (See Figure S7A and associated references). As shown in the analyses of cell fate markers (Figure S7B-K), this approach does not render signalling mishap issues as GSK3i (and other signals) are diluted two orders of magnitude below their functional concentration in cell culture.

We summarise the concentrations used for all signals/compounds in Table S1, and we also include now WNT reporter assays upon WNT3A and DKK1 (co)treatments (Figure S2E).

Referee figure 4: WNT reporter assays (TOPflash) of hiPSCs treated with indicated concentrations of the GSK3 inhibitor CHIR99021.

10. Back to the issue of differentiation v cell renewal. The protocol used to derive neural stem cells/NPCs that the authors use here includes 5 days of treatment with Noggin and DKK1. Wouldn't it be expected that according to their data in the process of differentiation 10-30% of all divisions are missegregated and thus the resulting neuronal cells should be highly aneuploid at the start of the experiment? Could it be that after 5 days of continuous treatment with DKK1, the cells have selected for clones that are genetically more stable and less likely to conduct chromosome missegregation? I believe the experiments on Wnt signaling following a 5 day culture in DKK1 are difficult to interpret and require a better discussion. What would happen to hiPSCs that were only DKK1 treated for 5 days? After this time, would they still be responsive to differences in Wnt levels in regard to Chr. missegregation, or would they become insensitive like primitive streak cells?

We would like to note that most chromosome missegregation events are known to result in growth disadvantage and are lost after few passages.

We performed the suggested experiment: Treatment of hiPSCs with DKK1 in E8 media (pluripotency conditions) for 5 days resulted in cells with elevated chromosome missegregation (New Figure 5D). Unlike hiPSCs differentiated to neuroectoderm with TGFβi+ Noggin (Figure S7A,H,I), continuous treatment with DKK1 for 5 days did not make hiPSCs insensitive to WNT perturbations: DKK1 withdrawal, as well as WNT activation by GSK3i or WNT3A, partially rescued chromosome missegregation in these cells (New Figure 5D). Unlike cells differentiated to neuroectoderm (Figure S7H), hiPSCs treated with DKK1 in E8 for 5 days retained the expression of pluripotency genes (Figure S8A).

Karyotype analyses showed similar levels of aneuploidy in cells treated for 1 or 5 days with DKK1 (Figure S8B,C). As discussed above, this result is consistent with a loss of fitness associated with aneuploidy (i.e., most aneuploidic cells would not further progress, and therefore we likely measure *de novo* missegregation events). We believe that to fully characterise the impact of aneuploidy on future DNA replication stress or chromosome missegregation events require detailed single genome analyses of large populations, which falls out of the scope of this manuscript.

An important conclusion from the above indicated experiments is that signal perturbation and cell identity (pluripotency), but not exposure or “differentiation” time, are key determinants for the cell signal control of chromosome segregation fidelity.

11. In Figure S5 it is difficult to separate the effect of the cells' lineage vs the signaling treatment the cells experiences leading up to the experiment. Are cells more or less responsive to the signaling stimuli because they are in a different differentiation state, or because they were treated with activators and inhibitors of the very signaling pathways in question for the days leading up to the experiment? This would be an important point to raise in the discussion or a short “Limitations of the study” section.

We agree with the referee that these phenomena are difficult to disentangle. As discussed above, several lines of evidence now support that cell potency/fate (pluripotency and neurogenesis) and not their signalling *history* or duration of the treatment determines the responsiveness of cells towards signalling control of chromosome segregation fidelity: 1) mESCs, mFSCs, mEpiLCs (new data) and hiPSCs, which are maintained with different media and growth factors, all show responsiveness to the analysed factors. 2) Cells differentiated towards paraxial vs lateral mesoderm react to opposite changes in WNT, BMP and FGF signalling in the context of cell fate, but not in chromosome segregation fidelity (Figure S7A,F,G). 3) hiNPCs and mNPCs were not treated with WNT inhibitors or activators prior the analyses. Still, cells committed to neurogenesis in both cases, but not earlier during self-renewal, respond to WNT perturbations towards chromosome segregation. 4) We show that WNT, FGF and BMP roles in chromosome segregation depend on ATM (Figure 5F), which does not monitor the response to replicative stress in the differentiated lineages (Figure 5E,G). 5) As shown above, cell signalling perturbation induces chromosome missegregation even after 5 days, as far as cells stay in pluripotency.

We believe that our new results further support a critical role of cell signalling regulating chromosome segregation fidelity selectively during pluripotency and neurogenesis.

Given the points made above, I believe the sentence on page 8: “These results indicate that WNT, BMP, and FGF roles in chromosome segregation fidelity are largely impaired after specification into the three human germ layers, regardless of their critical signalling functions in cell fate determination in those lineages”, may be an overstatement and not sufficiently proven.

Building on the new data and the referee’s suggestions, we toned down the statement at that part of the manuscript (p9, lines 365-368): *“These results indicate that WNT, BMP, and FGF roles in chromosome segregation fidelity are largely impaired after exit from pluripotency, regardless of their critical signalling functions in cell fate determination in the three human germ layers”.*

12. Figure 5B requires more explanation in terms of how and how long cells were differentiated. The Western blot anti p-CHK1, p-CHK2 should be complemented by WB anti total-CHK1 and total-CHK2 to demonstrate phosphorylation-specific differences and not total abundance differences.

We now show total CHK1/2 protein levels (Figure 5E). We are comparing cells during the initial day of differentiation (day 0) and the day 2 following the protocol shown in Figure S7A. A detailed explanation of differentiation procedures is now in the Methods section (Cell lineage specification experiments, Page 43-44), and summarized in the figure legend (New figure 5E).

13. According to Figure S5, hiNPCs are generated by 5 days treatment of DKK1, but then the assay in Figure 6C tests 16 hours of DKK1 treatment? The protocol to derive NPCs outlined in Figure 6A may be different from way the authors derive NPCs in Figure S5. The authors seem to use the term “NPC” for multiple different stages of differentiation. This needs to be clarified and the presence or absence of signaling factors in the differentiation protocol needs to be highlighted in Figure 6A. Furthermore, it would be good to show in more detail at what stage of differentiation the NPC are. E.g. Neural Rosette stage should be Hes5 positive, while later neural stem cells should be Hes5 negative again, this would be good to demonstrate with the used protocol.

We utilized two protocols to generate neural stem/progenitor cells. In the current Figure S7, we used a differentiation protocol that allowed us to generate and compare neuroectoderm, self-renewing neural stem cells and neural crest-like cells (Tchieu *et al*, 2017). In Figure 6 and Figure S9H-L, we use a standard protocol to generate neural progenitor cells committed to cortical neurogenesis (Liu *et al*, 2013), more in line with the hepatocyte and cardiomyocyte-like lineages of Figure S9A-G from a developmental time perspective. To avoid confusion, we mark now the neural stem cells in figure S7 as “NSCs”.

The induced Neural stem cells (NSCs) in Figure S7 behave similarly as self-renewing hiNPCs at day 10 in Figure 6, and self-renewing mouse neural progenitors from E12.5 embryos (Figure 7). We now clarify these different models in the text and methods.

We have performed Hes5 expression analyses. Hes5 is a Notch-directed neuronal marker present in early neural progenitor cells. In our system, *HES5* mRNA increases from pluripotency until Day 16 NPC differentiation (See Referee Figure 5A). Furthermore, HES5 protein is present in the majority of Day10 and Day16 hiNPCs (Referee Figure 5B). This data is in concordance with (Edri *et al*, 2015)-Figure 1. The authors observe indeed HES5 protein levels in human iPSC derived NPCs from neuroectodermal stage until late-radiogial stage (~Day 35 differentiation protocol). Thus, we believe that our Day 16 hNPCs are not yet mature enough to lose *HES5* expression, since they are more likely to represent an early radiogial progenitor stage, rather late.

Referee figure 5: A, HES5 expression in hiPSCs (Day 0) or cells differentiating towards NPCs for the indicated days. HES5 protein levels in hiNPCs at day 10 and 16.

14. In Figure S6J, the cultures look like a mix between different stages of differentiation. The *Tuj1* staining (*b-tubulin III*) highlights differentiated neuronal lineages, rather than NPCs. The Nestin positive, *b-tubIII* negative cells are likely NPCs. Furthermore, especially in the FGF2 treated condition, there seems to be a third population that is double negative. What are these cells? GFAP staining would help to test whether these are astrocytic lineages. Given this diversity of the cells present, it would be great to take apart the results of Figure 6C-D by cell type.

As we now show in figure S9J,K, most of the mitotic hiNPCs are Nestin⁺/Beta-Tubulin III⁻, which were the ones we analysed. We performed the GFAP staining as suggested and they were negative. Although we know the antibody works in mouse and should react with human GFAP, we cannot be sure about its specificity as no astrocyte-like cells were found in our hNPC preparations to ensure its validation.

15. The experiments in Figure 7 are not really able to inform about the ratio of Wnt and FGF signaling, but rather demonstrate that higher levels of WNT are preventing chromatin missegregation at E14.5 and lower levels of Wnt increase the amount of missegregation and DNA damage. Increased levels of FGF2 increase DNA damage and missegregation, but the beneficial effect of Wnt signaling is dominant over the effect of FGF. In short, WNT is protective against missegregation, FGF increases missegregation, and WNT is dominant over FGF. However, given the dominance of WNT in this, I am not sure that talking about ratios is the correct way of summarizing it.

We rephrased the text accordingly, and indicate in the results and discussion that high WNT protects or rescues from FGF effects, instead of mentioning ratios (Page 12, Lines 477-481). We also discuss that, despite the expression of several WNT ligands, basal WNT activity in NPCs is not sufficient to prevent FGF-driven chromosome missegregation. Our results suggest that the wave of FGF activity during neurogenesis, which is mimicked *ex vivo* by adding FGF2, as previously shown, could contribute to the high levels of chromosome missegregation of the developing brain (also stated in the abstract). We also indicate in the limitations of the study that further perturbations of FGF2 activity *in vivo* are required to address and disentangle the spatio-temporal roles of this pathway during developmental neurogenesis (Page 14, Lines 579-583).

16. In the discussion, what do the authors mean with “but not in specified human three germ layers or their subsequent differentiated lineages.”? Is neuroectoderm and neurogenesis outside of the realm of three germ layers?

We now indicate: “we showed that cells rely on extracellular signals for faithful chromosome segregation before exit from pluripotency and during neurogenesis, but not in other early specified human lineages (e.g., Primitive streak, mesoderm) or subsequent differentiated lineages such as hepatocyte- and cardiomyocyte-like cells.” (Page 13, Lines 502-505).

17. In the discussion, the term ‘moonlighting’ is not appropriate as it might well be that there is no main role of these signalling pathways, simply that the focus has been on one of their functions. For example, one would not call the PCP function of Wnt signalling a moonlighting role. They may want to use ‘additional’, ‘novel’ or other related term that extends the function of the signals.

We changed the text accordingly.

In summary, there is definitely something both interesting and important here but the authors should be careful with the interpretation. The comment contrasting the impact of anteriorizing and posteriorizing signals on DNA damage and chromosome segregation is warranted by the data but is weakened by the fact that what they are measuring is cellular responses in self renewal conditions, not differentiation. All signalling pathways tested have a role in the maintenance of pluripotency in hiPSCs (and EpiSCs, see above) and naïve hPSCs but not in naïve mouse PSCs. So, not sure what it means to talk about differentiation in the context of self renewal. A different thing is if they implied that during anterior differentiation cells have a higher susceptibility to damage. This would be interesting as it is clear that the initial phase of anterior neural development requires a maintenance of pluripotency.

We thank the referee for the supporting comment. As per the referee’s suggestions we have toned down some of the statements and further discussed the limitations of the data and the interpretations.

Our previous data, plus the experiments suggested by the referee, strongly support that pluripotent stem cells are the reacting agents towards signaling-dependent chromosome missegregation: i) mouse naive, formative and primed stem cells but not differentiating (-LIF), react to DKK1, ii) hiPSCs react under E8 conditions, as well as under E6+anteriorising signals for 24h, in all cases resulting in cells

expressing pluripotency genes as the controls. As soon cells exit pluripotency (E6+posteriorising signals, commitment to PS, Mesoderm or DE), WNT control of chromosome segregation is also lost. We indicate now more clearly that exit of pluripotency is the main boundary for cell signal control of chromosome segregation fidelity (See Discussion, Page 13, Lines 505-506).

We thank the referee for the comment, which we agree helped clearing up the key relevance of pluripotency, including upon anteriorising signals, for cell signalling control of chr. segregation fidelity.

Although we have map signalling roles across over a dozen defined lineages using 2D approaches, future studies *in vivo* and in 3D stem cell models of early development are required to further disentangle the spatio-temporal roles of patterning signals in genome and chromosomal stability. These future studies would prove highly valuable together with recent evidence of cell competition and apoptosis across early mammalian development as a consequence of chromosomal mosaicism (Bolton et al. 2016).

One issue that is left outstanding is the effectors of the functions observed. We appreciate that this might belong to a different study but the authors might mention this in the discussion as something for the future,

We discuss now the need for future studies to further investigate both the downstream cascades of each pathway, as well as the molecular targets in the limitations sections (Page 15, Lines 583-585). Nevertheless, and for the referee's perusal, we have expanded our signalling analyses of WNT and FGF utilising tools available to our labs. Briefly, we confirmed that WNT functions independently of beta-catenin (Figure S2C, S2D), and likely through WNT/STOP, in the control of chromosome segregation fidelity, as we have shown in cancer cells. We also show now that FGF functions through MEK and ERK to modulate chromosome segregation fidelity (Figure S2C). We briefly discuss these points in the results and discussion.

Minor comments

In page 6 and others, The authors use the term "Epistatic studies" several times, but I believe it should be "epistasis studies".

The referee is right. We correct it.

In p4: "activation of Transforming Growth Factor (TGF) by TGF- β 1/2, inhibition of Nodal signalling by LEFTY-A, as well as Fibroblast Growth Factor (FGF) signalling activation by various ligands (FGF2/4/8A/8B/9), also promoted chromosome segregation errors in mESCs and hiPSCs (Figure 1B)" This might be a mistake and they mean "inactivation of TGF...?"

We are not sure whether the referee refers to TGFB1/2 or LEFTY. We mean that TGFB1/2 activates TGF signalling, and that LEFTY inhibits primarily Nodal signalling. There are studies on other models showing that TGFB signalling can also be inhibited by LEFTY.

If the referee prefers it, we could indicate: inhibition of Nodal/TGF signalling by LEFTY2,

In p5, when describing the results of the single cell RNAseq, they refer to Figure 2A, when it should be 3A.

Corrected

Reviewer #2 (Remarks to the Author):

In this interesting manuscript, De Jaime-Soguero et al. introduce the intriguing premise that developmental signaling pathways such as BMP, FGF, and WNT may directly control chromosome segregation. This builds on the authors' past work suggesting that WNT signaling supports faithful chromosome segregation and therefore prevents aneuploidy (PMID 33257473, 25656539). One of the main questions prompted by this line of work is whether these signaling pathways simply direct differentiation into distinct cell-types, and that differing cell-types have different chromosome stability profiles (i.e., whether these signaling pathways indirectly impact chromosome segregation). However, multiple lines of data support the authors' argument that these signaling pathways act directly on chromosome segregation, including the fact that short treatment (16 hours) with various signaling modulators is sufficient to lead to chromosome segregation phenotypes in pluripotent stem cells (PSCs). The effects seen after short-term signaling manipulation suggests that these effects are less likely to reflect differentiation phenotypes observed after manipulating these various signaling pathways. Additionally, conserved chromosome segregation phenotypes are seen in both human and mouse PSCs treated with the same signal (Fig. 1B), suggesting that the observed effects are less likely to reflect differentiation phenotypes, since primed human PSCs and naïve mouse PSCs often differentiate into quite different cell-types after exposure to the same signal.

Multiple other aspects of the paper are also quite interesting. First, these signaling pathways have cell-type-specific effects in controlling chromosome segregation, and do not affect all cell-types similarly (Fig. 5). The authors focus on how these signaling pathways control neural progenitor chromosome segregation and perform an incredibly challenging intrauterine injection experiment to provide in vivo support for their work, which otherwise is largely performed in vitro (Fig. 7). Second, the authors assign specific, and different, mechanisms for how WNT, FGF, and BMP signaling pathways control chromosome segregation through varied mechanisms such as fork progression, origin firing, or microtubule polymerization.

Overall this is quite an intriguing manuscript. While previous studies have implicated developmental signaling pathways in cell cycle regulation, their specific involvement in chromosome segregation was less well known, with the notable exception of the authors' past work on WNT (PMID 33257473, 25656539). More broadly, their results also have implications regarding the future design of cell culture media to expand various types of stem cells and progenitor cells ex vivo, and what signaling pathway manipulations may be detrimental for the long-term genomic stability of these cultured cells.

We thank the referee for the critical discussion of our results and for acknowledging the challenges of the study.

Major comments:

1. Rigor: The authors should be applauded for their rigorous use of multiple different proteins and small molecules to activate or inhibit a given pathway in order to demonstrate reproducible effects (e.g., Fig. 1C-D). Fig. 1C-D are particularly striking, because they show that signaling pathway levels (as assayed by BMP and WNT target gene expression) closely correlate with the degree of chromosome segregation. However, the authors almost exclusively use extracellular proteins and small molecules throughout the paper, and they may want to consider verifying key conclusions using genetic manipulations, if feasible. However, this reviewer is cognizant that the authors have already performed a massive amount of work for this manuscript, and have backed up their conclusions with multiple independent extracellular protein and small molecule pathway modulators, so this is a suggestion (not a request).

Thank you!

As discussed above with Referee #1, we faced a complex challenge disentangling cell fate from signalling effects. We favoured small compounds and recombinant proteins, as most genetic perturbations would impact cell fate before we could measure their effects in e.g., chromosome segregation fidelity.

We now further expand our epistasis analyses using additional validated compounds targeting downstream FGF signalling (Figure S2C). As suggested, we also performed knock down experiments for the WNT signalling pathway using siRNAs validated previously in our lab. We found that depletion of the canonical WNT co-receptor LRP6 phenocopied DKK1 treatment (Figure S2C,D). Knockdown of beta-catenin with a previously validated siRNA had no effect on chromosome segregation fidelity despite a strong effect in reporter assays (Figure S2C,D). These results are in agreement with our previous data in cancer cells suggesting that WNT/STOP – a WNT/LRP6/GSK3 dependent, but beta-catenin independent cascade, regulates chromosome segregation fidelity. Given the complex roles of WNT components in mitosis (Habib and Acebron 2021), we believe that further studies are required to disentangle other roles of WNT signalling cascades in the studied lineages.

Minor concerns:

1. Given that their work revolves around WNT and chromosome segregation, the authors should consider citing earlier work on how GSK3 inhibitors can induce chromosome instability (PMID 17697341).

We now indicate:

“Together with previous studies (Augustin et al, 2017; Habib & Acebron, 2022; Lin et al, 2021; Stolz et al, 2015), our work highlights a critical role of β -catenin-independent WNT/LRP6 signalling (Figure

S2C), possibly through the post-translational cascade WNT/STOP (Acebron et al, 2014), in reducing DNA replication stress, modulating microtubule polymerisation dynamics and promoting chromosome segregation fidelity. Intriguingly, alterations on key Wnt components that increase Wnt activity, including GSK3, have also been shown to induce chromosome segregation defects in other contexts (Alberici & Fodde, 2006; Fodde et al, 2001; Hadjihannas et al, 2006; Kikuchi et al, 2010; Rashid et al, 2018; Ross et al, 2014; Rusan & Peifer, 2008; Tighe et al, 2007). Furthermore, loss β -catenin can increase astral microtubule dynamics (Huang et al, 2007). Although many of these phenotypes are caused by WNT signalling independent mechanisms (Habib & Acebron, 2022), it remains to be characterised whether both excess or depletion of WNT activity outside of particular parameters might risk genome maintenance, similarly as it occurs with the up- and down-regulation of replication fork speed (Maya-Mendoza et al, 2018).”

As indicated in the paragraph above, it is important to note that several of the phenotypes associated with WNT components, including APC, beta-catenin and some of the GSK3i studies have been shown to be independent of WNT signalling (Alberici & Fodde, 2006; Fodde et al., 2001; Kikuchi et al., 2010), reviewed in (Habib & Acebron, 2022). For instance, knock down or inhibition of GSK3 in some cancer cell lines can also lead to chromosome segregation defects (Tighe et al., 2007), due to a direct regulation of the localization of Mad2, BubR1, and Bub1 during the SAC (Rashid et al., 2018), a function that has not been identified for WNT ligands. We think that it would be important to explore in the future how differential (low and high) WNT activity impacts chromosomal stability, especially in stem cells.

2. The authors suggest that FGF pathway activation increases the rate of chromosome mis-segregation. Standard media to maintain primed hPSCs, such as mTeSR, contains high FGF (100 ng/mL of FGF2; PMID 16388305). Does the authors' data have any implications for chromosome stability of undifferentiated hPSCs?

The standard culture levels of FGF2 may indeed contribute to the abnormally high basal levels of replicative stress and chromosome missegregation in hiPSCs/hESCs (Chromosome missegregation: 5% in hiPSCs vs <2% in other human cell lines such as RPE1 or HCT116). However, it is important to note that mESCs are not subjected to exogenous FGF and still show 5% missegregation rates, as well as similar response to the one of hiPSCs upon addition of exogenous FGF2. Of note, pluripotent stem cells display several special features associated with cell cycle progression, including short G1, which might contribute to these phenotypes.

Given that human primed fate is tightly linked to FGF activity, further studies of different long-term culture conditions are needed to resolve this question. For your perusal, we substituted E8 media (100 ng/mL FGF2) with E6 +/- 100 ng/mL FGF2 overnight and did not observe significant changes in chromosome segregation fidelity (Referee figure 6), similarly as in our previous analyses comparing E6 vs E8 (Figure S20).

In light of our results, we think it would be important to revisit different hiPSC/hESC culture conditions to reduce the risk of aneuploidy derived of long-term culture, as well as alternative molecules capable to maintain the primed pluripotent state without driving replicative stress (See discussion).

Referee figure 6: Chromosome segregation analyses in hiPSCs transferred for 16 hours to E6 media, E6 media supplemented with TGFB1 and FGF2 (E6++, equal to E8 media) or E6 media supplemented with TGFB1 (E6+TGFB1).

3. It is great that the authors comprehensively investigated the impact of various signaling pathways agonists and antagonists on chromosome missegregation (e.g., Fig. 1B), but do doses matter? This question of pathway dosage is important when considering the authors' model on signaling gradients and tissue patterning, presented in Fig. 1A.

Beyond the results of Figure 1C for WNT and Figure 1D for BMP, and in agreement of the importance of signalling gradients, we now show that different levels of WNT3A and DKK1 determine chromosome missegregation rates in hiPSCs (Figure S1A). Furthermore, we now mimic Anterior vs Posterior commitment in 2D using hiPSCs and find that anteriorising signals in E6 media for 24 hours induce chromosome missegregation (Figure 1E,F and Figure S1C-D). Although we use standard concentrations of signals for these experiments, analyses with 3D *in vitro* and *in vivo* models should further clarify how physiological signalling gradients impact chromosome segregation fidelity during embryogenesis and axial patterning (See the limitations section; Page 14, Lines 569-579).

4. One of the key conclusions drawn in this manuscript is how WNT and BMP regulate DNA replication stress. However, if the reviewer understands correctly, the effects of WNT3A, GSK3i, and BMP4 on fork progression and origin firing were tested only in the presence of aphidicolin (Figure 3F, 3G, 3K-N). Do they have the same effects in the absence of aphidicolin?

To address this question, we focused on WNT activity, as it functions downstream of the other cascades (Figure 2D). GSK3i treatment alone induced a mild increase –albeit not significant– in origin firing distance (Figure S3D), suggesting that endogenous WNT activity is already sufficient to handle basal replicative stress.

To further validate the role of WNT signalling in DNA replication stress, we recently started a collaboration with the group of Alexander van Oudenaarden to test their newly developed single cell EdU sequencing technique (scEdU-seq, van den Berg et al., BioRxiv 2023 <https://doi.org/10.1101/2022.12.13.520365>, currently in press in Nature Methods). As shown in the new

figures 3I-K and S4, treatment with DKK1 increases the average number of replication forks per cell from 2173 to 3031, consistent with an increase in compensatory origin firing (Figure 3D). We found that fork number increase across all chromosomes (Figure 3K), especially during mid S-phase (Figure 3J).

Although no referee requested specifically this data, we think that it strengthens the manuscript by reinforcing the results of DNA combing by further highlighting the requirement of basal WNT signalling in hiPSCs for proper DNA replication progression.

5. The authors could consider adding DAPI or markers of different cell-cycle phases in Figure 3I and 3M. This additional information could provide insights into the effects of these various signaling pathway perturbations across various cell cycle phases.

We used the DAPI channel to draw the nuclear outlines (white dashed lines in the current Figure 4) to avoid saturating the images with additional channels/panels. In our initial observations, we did not detect significant changes in gH2AX in EdU negative cells, which include G1/G2 cells (as well as mitotic cells marked with condensed chromosomes). However, we believe that detailed analyses of WNT roles in e.g., DNA repair in other cell phases (requiring either synchronisation or specific markers) falls out of the scope of this manuscript.

We also included additional data related to cell cycle progression: FGF2, DKK1, and Noggin neither impacted DNA replication initiation measured by MCM2 loading during G1-S transition (Figure S5F,G), nor cell cycle progression through G1/S-phase after nocodazole arrest and release ((Figure S5I). This data further supports our initial observations that unarrested hiPSCs treated for 16h with the morphogens do not impact cell cycle phase distribution (Figure S2N).

6. Figure S5: The authors test whether treatment of small molecules or recombinant proteins perturbs chromosome missegregation during hPSC differentiation. However, many of the manipulations tested will significantly impact hPSC differentiation into the corresponding cell-types, as shown by the papers cited by the authors. This therefore brings up the possibility that the observed chromosomal phenotypes simply reflect perturbed differentiation into different cell-types, which in turn have their own chromosomal stability programs. The authors should include data on the expression of various differentiation markers after each of these perturbations.

For the DNA damage and replication stress experiments the perturbations are too short (3h) to impact fate (Figures 3-4 and Figures S4-5). The referee is right that overnight treatments required to target S-phase and subsequent mitosis could also potentially perturbate cell fate. However, as we showed in figure S2M, hiPSCs treated with DKK1, FGF2 or Noggin in E8 for 16 hours do not exit pluripotency neither upregulate early germ layer markers.

We also analysed the expression of *TBX6* and *HAND1* in mesoderm-like cells committed to lateral or paraxial mesoderm upon 16h perturbation of WNT activity (Referee figure 7). The results suggest that

changes in cell signalling in already committed cells, do not substantially modify the fate of the targeted cells. We could also include this data in the manuscript, if required.

Referee figure 7: Change in WNT activity in mesoderm-like cells committed to paraxial mesoderm (LM+DKK1 is the standard condition) or to lateral mesoderm (PM+Wnt3a is the standard condition) during the last 16 hours of differentiation does not substantially changes their initial commitment, measured by the respective gene expression markers TBX6 (paraxial mesoderm) and HAND1 (lateral mesoderm).

7. “We have previously identified that WNT ligands promote chromosome segregation fidelity in somatic and cancer cells 22-24, although the underlying mechanisms and biological relevance remained uncharacterised.” Instead of “biological”, do the authors mean “physiological” here?

We changed it.

8. Fig. 1B: There are discrepant effects of various TGFβ pathway modulators. LEFTY (TGFβ inhibitor) and two TGFβ agonists (TGFβ1 and TGFβ2) all promote chromosome mis-segregation, whereas another TGFβ agonist (ACTIVIN) suppresses chromosome mis-segregation.

There is indeed discrepancy about the role of the different TGFbeta signalling cascades on chromosome segregation fidelity. In the mammalian embryo prior gastrulation, Nodal (activator) and Lefty (inhibitor) antagonize each other driving anterior-posterior patterning. On the one hand, Lefty-A induced chromosome missegregation in hiPSCs, while Nodal and Activin had no effect, suggesting that Nodal/Activin signalling inhibition induces chromosome missegregation. On the other hand, TGFB1/2 analyses supported a role for classical TGFbeta signalling activation in promoting chromosome missegregation. Although TGFbeta, Nodal and activin canonically signal through SMAD2/3, each pathway utilises different receptors and additional downstream cascades, which could be disentangled in a dedicated study.

9. Fig. 1B: Several questions. First, the different colors used to label various circles are a bit unclear. PORCNI (a WNT pathway modulator) is labeled a “compound”, not an “antagonist”, but as the authors know, it is a WNT pathway antagonist. Second, “retinoid acid” should be corrected to “retinoic acid”. Third, should “Lefty A” be re-named to the official gene name (LEFTY2)?

We corrected the names and now indicate all antagonists and agonists with the same colour, even if they are small molecule compounds.

10. Fig. 1F: Is there a negative control condition, where no signaling modulators were added?

We now added the negative control condition to the figure (Shown in Figure S1B). We also performed a new experiment with co-treatments with anteriorising vs posteriorising signals (New Figure 1F).

11. Fig. 3A: Were hPSCs treated with DKK1, FGF2, and NOGGIN simultaneously? This should be clarified in the Figure Legend.

Cells were treated independently with each factor. We were interested in the intersection between these pathways, as they all regulated many different functions likely unrelated to the studied phenotypes. We now show i) Differentially expressed genes and GO terms for each factor (Table S2), ii) individual plots for sequencing data on factors associated with DNA replication (Figure S3B), and iii) validation of the factors from the GO term associated with DNA unwinding during replication (Figure S3A) for each pathway by qRT-PCR (Figure S3C).

12. Fig. 3B-N: How long were hPSCs treated with the various signaling modulators? This should be clarified in the Figure Legend.

For 3 hours. We now indicate it in the figure legend, as suggested.

13. Fig. 6B: The “NPCs” appear heterogeneous, with NESTIN+ NPCs and TUBB3+ neurons co-existing. Does this cellular heterogeneity affect any of the authors’ conclusions regarding “NPCs”?

As discussed above with referee #1 (Question #14), the dividing cells that we analysed were NESTIN positive and TUBB3 negative (Figure S9J,K). The surrounding TUBB3 positive cells further support that NPCs are indeed undergoing neurogenesis.

14. Figure 7I: Define what “pHis3” means, in the Figure Legend or elsewhere.

Done! (Figure legend S9 and in methods)

15. The discovery that the WNT signaling pathway is implicated in microtubule polymerization is intriguing. Perhaps in the Discussion, the authors might consider speaking about the mechanisms through which WNT acts on microtubules, as the authors’ own past work (PMID 25656539) suggested that it is independent of beta-catenin, whereas other previous studies have reported the importance of activation and phosphorylation of beta-catenin in microtubule growth (P-C Hsieh et al., 2007, Oncogene; P Huang et al., 2007, Oncogene).

We now include it in a discussion’s paragraph on beta-catenin-independent WNT signalling, as indicated above in question #1 (Page 13, Lines 512-521).

Reviewer #3 (Remarks to the Author):

Summary:

Dr de Jaime-Soguero and Hattemer et al investigate the link between lineage specification and chromosomal instability in this very interesting manuscript. Various literature supports the notion of somatic mosaicism of chromosomal alterations, and outstanding questions remain as to the mechanisms driving initial chromosome segregation abnormalities, and why or how this might vary between lineages or temporally during development. To address these questions, the authors systematically analyse the impact on chromosome segregation fidelity in a set of models to mimic various cellular pluripotency states and lineages. They test the impact of adding exogenous lineage specification signals to these cellular models and carefully score segregation error rates using microscopy. They find that during pluripotency the addition of some of these signals, but not others, induces replication stress and mitotic defects, which they link mechanistically by the induction of aberrant microtubule dynamics caused by replication stress (shown previously by one of the senior authors). This effect diminishes after pluripotency but reemerges during the differentiation pathway to neural progenitors. Overall this is a very intriguing and well performed study, that is of great interest to the development and genomic stability fields.

Thank you!

However, in places the wording is confusing, and for such a study that bridges two fields it may be more important than usual to ensure complete clarity for all readers to gain the full understanding.

We rewrote parts of the manuscript to balance detailed information on specific functions and mechanisms (required for specialists), and more general terminology and take-home messages to bridge between the fields of cell signalling, genome stability and lineage specification. We hope the referee agrees on the challenge of such task.

We also included several models across the manuscript summarising key results and tried to avoid an even broader introduction/discussion of our results, which might overwhelm the non-specialists.

I have a few concerns that would need to be addressed in order to reassure me that the conclusions are properly supported by the data:

1. The rationale to investigate roles in DNA replication could perhaps be strengthened. H3K4 methylation (di/tri?) factors are identified in the RNA analysis. Did they directly detect increased H3H4me? At promoters? Seems primarily to be a mark of transcription and maybe a minor role in replication stress? Was total gene transcription unchanged? A stronger rationale to make the link to replication stress might be helpful.

We now include a more detailed paragraph summarising our reasoning to focus on replication stress:

“Gene ontology analyses of differentially regulated genes upon endogenous WNT and BMP signalling inhibition, or FGF signalling activation, showed the highest enrichment score for factors promoting H3K4 methylation (7.5-fold, $P = 6.5 \times 10^{-07}$), which modulates gene expression and is associated with the

mitigation of replication stress (Chong et al, 2020; Petruk et al, 2012), as well as DNA unwinding factors involved in DNA replication (7.4-fold, $P = 3 \times 10^{-06}$) and ATP-dependent chromatin remodelling factors (6.8-fold, $P = 3.8 \times 10^{-08}$) (Figure 3A and Table 2), suggesting a role of these pathways in chromatin remodelling and/or DNA replication. We have previously found that i) mild DNA replication stress can lead to whole chromosome missegregation through elevated microtubule dynamics in cancer cells (Bohly et al, 2022), ii) WNT regulates microtubule plus end dynamics through unknown mechanisms (Stolz et al., 2015) and iii) direct perturbation of WNT activity only during mitosis did not induce chromosome missegregation (Bufe et al., 2021). As such, we decided to examine the possibility that WNT, BMP, and FGF signalling function in S-phase during DNA replication. Intriguingly, i) pluripotent stem cells deal with elevated replicative stress under normal self-renewing conditions (Blakemore et al, 2021; Lamm et al, 2016), and ii) replicative stress is a major driver of structural and numerical chromosomal instability (Bohly et al., 2022; Burrell et al, 2013)."

Of note, contemporary results on phospho-proteomics, DNA damage and nucleoside rescue experiments further strengthened our rationale to characterise DNA replication instead of chromatin remodelling.

The analysed pathways have critical roles in transcription regulation and/or chromatin accessibility (e.g., (Pagella et al, 2023)), which could indeed explain the modulation of H3K4 methylation beyond any link to replication. We now state first the role of this modification in transcription (See above). We did not analyse H3K4 methylation in our samples, and we think that such comprehensive epigenetic analyses fall out of the scope of our manuscript, especially considering that the follow up epistasis studies of WNT and BMP already supported a role of these factors downstream of stalled forks.

2. Page 5: "Conversely, co-treatment with WNT3A, GSK3i, or BMP4 rescued origin firing induced by 100 nM aphidicolin without affecting replication fork velocity (Figure 3E, 3F and 3G), which indicates a role of WNT and BMP signalling in the mitigation of DNA replication stress downstream of stalled forks (Figure 3H)." This is confusing. 'Rescued' origin firing might be better rephrased as 'prevented compensatory origin firing' if I have understood their data correctly. I am also not sure this can be termed 'mitigation of replication stress', since if there is still slowed forks, and now no compensatory origin firing, then it is likely the downstream consequences on genomic stability would be even worse (but this is not their observation). So there seems to be a disconnect here.

The referee is correct that WNT3A, GSK3i, or BMP4 prevent compensatory origin firing upon aphidicolin. We changed the text accordingly. They also prevented downstream signalling cascades including gH2AX and chromosome missegregation (Figure 4C,K). However, future genomic analyses are required to assess whether WNT and BMP fully prevent unrepaired DNA damage or unreplicated DNA during mitosis induced by aphidicolin, and how these translates in survival of daughter cells. Taken in account our p-MS targets (RIF1 and BRCA1; Figure S5A-E) and our CPT-rescue analyses (Figure 4E,F), additional mechanistic analyses are required to determine whether each of the analysed pathways i) might protect, restart or reverse stalled forks (e.g. upon aphidicolin) thereby preventing most insults derived from DNA replication errors, which would ensure genome stability or ii) only rescue

part of the cascade by e.g. partially overriding checkpoint mechanisms, thereby resulting in unresolved problems.

3. It is also not clear what the 'co-treatment' is with; aphidicolin? Why not co-treat DKK1/FGF2/Noggin with WNT3A/GSK3i/BMP4 as they did when assessing chromosome segregation? Would these co-treatments rescue the replication stress? If not then the effect on chromosome segregation could be via a different mechanism. This is important to strengthen their conclusion that the pathway from lineage specification signals is between replication stress, microtubule dynamics, and whole chromosome mis-segregation.

Our rationale was that it was critical to show whether WNT and BMP activation can impact replication stress driven by a known effector (aphidicolin), instead of further confirm WNT and BMP downstream cascades. As suggested by the referee, we now show the interplay between DKK1 and GSK3 in DNA combing assays. In particular, GSK3i (WNT activation) rescued DKK1-dependent decrease in origin firing distance (Figure S3D).

4. Figure 2g: Only examples of whole chromosome gains are shown. Did they ever see any structural alterations? If the mechanism is as proposed (e.g. replication stress feeds only into aberrant microtubule dynamics and mis-segregation of whole, intact chromosomes, then only whole chromosomal alterations might be expected).

In our M-FISH/Giemsa analyses, we did not detect rearrangements or clear chromosomal fragments. However, we believe that future sequencing analyses should conclusively determine whether WNT inhibition also leads to structural CIN, especially considering that morphogenetic signals driving chromosome missegregation also trigger ultra-fine bridges (UFBs) (Figure S6C,D).

Also, were there recurrent aneuploidies observed in any of the treatments? Depending on the timescale of the experiment this might reveal and positive or negative selection for specific aneuploidies which might shed light on the potential benefit of elevated chromosomal instability. How do the abnormalities they detect compare to known mosaicism where previously reported? I.e. could their proposed mechanism explain the study's initiating unexplained observation of this phenomenon?

We performed the treatments within one cell cycle (Figure S2) to reduce the possibility of negative or positive selection. Accordingly, we did not detect enrichment of any particular chromosome in the M-FISH analyses. Based on previous literature, we would expect that long-term culture of the affected could lead chromosome-specific selection. In the case of hiPSCs, numerical alterations in chromosomes 1, 12, 17 and 20 would be expected for positive selection if cells would undergo long-term culture.

Minor points:

5. No line numbers so difficult to make specific comments.

We have now marked the lines and highlighted the changes. We apologise for the inconvenience.

6. *Confusing wording in places: e.g. in abstract: “Tug-of-war between FGF and WNT triggers chromosome mis-segregation”. What does this mean? Chromosome mis-segregation is triggered when one (which one)? Is winning? Or the balance between them is varying?*

We now indicate: *“Through ex vivo and in vivo analyses, we also show that FGF and WNT signalling display opposite roles in chromosome segregation fidelity in mouse neural progenitors during the onset of neurogenesis (E14.5), but not during their expansion (E12.5). In particular, we find that the neurogenic factor FGF2 induces DNA replication stress-mediated chromosome missegregation, which could provide a rationale for the elevated chromosomal mosaicism of the developing brain”.*

7. *Figure 3K,L,N: the numbers of H2AX/RPA foci are very high (reaching into the thousands). Can they show example segmented images? Are they sure these are real foci?*

The DNA damage analyses (gamma-H2AX and phospho-RPA) are displayed as median fluorescence intensity (MFI) instead of foci number. Please see our response to referee #1, point #6 and Referee Figure 2 for the details.

8. *Figure 8 legend: “Cell signalling control of chromosome segregation fidelity is lost after specification into the three germ layers following the withdrawal of ATM signalling as first responder during DNA replication stress, but reemerges during neurogenesis” confusing. Figure looks like fidelity is lost again upon neurogenesis (why?)*

We refer to the capacity of influence (positively or negatively) chromosome segregation.

In our *ex vivo* experiments in Figure 7, we show that treatment with FGF2, a physiological growth factor during neurogenesis, notably at E14.5, is sufficient to induce chromosome missegregation in mNPCs. Indeed, *in vivo* NPCs undergoing neurogenesis in E14.5 under high FGF signalling, as well as *in vitro* hNPCs and *ex vivo* mNPCs subjected with physiological levels of FGF2 present high levels of chromosome missegregation, which are not countered by endogenous WNT activity (only upon WNT3A or GSK3i exogenous addition, See Figure 6I and 7G). We think the model in Figure 8 reinforces the concept that FGF-driven neurogenesis also contributes to the notoriously high levels of chromosome missegregation in NPCs.

We rephrased the legend as “The capacity of investigated extracellular signals to influence chromosome segregation fidelity is largely lost after specification into the three germ layers following the withdrawal of ATM signalling as first responder during DNA replication stress, but reemerges during neurogenesis. In particular, a FGF signalling induces high levels of chromosome missegregation of neural progenitors committed to neurogenesis”

9. *Hard to follow the logic of paragraph starting on bottom of page 10. Could expand and clarify.*

We now further indicate in the discussion (Page 13, Lines 526-531) that ATM activity is critical during gastrulation and in neural progenitors, but no other lineages, which could further explain why the signals only plug into the regulation of genome stability in these two developmental stages/lineages.

10. The overall larger picture and logic of this intriguing concept is a bit lacking or confusing. Particularly since this work combines two fields, and most readers will be from one field or the other, it is particularly important to explain this intersect in very clear simple language. They seem to allude to a greater concept combining cell fate signalling and extra cellular clues and development in the control of accurate replication and segregation (e.g. last sentence of discussion). However it is not really clear what the rationale/benefit for such an entanglement. Why is increased replication stress or aneuploidy beneficial? If this leads to an increase in cell death how does this play in the scheme of development or cell fate? Could this instead be an unintentional consequence of lineage specification that is under the tolerable limit? The discussion seems to be hinting that diversification could occur at the cost of aneuploidy, but it is not clear what sort of diversification this means, if aneuploid cells are ultimately removed (as they also mention). Alternative (smaller) chromosomal alterations? A clearer and simpler discussion or speculation on these issues could help place this work in context. Perhaps even some of this literature could be explained in the introduction.

While the protective nature of e.g., WNT activity could be easily understood as a new layer of control/protection for genome stability, it is less clear to us why development would “allow” the nefarious consequences of high DKK1 and Noggin during anteriorisation or high FGF activity during neurogenesis. The referee is correct that it is possible that these errors can be tolerated in those lineages as possible trade off, or because follow up repair/surveillance mechanisms could largely correct/cleanse them. While aneuploidy usually reduces fitness, recent evidence shows that replicative stress during neurogenesis results in clusters of mutations in specific neuronal genes, which might contribute to neuronal variability (Wei *et al*, 2016; Wu *et al*, 2021). Although this is a provocative idea, we think it would be relevant to the reader to be aware of the concept, as well as for follow up studies to consider the possibility of replicative stress not only as a bug, but perhaps in some cases as a feature. We also briefly indicate now that these cells might just tolerate RS, in exchange of e.g., required fast proliferation (See Discussion, Page 14, Lines 556-561).

We think it could take several studies to understand why patterning signals are linked to chromosome segregation fidelity during early lineage specification and neurogenesis, as well as their impact during the organismal lifetime.

11. Bottom of page 4, used SiR-DNA to track chromosome mis-segregation which cannot distinguish between whole, or large fragments of chromosomes, so cannot be certain these are whole chromosome mis-segregation events. This would need to be determined using either a second live cell marker for the centromere (such as CENP-A-GFP) or with fixed cell immunofluorescence against centromere proteins.

We agree with the referee that the live imaging analyses in Figure S2 have a lower capacity to resolve different segregation defects. However, this is just complementary figure in the context of chromosome segregation (no conclusion on chromosome segregation fidelity is based on the figure), and rather to show that cells treated with DKK1, FGF2 and Noggin progress through mitosis even after missegregation events. We can remove the quantification of chromosome missegregation and leave

the quantifications of mitotic progression (Figure S2I,J). However, we think the readers would appreciate a live cell imaging experiment, even if it cannot fully distinguish between whole vs fragments. Please also see our answer to the next question.

12. During the study, the authors focus on whole chromosome mis-segregation events arising from replication stress-induced increases in MT dynamics. Is the incidence of RS-induced anaphase bridges and acentric lagging chromosome fragments negligible in these cell lines? It is not clear whether they scored these types of errors. If not, then presumably the levels of RS induced by DKK1/FGF2/Noggin are not severe enough to induce these errors?

We now also show the effect of aphidicolin in MT dynamics (Figure 4I), which we think it would help the readers unfamiliar with this S- to M-phase connection.

As indicated in the methods, for the immunofluorescence analyses, we scored “chromosomes clearly separated (Lagging) from the bulk of segregated DNA chromatids were considered as chromosome missegregation”.

We now include in Figure S6A,B detail analyses of CENPC positive and CENPC negative (likely acentric) missegregated chromosomes pooled from 3 independent experiments. As discussed in page 8, line 323, DKK1, Noggin or FGF2 treatment increased the proportion of anaphases with acentric chromosomes from 0.7% to 2-3% (Figure S6A), representing ~15% of the total missegregated chromosomes across experiments. In agreement with a more severe effect during DNA replication (Figure 3D), treatment with 50 nM Aphidicolin resulted in 5% of anaphases displaying acentric chromosomes, representing ~35% of the total missegregated chromosomes (Figure S6B).

We also performed BLM staining analyses and found that -interestingly- DKK1, Noggin or FGF2 treatments lead to the formation of ultra-fine bridges during anaphase in hiPSCs (Figure S6C and S6D) to similar extend as 50 nM aphidicolin.

Taken together these results suggest that replicative stress associated with DKK1, Noggin or FGF2 is sufficient to induce UFBs and slightly increase the missegregation of acentric chromosomes. Given that hiPSCs have high basal levels of RS, it would be interesting to disentangle the thresholds and dynamics of different missegregation effects upon different sources of replicative stress. Furthermore, as indicated above in limitations section, follow up sequencing studies should shed light on possible structural defects associated with cell signalling perturbation in pluripotent stem cells (Page 15, Lines 588-589).

13. Results section paragraph 1 – should “Table 1” refer to Supplementary Table 1?

Corrected

14. Fig 1C – right side y axis label should read Relative AXIN2 expression.

Corrected

15. There is no reference to Fig 1E in the text.

Added to the results section and discussion of anteriorising vs posteriorising signals.

16. Fig 2F-G – is whole chromosome loss ever observed in these cells following DKK1/FGF2/Noggin treatment?

We also identified a similar increase in chromosome losses in the karyotype analyses upon DKK1, FGF2, Noggin. However, chromosome spreads can stochastically result in chromosome losses during the preparation phase. Accordingly, basal levels of “chromosome losses” were >10%. As such, and building on previous experience (de Carcer *et al*, 2018), we only plotted gains.

17. Fig 4C – 10 nM Taxol (16hrs) isn't a particularly “low dose” since much lower concentrations have been used in previous studies (e.g. 0.2 – 0.5 nM in Stolz *et al*. (2015) *EMBO Reports*). What is the reason for selecting the concentration used here? Are iPSCs inherently less sensitive to Taxol? Would a sub-nanomolar concentration of Taxol still rescue chromosome missegregation?

Because handling hiPSCs is more time consuming than RPE1/HCT116 cells, we settle for the highest concentration of taxol that allowed cell cycle progression. For your perusal, we have titrated down taxol and found that 0.5 nM was sufficient to rescue DKK1 in hiPSCs (Referee Figure 8).

Referee figure 8: Subnanomolar concentration of Taxol rescues chromosome missegregation induced by DKK1 in hiPSCs.

18. Fig 5B – drawing conclusions from phosphorylation blots in the absence of a blot for total levels of the protein in question should be discouraged. Total CHK1 and total CHK2 western blots should be included to confirm the authors' hypothesis regarding a switch from ATM to ATR signalling.

We include the total CHK1/2 blots showing that CHK2 is still present in differentiated cells (albeit showing a slight reduction in protein levels); (Figure 5E).

19. The fork progression rate in untreated hiPSCs is 0.75 kb/min (Fig 3C), but in d16 hiNPCs the control fork rate is ~1.5 kb/min (Fig 6E). Is this increased fork speed with differentiation expected? Could the authors speculate on the reasons for the observed increase?

Recent studies proposed an acceleration of fork speed in mouse embryos from zygote to blastocyst (Nakatani *et al*, 2022). We could not find a comprehensive/comparative study on fork velocity analysis after gastrulation or in neuronal lineage to compare with our results. We think it would be interesting to analyse in future studies how replication dynamics change across human lineages and which intrinsic and extrinsic factors might impact those dynamics. In the context of NPCs, we think that this could be caused by short S-phase during their commitment to neurons (Arai *et al*, 2011). We added this comment to the data:

“Of note, hiNPCs displayed higher replication speed compared to hiPSCs, possibly due to the shortening of S-phase during their commitment to neurogenesis (Arai *et al.*, 2011).”

Please note that we revisited our fork speed analyses and identified a minor error in conversion of the measures of the original figure of replication fork speed in hiPSCs, which is now further validated and corrected in new Figure 3C.

20. Please ensure all IF images include scale bars with sizes clearly stated in the figure legends.

Done!

21. Methods section, Protein and small compound treatments heading, line 1 – please correct references to figures (e.g. there is no Fig 2J or 4G).

We treated cells for 3 hours for direct S-phase analyses and for 16 hours for chromosome segregation and other mitotic analyses to ensure that cells underwent S-phase during the treatments. We now indicate the specific times in each figure legend instead of the methods.

Reviewer #4 (Remarks to the Author):

*I have a few comments regarding the phosphoproteomics work mentioned in the manuscript by De Jaime-Soguero *et al*.*

1. In the methods section, the details of MS experiments are missing. The reader is guided to reference 36, which itself redirects to another reference. I was able to find some detail in the supplemental document of reference 36, however, we cannot expect the reader of the current manuscript to keep digging. Please include either in the Method section itself or as a supplement, the complete phosphoproteomic workflow, including peptide digestion, mass spectrometer method and parameters, and data analysis steps.

The referee is right. We now explain in Methods section (Page 48, Lines 1523-1547) the whole workflow of the experiment only refereeing to specific external and primary sources when specifically required.

2. Was the phosphoproteomic analysis quantitative, in other words, was SILAC or another labeling incorporated in the samples? I assume it was, since differentially regulated phosphopeptides were shown in Figure S3. In that case, please add that information for the reader.

Quantification was done using a label free quantification approach based on the MaxLFQ algorithm (Cox *et al*, 2014) at the DKFZ proteomics facility. Given that a minimum number of quantified peptides is required for protein quantification using this method, only 7394 proteins could have been quantified, for which 4140 were quantified in all samples.

We added this information in the methods section and in the main text.

References of the rebuttal letter

- Acebron SP, Karaulanov E, Berger BS, Huang Y-L, Niehrs C (2014) Mitotic wnt signaling promotes protein stabilization and regulates cell size. *Molecular cell* 54: 663-674
- Ahuja AK, Jodkowska K, Teloni F, Bizard AH, Zellweger R, Herrador R, Ortega S, Hickson ID, Altmeyer M, Mendez J *et al* (2016) A short G1 phase imposes constitutive replication stress and fork remodelling in mouse embryonic stem cells. *Nat Commun* 7: 10660
- Alberici P, Fodde R (2006) The role of the APC tumor suppressor in chromosomal instability. *Genome Dyn* 1: 149-170
- Arai Y, Pulvers JN, Haffner C, Schilling B, Nusslein I, Calegari F, Huttner WB (2011) Neural stem and progenitor cells shorten S-phase on commitment to neuron production. *Nat Commun* 2: 154
- Augustin I, Dewi DL, Hundshammer J, Erdmann G, Kerr G, Boutros M (2017) Autocrine Wnt regulates the survival and genomic stability of embryonic stem cells. *Sci Signal* 10
- Blakemore D, Vilaplana-Lopera N, Almaghrabi R, Gonzalez E, Moya M, Ward C, Murphy G, Gambus A, Petermann E, Stewart GS *et al* (2021) MYBL2 and ATM suppress replication stress in pluripotent stem cells. *EMBO Rep* 22: e51120
- Bohly N, Schmidt AK, Zhang X, Slusarenko BO, Hennecke M, Kschischo M, Bastians H (2022) Increased replication origin firing links replication stress to whole chromosomal instability in human cancer. *Cell Rep* 41: 111836
- Bufe A, Garcia Del Arco A, Hennecke M, de Jaime-Soguero A, Ostermaier M, Lin YC, Ciprianidis A, Hattemer J, Engel U, Beli P *et al* (2021) Wnt signaling recruits KIF2A to the spindle to ensure chromosome congression and alignment during mitosis. *Proc Natl Acad Sci U S A* 118
- Burrell RA, McClelland SE, Endesfelder D, Groth P, Weller MC, Shaikh N, Domingo E, Kanu N, Dewhurst SM, Gronroos E *et al* (2013) Replication stress links structural and numerical cancer chromosomal instability. *Nature* 494: 492-496
- Chong SY, Cutler S, Lin JJ, Tsai CH, Tsai HK, Biggins S, Tsukiyama T, Lo YC, Kao CF (2020) H3K4 methylation at active genes mitigates transcription-replication conflicts during replication stress. *Nat Commun* 11: 809
- Cox J, Hein MY, Lubner CA, Paron I, Nagaraj N, Mann M (2014) Accurate proteome-wide label-free quantification by delayed normalization and maximal peptide ratio extraction, termed MaxLFQ. *Mol Cell Proteomics* 13: 2513-2526
- Davenport C, Diekmann U, Budde I, Detering N, Naujok O (2016) Anterior-Posterior Patterning of Definitive Endoderm Generated from Human Embryonic Stem Cells Depends on the Differential Signaling of Retinoic Acid, Wnt-, and BMP-Signaling. *Stem Cells* 34: 2635-2647
- de Carcer G, Venkateswaran SV, Salgueiro L, El Bakkali A, Somogyi K, Rowald K, Montanes P, Sanclemente M, Escobar B, de Martino A *et al* (2018) Plk1 overexpression induces chromosomal instability and suppresses tumor development. *Nat Commun* 9: 3012
- Edri R, Yaffe Y, Ziller MJ, Mutukula N, Volkman R, David E, Jacob-Hirsch J, Malcov H, Levy C, Rechavi G *et al* (2015) Analysing human neural stem cell ontogeny by consecutive isolation of Notch active neural progenitors. *Nat Commun* 6: 6500

- Fodde R, Kuipers J, Rosenberg C, Smits R, Kielman M, Gaspar C, van Es JH, Breukel C, Wiegant J, Giles RH *et al* (2001) Mutations in the APC tumour suppressor gene cause chromosomal instability. *Nat Cell Biol* 3: 433-438
- Habib SJ, Acebron SP (2022) Wnt signalling in cell division: from mechanisms to tissue engineering. *Trends Cell Biol*
- Hadjihannas MV, Bruckner M, Jerchow B, Birchmeier W, Dietmaier W, Behrens J (2006) Aberrant Wnt/beta-catenin signaling can induce chromosomal instability in colon cancer. *Proc Natl Acad Sci U S A* 103: 10747-10752
- Huang P, Senga T, Hamaguchi M (2007) A novel role of phospho-beta-catenin in microtubule regrowth at centrosome. *Oncogene* 26: 4357-4371
- Kastan N, Gnedeva K, Alisch T, Petelski AA, Huggins DJ, Chiaravalli J, Aharanov A, Shakked A, Tzahor E, Nagiel A *et al* (2021) Small-molecule inhibition of Lats kinases may promote Yap-dependent proliferation in postmitotic mammalian tissues. *Nat Commun* 12: 3100
- Kikuchi K, Niikura Y, Kitagawa K, Kikuchi A (2010) Dishevelled, a Wnt signalling component, is involved in mitotic progression in cooperation with Plk1. *EMBO J* 29: 3470-3483
- Lamm N, Ben-David U, Golan-Lev T, Storchova Z, Benvenisty N, Kerem B (2016) Genomic Instability in Human Pluripotent Stem Cells Arises from Replicative Stress and Chromosome Condensation Defects. *Cell Stem Cell* 18: 253-261
- Lin YC, Haas A, Bufe A, Parbin S, Hennecke M, Voloshanenko O, Gross J, Boutros M, Acebron SP, Bastians H (2021) Wnt10b-GSK3beta-dependent Wnt/STOP signaling prevents aneuploidy in human somatic cells. *Life Sci Alliance* 4
- Liu Y, Liu H, Sauvey C, Yao L, Zarnowska ED, Zhang SC (2013) Directed differentiation of forebrain GABA interneurons from human pluripotent stem cells. *Nat Protoc* 8: 1670-1679
- Maya-Mendoza A, Moudry P, Merchut-Maya JM, Lee M, Strauss R, Bartek J (2018) High speed of fork progression induces DNA replication stress and genomic instability. *Nature* 559: 279-284
- McLin VA, Rankin SA, Zorn AM (2007) Repression of Wnt/beta-catenin signaling in the anterior endoderm is essential for liver and pancreas development. *Development* 134: 2207-2217
- Nakatani T, Lin J, Ji F, Ettinger A, Pontabry J, Tokoro M, Altamirano-Pacheco L, Fiorentino J, Mahammadov E, Hatano Y *et al* (2022) DNA replication fork speed underlies cell fate changes and promotes reprogramming. *Nat Genet* 54: 318-327
- Orlando L, Tanasijevic B, Nakanishi M, Reid JC, Garcia-Rodriguez JL, Chauhan KD, Porras DP, Aslostovar L, Lu JD, Shapovalova Z *et al* (2021) Phosphorylation state of the histone variant H2A.X controls human stem and progenitor cell fate decisions. *Cell Rep* 34: 108818
- Pagella P, Soderholm S, Nordin A, Zambanini G, Ghezzi V, Jauregi-Miguel A, Cantu C (2023) The time-resolved genomic impact of Wnt/beta-catenin signaling. *Cell Syst* 14: 563-581 e567
- Petruk S, Sedkov Y, Johnston DM, Hodgson JW, Black KL, Kovermann SK, Beck S, Canaani E, Brock HW, Mazo A (2012) TrxG and PcG proteins but not methylated histones remain associated with DNA through replication. *Cell* 150: 922-933
- Rashid MS, Mazur T, Ji W, Liu ST, Taylor WR (2018) Analysis of the role of GSK3 in the mitotic checkpoint. *Sci Rep* 8: 14259
- Ross J, Busch J, Mintz E, Ng D, Stanley A, Brafman D, Sutton VR, Van den Veyver I, Willert K (2014) A rare human syndrome provides genetic evidence that WNT signaling is required for reprogramming of fibroblasts to induced pluripotent stem cells. *Cell Rep* 9: 1770-1780
- Rusan NM, Peifer M (2008) Original CIN: reviewing roles for APC in chromosome instability. *J Cell Biol* 181: 719-726
- Stolz A, Neufeld K, Ertych N, Bastians H (2015) Wnt-mediated protein stabilization ensures proper mitotic microtubule assembly and chromosome segregation. *EMBO Rep* 16: 490-499
- Tchieu J, Zimmer B, Fattahi F, Amin S, Zeltner N, Chen S, Studer L (2017) A Modular Platform for Differentiation of Human PSCs into All Major Ectodermal Lineages. *Cell Stem Cell* 21: 399-410 e397
- Tighe A, Ray-Sinha A, Staples OD, Taylor SS (2007) GSK-3 inhibitors induce chromosome instability. *BMC Cell Biol* 8: 34

Wei PC, Chang AN, Kao J, Du Z, Meyers RM, Alt FW, Schwer B (2016) Long Neural Genes Harbor Recurrent DNA Break Clusters in Neural Stem/Progenitor Cells. *Cell* 164: 644-655

Wu W, Hill SE, Nathan WJ, Paiano J, Callen E, Wang D, Shinoda K, van Wietmarschen N, Colon-Mercado JM, Zong D *et al* (2021) Neuronal enhancers are hotspots for DNA single-strand break repair. *Nature* 593: 440-444

REVIEWERS' COMMENTS

Reviewer #1 (Remarks to the Author):

The new version of the manuscript, as well as the comments to the referees reports have strengthen the work and I believe that it is now ready for publication

Reviewer #2 (Remarks to the Author):

The authors have satisfactorily addressed our comments, and should be congratulated for their intriguing contribution to the scientific literature.

Reviewer #3 (Remarks to the Author):

The authors have satisfactorily addressed all my initial comments, and I congratulate them on a very interesting in-depth study of this interesting phenomenon.

Only one minor comment in case it is helpful: : In line 106 the term 'ploidy' is confusing here – implying control of cell ploidy (eg diploidy, tetraploidy) rather than control of chromosomal stability which is what I think they mean. Perhaps the term euploidy could be used, or chromosomal/karyotypic stability.

Reviewer #4 (Remarks to the Author):

I thank the authors for addressing my questions. The added information really helps the reader understand the complexity of the experiments.

Response of the referees to the addressed points (final response)

Dear Editor and Referees,

Thank you for your critical input, which we believe has substantially improved our manuscript.

REVIEWERS' COMMENTS

Reviewer #1 (Remarks to the Author):

The new version of the manuscript, as well as the comments to the referees reports have strengthen

the work and I believe that it is now ready for publication

Thank you!

Reviewer #2 (Remarks to the Author):

The authors have satisfactorily addressed our comments, and should be congratulated for their intriguing contribution to the scientific literature.

Thank you!

Reviewer #3 (Remarks to the Author):

The authors have satisfactorily addressed all my initial comments, and I congratulate them on a very

interesting in-depth study of this interesting phenomenon.

Thank you!

Only one minor comment in case it is helpful: In line 106 the term 'ploidy' is confusing here – implying

control of cell ploidy (eg diploidy, tetraploidy) rather than control of chromosomal stability which is what I think they mean. Perhaps the term euploidy could be used, or chromosomal/karyotypic stability.

We changed it accordingly

Reviewer #4 (Remarks to the Author):

I thank the authors for addressing my questions. The added information really helps the reader understand the complexity of the experiments.

Thank you!